# Widespread horse-based mobility arose around 2200 BCE in Eurasia

Horses revolutionized human history with fast mobility[1]. However, the timeline between their domestication and their widespread integration as a means of transport remains contentious[2–4]. Here we assemble a collection of 475 ancient horse genomes to assess the period when these animals were first reshaped by human agency in Eurasia. We find that reproductive control of the modern domestic lineage emerged around 2200 BCE, through close-kin mating and shortened generation times. Reproductive control emerged following a severe domestication bottleneck starting no earlier than approximately 2700 BCE, and coincided with a sudden expansion across Eurasia that ultimately resulted in the replacement of nearly every local horse lineage. This expansion marked the rise of widespread horse-based mobility in human history, which refutes the commonly held narrative of large horse herds accompanying the massive migration of steppe peoples across Europe around 3000 BCE and earlier[3,5]. Finally, we detect significantly shortened generation times at Botai around 3500 BCE, a settlement from central Asia associated with corrals and a subsistence economy centred on horses[6,7]. This supports local horse husbandry before the rise of modern domestic bloodlines.

The genetic make-up of modern domestic horses (hereafter, DOM2) emerged in the western Eurasian steppes during the third millennium BCE[2]. The spread of DOM2 horses, alongside the development of Sintashta spoke-wheeled chariots in Asia (around 2200–1800 BCE) and the apparently limited DOM2 genetic influence in Europe before that time, has indicated that long-distance horse-based mobility developed no earlier than the late third millennium BCE. This chronology implies that the spread of steppe-related ancestry that reshaped the human genetic landscape of nearly all regions of central and western Europe over the course of the third millennium BCE[8,9] was not driven by DOM2 horseback riding.

However, recent population models have claimed significant DOM2 genetic ancestry into European horses affiliated with the Corded Ware complex (CWC), a culture that developed from roughly 3000 BCE against the backdrop of the Yamnaya steppe migration[4]. Bone pathologies potentially resulting from regular horseback riding also occur in about 5% of the human skeletons from the Carpathian Basin, mainly in steppe-related[8] Yamnaya individuals, but also in pre-Yamnaya people, up to the fifth millennium BCE[5]. Moreover, horse-related terminology commonly shared across Indo-European languages is often considered indicative of established equestrianism in the steppes, among Yamnaya-related proto-Indo-European speakers[3]. These findings have revived theories associating horseback riding with the Yamnaya expansion[3], and possibly with earlier human steppe migrations into the Carpathian Basin after about 4500 BCE[10].

Whether or not rapid mobility was the only incentive for horse domestication is also a matter of controversy. Equine milk peptides were reported in Yamnaya human dental calculus from around 3300–2600 BCE[11], but further work has shown that western steppe pastoral practices shifted from sheep and cattle dairying to horse milking no earlier than around 1000 BCE[12]. Archaeological evidence for pre-Yamnaya horse milking and harnessing[6,7] exists further east in central Asia, in the 5,500-year-old Botai culture, which developed a subsistence economy almost entirely focused on horses[13]. At this site, evidence for horse milk consumption is supported by residue analysis of fatty acids absorbed into pottery shards ($n = 5$), but this is not corroborated by the palaeoproteomic analysis of human dental calculus ($n = 2$)[6,11,14].

Furthermore, the unusual pattern of dental attrition on Botai horse teeth was initially identified as bit wear[15], but this interpretation has since been challenged[16]. Unchanged sex ratios in pre Botai and Botai bone assemblages have also advocated against the emergence of new horse management practices at Botai[17,18]. Considering that DOM2 and Botai horses originate from two genetically distinct lineages[7], new evidence is needed to assess the exact part played by horses in Botai society, and, more generally, how domestic horses contributed to the steppe migrations and the possibly concurrent spread of Indo-European languages (although see ref. 19).

## Datasets and experimental design

To address the context in which horse husbandry developed in the fourth and third millennia BCE, we analysed 475 ancient horse genomes (Fig. 1a), combined with 77 publicly available modern horse genomes, including 40 worldwide domestic breeds and 6 endangered Przewalski's horses (Supplementary Table 1 and Extended Data Figs. 1 and 2). The 124 newly generated genomes show a median coverage of 1.40-fold (minimum 0.29; maximum 10.92) and span Eurasian archaeological contexts dating to more than 50,000 years ago, including in the Carpathian Basin, where bioanthropological evidence for horseback riding was reported[5,20]. Together with 401 radiocarbon dates, 140 of which are new, our dataset provides an unprecedented genome time series spanning the whole domestication process.

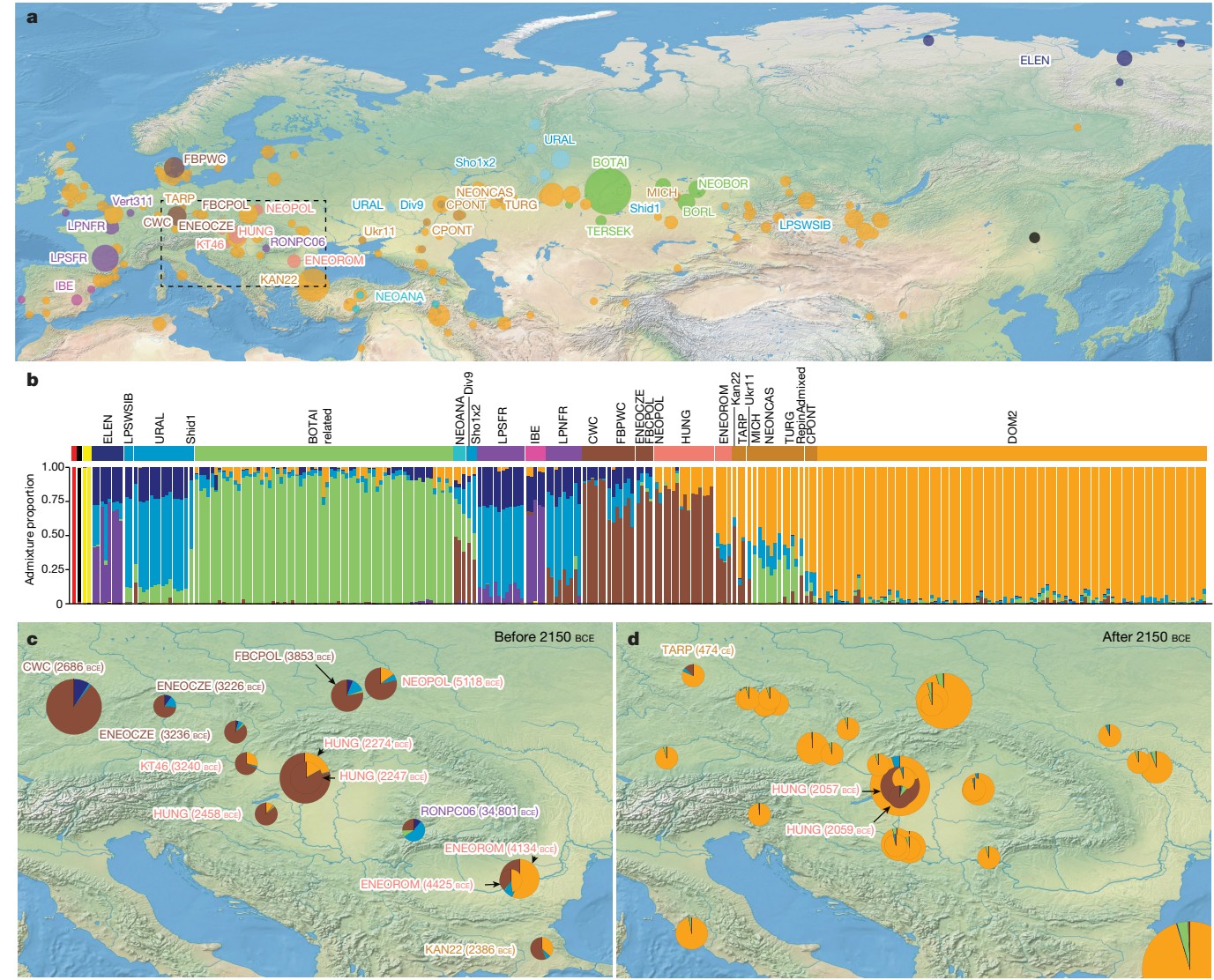

**Fig. 1 | Geographic distribution and genetic profiles of the 475 ancient horse genomes analysed in this study. a**, Geographic location of the archaeological sites. The size of each location is proportional to the number of horse genomes sequenced. The black dot points to the location of *E. ovodovi* outgroups. **b**, Struct-f4 genetic ancestry profiles considering *K* = 9 components. The top panel provides the colour legend for **a**. **c,d**, Genetic ancestry profiles (*K* = 9) across central Europe, the Carpathian and Transylvanian Basins before (**c**) and after (**d**) 2150 BCE. The midpoint of the radiocarbon dating range obtained for each site is indicated between parentheses.

In this study, we investigate three possible markers of horse husbandry. First, we examine changes in the genomic make-up of horses across central and eastern Europe to test whether they accompanied the humans who moved from the steppe. Second, we reconstruct horse demographic trajectories to evaluate the existence, timing and severity of domestication bottlenecks. This shows when horses were bred in significant numbers to sustain large-scale mobility. Third, we track evidence for controlled reproduction of horses, in the form of close-kin mating and accelerated generation times.

## Spread of DOM2 horses across Europe

Assuming that steppe humans and horses moved together implies parallel shifts of genetic ancestry in both species. Such concurrent shifts were supported by the population graphs presented by Maier et al.[4], who identified horses excavated from a CWC context in Germany with roughly 20% DOM2 ancestry, somehow mirroring the approximately 70% Yamnaya-related steppe ancestry observed in humans[8]. However, Locator[21] analyses predict that the geographic origin of CWC horses is exclusively within central Europe (Extended Data Fig. 3c,d). We also identify population graphs fitting published data significantly better than those previously proposed[2,4] (*P* < 10^{-5}; Extended Data Fig. 3b), and refining our understanding of the connectivity between the steppes and the rest of Europe by including four extra population groups (Extended Data Fig. 4). No such graphs support DOM2 genetic contribution to CWC horses (Extended Data Figs. 3a,b and 4), with the most comprehensive placing CWC horses close to pre-Yamnaya populations from central Europe (ENEOCZE, around 3364–3102 BCE, and NEOPOL, around 5210–5006 BCE). That a central European horse lineage remained isolated from the steppe is also supported by adjacent positioning in multidimension scaling analysis (Extended Data Fig. 5), distinctive ancestry profiles sharing the main genetic component of CWC horses (Fig. 1b,c and Extended Data Fig. 6) and qpAdm modelling (Supplementary Table 2). qpAdm models including two population sources depict CWC horses as a mixture between ENEOCZE (32.4%) and northern European horses (FBPWC, around 3050–2950 BCE; 67.6%), whereas allowing for a third source returns negligible steppe contribution (less than or

equal to 1.7%). Combined, these analyses uncover a distinct cline of genetic ancestry peaking in CWC horses and declining both westwards (LPNFR, around 13969–12090 BCE) and eastwards across central Europe (ENEOCZE and NEOPOL), the Carpathian and Transylvanian Basins (HUNG, around 3364–1971 BCE, and ENEOROM, around 4494–3658 BCE) and Anatolia (NEOANA, around 6396–4456 BCE) (Fig. 1b,c).

A substantial proportion of the CWC-related ancestry survives in wild European horses called 'tarpans' (about 45.1%) until roughly 1868 CE in our dataset (and possibly later in the last surviving captive or free-ranging tarpans[22]), but is at best residual in the genetic make-up of modern domestic horses (Fig. 1b). In fact, it vanishes with the expansion of the typical DOM2 ancestry profile outside the steppe (Fig. 1c). Our extended time-stamped panel of ancient genomes from the Carpathian Basin provided increased temporal resolution regarding the arrival of DOM2 horses and the replacement of the local lineage found there (HUNG). This is pivotal for clarifying the role of horses in human migrations from the steppe. The date for the first typical DOM2 horse in the Carpathian Basin is approximately 1822 BCE (1895–1749 BCE), whereas that for the last horse with a typical local HUNG genetic profile is around 2033 BCE (2120–1945 BCE). Considering individual archaeological sites, rather than the whole region, indicates similar chronologies (at Budapest-Királyok Útja: about 1822 BCE (1895–1749 BCE) versus about 2211 BCE (2284–2138 BCE); at Százhalombatta-Földvár: about 1822 BCE (1893–1751 BCE) versus about 2033 BCE (2120–1945 BCE))) (Supplementary Table 1). Combined, these findings narrow down the time for the genomic turnover accompanying the arrival of DOM2 horses in the Carpathian Basin to roughly 2033–1945 BCE. This timeline is consistent with the first evidence of DOM2 horses outside the steppe, reported by Librado et al.[2], in Moldavia around 2063 BCE (2140–1985 BCE), Anatolia around 2125 BCE (2205–2044 BCE) and Czechia around 2037 BCE (2137–1936 BCE), post-dating the arrival of human steppe-related ancestry in the respective regions by at least 600 years[10,23]. Yamnaya-related steppe migrations and the spread of DOM2 horses are, thus, chronologically incompatible.

However, humans may have migrated from the steppe using horses other than DOM2. To investigate this, we mapped the genetic ancestry identified by Struct-f4 (ref. 24) as characteristic of horse populations living across the steppe before the expansion of DOM2 (CPONT, TURG and NEONCAS; roughly 5616–2636 BCE; Fig. 1b). Around 17.2% of this ancestry was present in the Carpathian Basin during the fourth and third millennia BCE (around 3364–1971 BCE). However, we find it also in Austria about 3300 BCE (28.9%, KT46), and in the Transylvanian Basin about 4200 BCE (54.5%, ENEOROM), at the Pietrele site where the genomic make-up of human populations suggests no steppe contact[10]. In fact, the steppe-related genetic ancestry is found in even earlier horse populations spanning a broad geographic range, including Poland (NEOPOL, around 5210–5006 BCE), Anatolia (NEO-ANA, around 6396–4456 BCE) and Iberia (IBE, around 5299–1900 BCE), and as far back in time as in the Upper Palaeolithic of France (LPNFR, around 13969–12090 BCE; LPSFR, around 21909–14646 BCE). This is consistent with the best-fitting population graph showing ENEOROM horses receiving steppe genetic material from an ancestor that also contributed to LPSFR populations (Extended Data Fig. 4). Therefore, the spread of steppe-related horse genetic ancestry into Europe must predate about 14646 BCE, which is considerably earlier than any claimed evidence for horse husbandry[3], and, thus, occurred through natural contacts between wild populations, most probably dispersing in the aftermath of the Last Glacial Maximum (roughly 24000–17500 BCE)[25]. Combined, the genomic make-up of ancient European horses does not endorse widespread horse-driven mobility before the end of the third millennium BCE. It thus dismisses any substantial involvement of horses in the Yamnaya-related or earlier human migrations from the steppe.

## DOM2 demographic history

To time precisely the rise of widespread horse-based mobility, we next estimated the period when DOM2 horses were bred in sufficiently large numbers to sustain their global spread. Specifically, we tracked changes in the DOM2 effective population size ($N_e$) during the 200 generations preceding about 1864 BCE, which is the average date of the earliest 24 DOM2 horses in our dataset with sufficient sequence data (Fig. 2a). Crucially, linkage disequilibrium-based demographic reconstructions[26] indicate a sharp demographic burst of about 13.7-fold increase within the 30 generations preceding that period. Matching those 30 generations with the Yamnaya-related steppe expansion, which had already reached central Europe by about 2750 BCE at the latest[8], would require unrealistic average generation times of roughly 27 years, largely exceeding horse life expectancy under modern intensive veterinary care[27,28]. Assuming instead the commonly accepted generation time of 8 (7–12) years[29–32] leads to about 2190 (2310–2160) BCE for the rise of widespread horse-based mobility. Restricting analyses to horses from Sintashta contexts, which are associated with the spread of spoke-wheeled chariots in Asia, returns similar demographic profiles and time estimates (about 2100 BCE (2200–2075 BCE); Extended Data Fig. 7a). These timelines coincide not only with the radiocarbon dating of the earliest DOM2 horses outside the steppe, but also with the earliest horse images in Akkadian art[33,34], and with major evidence of conflicts, crises and political disruption, from the Balkans to Egypt and the Indus valley[35,36].

Our demographic reconstructions also provide evidence for a strong domestication bottleneck in horses during the 75 generations preceding the DOM2 expansion (Fig. 2a). The interval associated with minimal effective sizes ($N_e \approx 500$ diploid individuals) starts about 2664 (3064–2564) BCE. Therefore, the time when steppe people migrated did not coincide with expanding, but rather plummeting, availability of DOM2 reproductive horses, which aligns with horses not driving Yamnaya-related steppe migrations. Interestingly, the first evidence for horses carrying long runs of homozygosity (ROHs) only (greater than or equal to 15 cM), which is indicative of close-kin mating, is found in some of the earliest DOM2 sequenced (Fig. 2c), including in the steppes of central Asia and Anatolia. This indicates that the reproductive control underlying early DOM2 spread involved some levels of inbreeding, which is avoided in the wild, but is a common practice when breeding animals for desirable traits[37].

## DOM2 generation time contracted 2200 BCE

In addition to the practice of close-kin mating, early DOM2 breeders may have aimed to produce more animals every year to meet the explosive demand for horses in the late third millennium BCE. To test whether breeders used younger animals for reproduction, we developed two complementary proxies measuring generation times from single pseudo-haploid time-stamped genomes. The first quantifies the number of generations required for a genome to accumulate an observed number of mutations post divergence from outgroup(s) (mutation clock; Supplementary Methods and Extended Data Fig. 8a). The second leverages recombination patterns to estimate the number of generations elapsed since the most recent common ancestor (MRCA) of the sampled specimens (recombination clock; Supplementary Methods and Extended Data Fig. 9a,b). We validate the performance of our methodology through coalescent simulations across various inbreeding levels and demographic trajectories (Extended Data Fig. 10), and apply it to all of our radiocarbon-dated horse genomes to estimate roughly 7.4 years as the average time between two consecutive generations in the past 15,000 years (Fig. 3b and Supplementary Information).

Our analyses also show that horse generation times did not remain constant, but accelerated around 1.8-fold (approximately 4.1 years) during the past approximately 200 years, as could be expected given

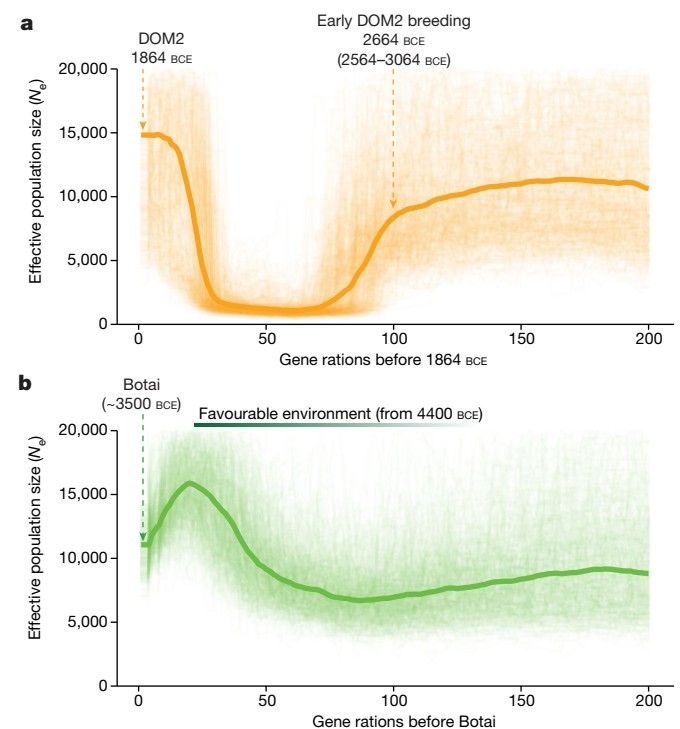

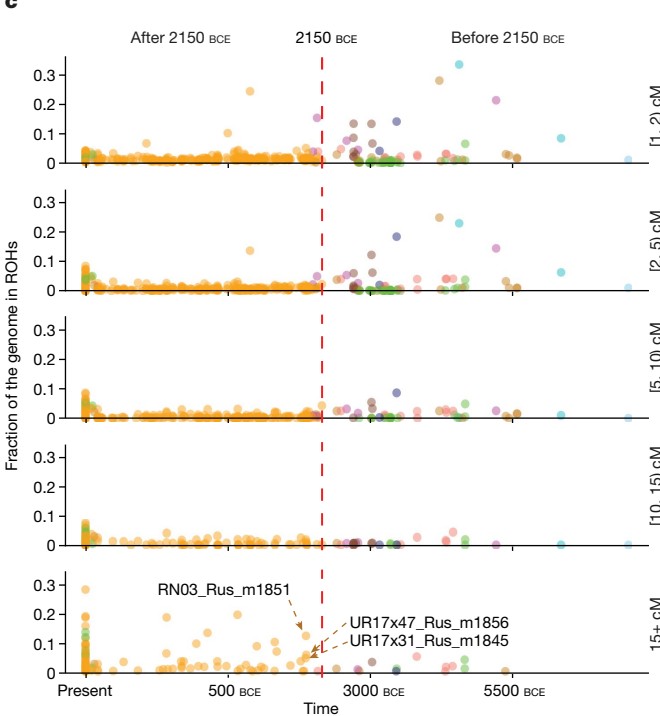

**Fig. 2 | Horse demographic trajectory and inbreeding profiles. a**, GONE[26] demographic reconstruction based on 24 early DOM2 horse genomes; the thicker line depicts the most likely effective population size up to 200 generations preceding about 1864 BCE, and the thinner lines are 500 bootstrap pseudo-replicates. Conversions to calendar years BCE assume either average generation times of 8 (7–12) years or our refined estimate for the time periods considered. **b**, Same as **a** but for a set of 28 Botai horse genomes. **c**, Total fraction of the genome encompassing ROHs of various sizes, in which each dot represents a horse genome. For example, the category [1, 2) cM indicates the fraction of a genome within ROHs that are longer than or equal to 1 cM, but shorter than 2 cM.

the development of modern breeding practices, optimized for animal production (Fig. 3a). Racing Quarter Horses and Thoroughbreds exemplify breeds with the least accelerated generation time, possibly due to the extended reproductive lifespan imposed on sport champions (Fig. 3a). No equivalent changes were detected backwards in time until about 2200–2100 BCE, which coincides with a roughly 2.1-fold acceleration of the generation time, relative to the average of about 7.4 years (to about 3.5 years; Fig. 3b). This acceleration did not affect any of the DOM2 relatives, including those with individuals affiliated with Yamnaya, Turganik and Steppe Maykop contexts (CPONT and TURG; Fig. 3 and Extended Data Fig. 7c), or the older horses living in the steppe (NEONCAS) or in the Carpathian and Transylvanian Basins (HUNG and ENEOROM; Extended Data Fig. 7c). This shows that new practices of DOM2 reproductive control, aimed at faster productivity, emerged by the late third millennium BCE, and were a prerequisite to early DOM2 breeding and adoption of widespread horse-based mobility.

## New evidence of horse husbandry at Botai

Earlier research established minimal connectivity between horse populations during the fourth millennium BCE[2]. As this encompasses the timeline of the Botai settlement (around 3500 BCE), where controversial evidence for horse domestication was found, the incentive for domestication at Botai, if any, could not be long-distance horseback riding. In the 36 horses from the Botai site analysed, we found no evidence for close-kin mating, but we did find shortened generation times, an acceleration comparable in magnitude to that accompanying DOM2 breeding (Fig. 3). This trend is specific to the Botai and to a group descending directly from the Botai (Borly4, around 3000 BCE; Fig. 3 and Extended Data Fig. 7d)[7], and remains unprecedented in scale throughout the Ice Age to the Eneolithic. Notably, the Botai horse population

experienced a 2.4-fold demographic expansion starting roughly 80 generations before settlement (Fig. 2b), that is, about 4140 (4460–4060) BCE, assuming average generation times of 8 (7–12) years. This largely concurs with paleoclimatic data suggesting more humid conditions, and pollen records indicating no forest encroachment on the steppes[38]. These favourable conditions for horses may have encouraged humans to settle and develop a subsistence economy almost entirely focused on horses[39], suggested to have been initially established through hunting[40]. However, our demographic reconstructions indicate that this once thriving resource progressively declined during the last 20 generations of Botai (that is, in 140–240 years; Fig. 2b). In response to declining food resources, Botai peoples may have exercised husbandry practices involving corralling and horse reproductive control through shortened generation times, in line with the prey domestication pathway[6,41].

## Discussion

This study tackles crucial debates regarding horse domestication, with major implications for both horse and human history. It shows that the horse genomic make-up remained entirely local in central Europe and in the Carpathian and Transylvanian Basins until the end of the third millennium BCE. This timeline post-dates the period of steppe contact in the Carpathian and Transylvanian Basins starting around 4500 BCE[10], as well as the migrations potentially spreading proto-Indo-European languages into Europe with the Yamnaya phenomenon about 3000 BCE. The pronounced spread of DOM2 horses immediately followed the foundation of this new bloodline, and marked a new era of widespread horse-based mobility from about 2200 BCE, ushering in a monumental increase in connectivity and trade. It mirrors the archaeological record, which witnesses a massive spread of horses in the Near East and Asia during the transition between the third and second

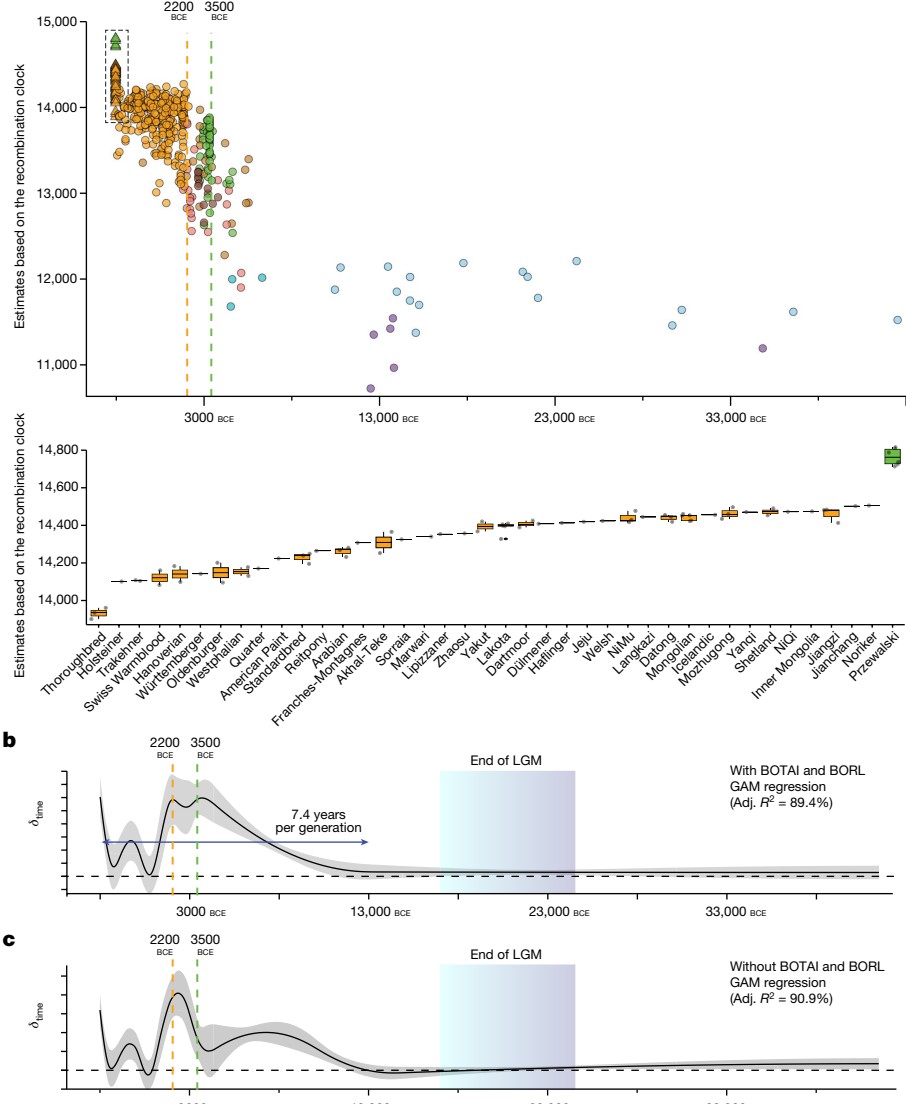

**Fig. 3 | Horse generation times. a**, Number of generations evolved since the MRCA of all samples, as estimated from the recombination clock (*y* axis) for each radiocarbon-dated horse specimen (*x* axis, age of the specimen; *n* = 483). Samples are colour-coded according to Fig. 1a. The bottom panel breaks down the number of generations evolved for modern breeds. Each box plot summarizes the estimates per breed (Supplementary Table 1), including its corresponding centre (median), box boundaries (interquartile range) and whiskers (1.5 times the interquartile range). **b**, Time periods associated with significant changes in horse generation times. The graph represents the slope ($\delta_{\text{time}}$) of a GAM regressing radiocarbon dates and number of generations evolved since the MRCA while controlling for sequencing depth and population structure. This slope is, thus, proportional to the generation time at a particular time period. The double-sided arrow reports the average generation time in the past 15,000 years (Supplementary Information). The error band represents the 95% confidence interval for the GAM regressions. **c**, Same as **b** but excluding BOTAI and BORL population groups. LGM, Last Glacial Maximum.

millennium BCE[2,42,43]. Intensified herding practices[12], growing aridity (the '4.2 ka BP aridification event'[44]) and/or increased exploitation of the steppe may have heightened the demand for expanding grazing areas, potentially facilitated by horse-mediated mobility. Domestic horses and spoke-wheeled chariots[3,42] may also have aided the conquest and defence of larger geographic areas in the face of uprising violence and social conflicts[35,36].

Our work does not reject the possibility of equestrianism developing in the Pontic steppe or the Carpathian Basin before 2200 BCE. However, in such a scenario, the associated breeding practices would not have involved close-kin mating or accelerated generation times. The phenomenon would also have remained confined in scale, both demographically and geographically, excluding long-distance fast mobility as the primary domestication incentive. Our research strengthens the case for recognizing Botai as one such location in the central Asian steppe where horse husbandry developed before large-scale horse-based

mobility. There, the domestication process did not aim at global production, but remained regional. It is aligned with the expectations of the prey pathway[41], in which a settled group of humans developed husbandry through corralling and reproductive control, in the form of shortened generation times, but not close-kin mating, to ensure access to an otherwise depleting meat resource[13].

Manipulating the animal life cycle by forcing earlier reproduction offers breeders enhanced productivity, especially for species with long gestational periods and/or small litter sizes. Our research demonstrates that this practice was integral to the array of breeding techniques developed to sustain the massive global demand for horses from the Early Bronze Age. The pressure for accelerated production relaxed quickly after around 1000 BCE, as a large enough horse breeding pool became available across extensive geographic areas. However, the development of modern breeds required the fast production of specific bloodlines from limited foundational stocks, which again shortened

the horse generation time over the past few centuries. Apparently, this process affected Asian breeds more than racehorses (Fig. 3a), especially Thoroughbreds, for which artificial insemination is forbidden. These findings align with stud book pedigrees recording increasingly faster generation times during the past three centuries, especially in coldblood horses[45].

Our methodological framework for measuring generation times expands the bioarchaeological toolkit to detect molecular evidence of reproductive control. Together with close-kin mating, it may prove instrumental in clarifying the timing and context(s) into which human groups first developed animal husbandry, not only in horses, especially as early domestication processes may not always leave obvious skeletal modifications and marked foundational bottlenecks. Beyond domestic animals, our approach could be applied to measure the long-term generation times of ancient hominin groups, including Neanderthals and Denisovans, and their potential shifts in the face of major lifestyle transitions, such as following the out-of-Africa dispersal, during the Ice Age[46] and during the Neolithic revolution[47,48]. For now, our analyses suggest that the last Ice Age may have affected horse generation times, although to a lesser extent than domestication (Fig. 3). Our work opens the way for a new line of research investigating the possible consequences of past and present environmental and epidemiological crises on the reproduction of both human groups and other species.

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

Pablo Librado[1,108 ✉], Gaetan Tressières[1], Lorelei Chauvey[1], Antoine Fages[1,109], Naveed Khan[1,110], Stéphanie Schiavinato[1], Laure Calvière-Tonasso[1], Mariya A. Kusliy[1,111], Charleen Gaunitz[1,112], Xuexue Liu[1], Stefanie Wagner[1], Clio Der Sarkissian[1], Andaine Seguin-Orlando[1], Aude Perdereau[3], Jean-Marc Aury[4], John Southon[5], Beth Shapiro[6], Olivier Bouchez[7], Cécile Donnadieu[7], Yvette Running Horse Collin[1,113], Kristian M. Gregersen[8], Mads Dengsø Jessen[9], Kirsten Christensen[10], Lone Claudi-Hansen[10], Mélanie Pruvost[11], Erich Pucher[12], Hrvoje Vulic[13], Mario Novak[14], Andrea Rimpf[15], Peter Turk[16], Simone Reiter[17], Gottfried Brem[17], Christoph Schwall[18,19], Éric Barrey[20], Céline Robert[20,21], Christophe Degueurce[21], Liora Kolska Horwitz[22], Lutz Klassen[23], Uffe Rasmussen[24], Jacob Kveiborg[25], Niels Nørkjær Johannsen[26], Daniel Makowiecki[27],

Przemysław Makarowicz[28], Marcin Szeliga[29], Vasyl Ilchyshyn[30], Vitalii Rud[31],
Jan Romaniszyn[28], Victoria E. Mullin[32], Marta Verdugo[32], Daniel G. Bradley[32],
João L. Cardoso[33,34], Maria J. Valente[35], Miguel Telles Antunes[36], Carly Ameen[37],
Richard Thomas[38], Arne Ludwig[39,40], Matilde Marzullo[41], Ornella Prato[41],
Giovanna Bagnasco Gianni[41], Umberto Tecchiati[41], José Granado[42], Angela Schlumbaum[42],
Sabine Deschler-Erb[42], Monika Schernig Mráz[42], Nicolas Boulbes[43], Armelle Gardeisen[44],
Christian Mayer[45], Hans-Jürgen Döhle[46], Magdolna Vicze[47], Pavel A. Kosintsev[48,49],
René Kyselý[50], Lubomír Peške[51], Terry O'Connor[52], Elina Ananyevskaya[53], Irina Shevnina[54],
Andrey Logvin[54], Alexey A. Kovalev[55], Tumur-Ochir Iderkhangai[56], Mikhail V. Sablin[57],
Petr K. Dashkovskiy[58], Alexander S. Graphodatsky[59], Ilia Merts[60,61], Viktor Merts[60],
Aleksei K. Kasparov[62], Vladimir V. Pitulko[62,63], Vedat Onar[64], Aliye Öztan[65],
Benjamin S. Arbuckle[66], Hugh McColl[67], Gabriel Renaud[1,114], Ruslan Khaskhanov[68],
Sergey Demidenko[69], Anna Kadieva[70], Biyaslan Atabiev[71], Marie Sundqvist[72],
Gabriella Lindgren[73,74], F. Javier López-Cachero[75], Silvia Albizuri[75],
Tajana Trbojević Vukičević[76], Anita Rapan Papeša[13], Marcel Burić[77], Petra Rajić Šikanjić[78],
Jaco Weinstock[79], David Asensio Vilaró[80], Ferran Codina[81], Cristina García Dalmau[82],
Jordi Morer de Llorens[83], Josep Pou[84], Gabriel de Prado[85], Joan Sanmartí[86,87], Nabil Kallala[88,89],
Joan Ramon Torres[90], Bouthéina Maraoui-Telmini[89], Maria-Carme Belarte Franco[86,91,92],
Silvia Valenzuela-Lamas[93,94], Antoine Zazzo[95], Sébastien Lepetz[95], Sylvie Duchesne[1],
Anatoly Alexeev[96], Jamsranjav Bayarsaikhan[97,98], Jean-Luc Houle[99], Noost Bayarkhuu[100],
Tsagaan Turbat[100], Éric Crubézy[1], Irina Shingiray[101], Marjan Mashkour[95,102],
Natalia Ya. Berezina[103], Dmitriy S. Korobov[69], Andrey Belinskiy[104], Alexey Kalmykov[104],
Jean-Paul Demoule[105], Sabine Reinhold[106], Svend Hansen[106], Barbara Wallner[17],
Natalia Roslyakova[107], Pavel F. Kuznetsov[107], Alexey A. Tishkin[61], Patrick Wincker[4],
Katherine Kanne[37,115], Alan Outram[37] & Ludovic Orlando[1✉]

[1]Centre d'Anthropobiologie et de Génomique de Toulouse, CNRS UMR 5288, Université Paul Sabatier, Faculté de Médecine Purpan, Toulouse, France. [2]INRAE Division Ecology and Biodiversity (ECODIV), Plant Genomic Resources Center (CNRGV), Castanet Tolosan Cedex, France. [3]Genoscope, Institut de Biologie François Jacob, CEA, CNRS, Université d'Évry, Université Paris-Saclay, Évry, France. [4]Génomique Métabolique, Genoscope, Institut François Jacob, CEA, CNRS, Université d'Évry, Université Paris-Saclay, Évry, France. [5]Department of Earth System Science, University of California, Irvine, CA, USA. [6]Department of Ecology and Evolutionary Biology, University of California Santa Cruz, Santa Cruz, CA, USA. [7]INRAE, GeT-PlaGe, Genotoul, Castanet-Tolosan, France. [8]The Royal Danish Academy, Institute of Conservation, Copenhagen, Denmark. [9]Department for Prehistory Middle Ages and Renaissance, National Museum of Denmark, Copenhagen K, Denmark. [10]Museum Vestsjælland, Holbæk, Denmark. [11]UMR 5199 De la Préhistoire à l'Actuel: Culture, Environnement et Anthropologie (PACEA), CNRS, Université de Bordeaux, Pessac Cédex, France. [12]Museum of Natural History, Vienna, Austria. [13]Vinkovci Municipal Museum, Vinkovci, Croatia. [14]Centre for Applied Bioanthropology, Institute for Anthropological Research, Zagreb, Croatia. [15]Ilok Town Museum, Ilok, Croatia. [16]Narodni muzej Slovenije, Ljubljana, Slovenia. [17]Institute of Animal Breeding and Genetics, Department of Biomedical Sciences, University of Veterinary Medicine Vienna, Vienna, Austria. [18]Leibniz-Zentrum für Archäologie (LEIZA), Mainz, Germany. [19]Department of Prehistory & Western Asian/Northeast African Archaeology, Austrian Archaeological Institute (OeAI), Austrian Academy of Sciences (OeAW), Vienna, Austria. [20]Université Paris-Saclay, AgroParisTech, INRAE GABI UMR1313, Jouy-en-Josas, France. [21]Ecole Nationale Vétérinaire d'Alfort, Maisons-Alfort, France. [22]National Natural History Collections, Edmond J. Safra Campus, Givat Ram, The Hebrew University, Jerusalem, Israel. [23]Museum Østjylland, Grenaa, Denmark. [24]Department of Archaeology, Moesgaard Museum, Højbjerg, Denmark. [25]Department of Archaeological Science and Conservation, Moesgaard Museum, Højbjerg, Denmark. [26]Department of Archaeology and Heritage Studies, Aarhus University, Højbjerg, Denmark. [27]Institute of Archaeology, Faculty of History, Nicolaus Copernicus University, Toruń, Poland. [28]Faculty of Archaeology, Adam Mickiewicz University, Poznań, Poland. [29]Institute of Archaeology, Maria Curie-Skłodowska University, Lublin, Poland. [30]Krem enetsko-Pochaivskii Derzhavnyi Istoriko-arkhitekturnyi Zapovidnik, Kremenets, Ukraine. [31]Institute of Archaeology, National Academy of Sciences of Ukraine, Kyiv, Ukraine. [32]Smurfit Institute of Genetics, Trinity College Dublin, Dublin, Ireland. [33]ICArEHB, Campus de Gambelas, University of Algarve, Faro, Portugal. [34]Universidade Aberta, Lisbon, Portugal. [35]Faculdade de Ciências Humanas e Sociais, Centro de Estudos de Arqueologia, Artes e Ciências do Património, Universidade do Algarve, Faro, Portugal. [36]Centre for Research on Science and Geological Engineering, Universidade Nova de Lisboa, Lisbon, Portugal. [37]Department of Archaeology and History, University of Exeter, Exeter, UK. [38]School of Archaeology and Ancient History, University of Leicester, Leicester, UK. [39]Department of Evolutionary Genetics, Leibniz-Institute for Zoo and Wildlife Research, Berlin, Germany. [40]Albrecht Daniel Thaer-Institute, Faculty of Life Sciences, Humboldt University Berlin, Berlin, Germany. [41]Dipartimento di Beni Culturali e Ambientali, Università degli Studi di Milano, Milan, Italy. [42]Department of Environmental Sciences, Integrative Prehistory and Archaeological Science, Basel University, Basel, Switzerland. [43]Institut de Paléontologie Humaine, Fondation Albert Ier, Paris/UMR 7194 HNHP, MNHN-CNRS-UPVD/EPCC Centre Européen de Recherche Préhistorique, Tautavel, France. [44]Archéologie des Sociétés Méditeranéennes, Archimède IA-ANR-11-LABX-0032-01, CNRS UMR 5140, Université Paul Valéry, Montpellier, France. [45]Department for Digitalization and Knowledge Transfer, Federal Monuments Authority Austria, Vienna, Austria. [46]Landesamt für Denkmalpflege und Archäologie Sachsen-Anhalt – Landesmuseum für Vorgeschichte, Halle (Saale), Germany. [47]National Institute of Archaeology, Hungarian National Museum, Budapest, Hungary. [48]Paleoecology Laboratory, Institute of Plant and Animal Ecology, Ural Branch of the Russian Academy of Sciences, Ekaterinburg, Russia. [49]Department of History of the Institute of Humanities, Ural Federal University, Ekaterinburg, Russia. [50]Department of Natural Sciences and Archaeometry, Institute of Archaeology of the Czech Academy of Sciences, Prague, Czechia. [51]Independent researcher, Prague, Czechia. [52]Department of Archaeology, University of York, York, UK. [53]Department of Archaeology, History Faculty, Vilnius University, Vilnius, Lithuania. [54]Laboratory for Archaeological Research, Akhmet Baitursynuly Kostanay Regional University, Kostanay, Kazakhstan. [55]Department of Archaeological Heritage Preservation, Institute of Archaeology of the Russian Academy of Sciences, Moscow, Russia. [56]Department of Innovation and Technology, Ulaanbaatar Science and Technology Park, National University of Mongolia, Ulaanbaatar, Mongolia. [57]Zoological Institute, Russian Academy of Sciences, St Petersburg, Russia. [58]Department of Russian Regional Studies, National and State-confessional Relations, Altai State University, Barnaul, Russia. [59]Department of the Diversity and Evolution of Genomes, Institute of Molecular and Cellular Biology, Novosibirsk, Russia. [60]Toraighyrov University, Joint Research Center for Archeological Studies, Pavlodar, Kazakhstan. [61]Department of Archaeology, Ethnography and Museology, Altai State University, Barnaul, Russia. [62]Institute of the History of Material Culture, Russian Academy of Sciences, St. Petersburg, Russia. [63]Peter the Great Museum of Anthropology and Ethnography (Kunstkamera), Russian Academy of Sciences, St Petersburg, Russia. [64]Osteoarchaeology Practice and Research Center and Department of Anatomy, Faculty of Veterinary Medicine, Istanbul University-Cerrahpaşa, Istanbul, Türkiye. [65]Archaeology Department, Ankara University, Ankara, Türkiye. [66]Department of Anthropology, Alumni Building, University of North Carolina at Chapel Hill, Chapel Hill, NC, USA. [67]Lundbeck Foundation GeoGenetics Centre, Globe Institute, University of Copenhagen, Copenhagen, Denmark. [68]Kh. Ibragimov Complex Institute of the Russian Academy of Sciences (CI RAS), Grozny, Russia. [69]Institute of Archaeology, Russian Academy of Sciences, Moscow, Russia. [70]Department of Archaeological Monuments, State Historical Museum, Moscow, Russian Federation. [71]Institute for Caucasus Archaeology, Nalchik, Russian Federation. [72]Östra Greda Research Group, Borgholm, Sweden. [73]Department of Animal Breeding and Genetics, Swedish University of Agricultural Sciences, Uppsala, Sweden. [74]Center for Animal Breeding and Genetics, Department of Biosystems, KU Leuven, Leuven, Belgium. [75]Institut d'Arqueologia de la Universitat de Barcelona (IAUB), Seminari d'Estudis i Recerques Prehistòriques (SERP-UB), Universitat de Barcelona (UB), Barcelona, Spain. [76]Department of Anatomy, Histology and Embryology, Faculty of Veterinary Medicine, University of Zagreb, Zagreb, Croatia. [77]Department of Archaeology, Faculty of Humanities and Social Sciences, University of Zagreb, Zagreb, Croatia. [78]Institute for Anthropological Research, Zagreb, Croatia. [79]Faculty of Arts and Humanities (Archaeology), University of Southampton, Southampton, UK. [80]Secció de Prehistòria i Arqueologia, IAUB Institut d'Arqueologia de la Universitat de Barcelona, Barcelona, Spain. [81]C/Major, 20, Norfeu, Arqueologia Art i Patrimoni S.C., La Tallada d'Empordà, Spain. [82]Mosaïques Archéologie, Espace d'activités de la Barthe, Cournonterral, France. [83]Mon IberRocs SCL, Vilanova i la Geltrú (Barcelona), Spain. [84]Ajuntament de Calafell, Calafell (Tarragona), Spain. [85]Museu d'Arqueologia de Catalunya (MAC-Ullastret), Ullastret, Spain. [86]IEC-Institut d'Estudis Catalans (Union Académique Internationale), Barcelona, Spain. [87]Departament d'Història i Arqueologia, Facultat de Geografia i Història, Universitat de Barcelona, Barcelona, Spain. [88]Ecole Tunisienne d'Histoire et d'Anthropologie, Tunis, Tunisia. [89]University of Tunis, Institut National du Patrimoine, Tunis, Tunisia. [90]Consell Insular d'Eivissa, Eivissa, Spain. [91]ICREA, Catalan Institution for Research and Advanced Studies, Barcelona, Spain. [92]ICAC (Catalan Institute of Classical Archaeology), Tarragona, Spain. [93]Archaeology of Social Dynamics (ASD), Institució Milà i Fontanals, Consejo Superior de Investigaciones Científicas (IMF-CSIC), Barcelona, Spain. [94]UNIARQ – Unidade de Arqueologia, Universidade de Lisboa, Alameda da Universidade, Lisboa, Portugal. [95]Centre National de Recherche Scientifique, Muséum national d'Histoire naturelle, Archéozoologie, Archéobotanique (AASPE), CP 56, Paris, France. [96]Institute for Humanities Research and Indigenous Studies of the North (IHRISN), Yakutsk, Russia. [97]Max Planck Institute of Geoanthropology, Jena, Germany. [98]Institute of Archaeology, Mongolian Academy of Science, Ulaanbaatar, Mongolia. [99]Department of Folk Studies and Anthropology, Western Kentucky University, Bowling Green, KY, USA. [100]Archaeological Research Center and Department of Anthropology and Archaeology, National University of Mongolia, Ulaanbaatar, Mongolia. [101]Faculty of History, University of Oxford, Oxford, UK. [102]Central Laboratory, Bioarchaeology Laboratory, Archaeozoology section, University of Tehran, Tehran, Iran. [103]Research Institute and Museum of Anthropology, Lomonosov Moscow State University, Moscow, Russia. [104]Nasledie Cultural Heritage Unit, Stavropol, Russia. [105]UMR du CNRS 8215 Trajectoires, Institut d'Art et Archéologie, Paris, France. [106]Eurasia Department of the German Archaeological Institute, Berlin, Germany. [107]Department of Russian History and Archaeology, Samara State University of Social Sciences and Education, Samara, Russia. [108]Present address: Institut de Biologia Evolutiva (CSIC – Universitat Pompeu Fabra), Barcelona, Spain. [109]Present address: Zoological institute, Department of Environmental Sciences, University of Basel, Basel, Switzerland. [110]Present address: Department of Biotechnology, Abdul Wali Khan University, Mardan, Pakistan. [111]Present address: Department of the Diversity and Evolution of Genomes, Institute of Molecular and Cellular Biology, Novosibirsk, Russia. [112]Present address: Lundbeck Foundation GeoGenetics Centre, Globe Institute, University of Copenhagen, Copenhagen, Denmark. [113]Present address: Taku Skan Skan Wasakliyapi: Global Institute for Traditional Sciences, Rapid City, SD, USA. [114]Present address: Department of Health Technology, Section for Bioinformatics, Technical University of Denmark (DTU), Copenhagen, Denmark. [115]Present address: School of Archaeology, University College Dublin, Dublin, Ireland. ✉e-mail: pablo.librado@ibe.upf-csic.es; ludovic.orlando@univ-tlse3.fr

## Methods

### Archaeological samples and radiocarbon dating

We have gathered an extensive collection of 475 ancient horse remains spread across 230 sites in 41 countries. Sampling of archaeological horse remains was undertaken in collaboration with co-authors responsible for the curation and description of underlying contexts, and with the approval of the relevant institutions responsible for the archaeological remains, as detailed in the Reporting Summary. A total of 105 of the 124 newly sequenced specimens originate from archaeological sites for which no ancient horse genomes were characterized previously. Their underlying archaeological contexts are described in the Supplementary Information. A total of 140 new radiocarbon dates were obtained in this study, at the Keck Carbon Cycle Accelerator Mass Spectrometer Laboratory, University of California, Irvine (Supplementary Table 1). Collagen was extracted and ultra-filtered following mechanical cleaning of about 200 mg of cortical bone. Radiocarbon dates were calibrated using OxCalOnline[49] and the IntCal20 calibration curve[50]. Samples were named with reference to their original internal label, followed by a three-letter country code and their associated age in calendar years BCE or CE, all separated by underscore signs and appending the age with the 'm' prefix if BCE (for example, KT46_Aus_m3240 refers to sample KT46, originating from the Kittsee site from Austria, which showed a midpoint radiocarbon date of 3240 BCE).

### Genome sequencing

Osseous samples were processed for DNA extraction, library construction and shallow sequencing in the ancient DNA facilities of the Centre for Anthropobiology and Genomics of Toulouse (Centre national de la recherche scientifique (CNRS) and University Paul Sabatier), France. The overall methodology followed the work from ref. 2, including: (1) powdering with the Mixel Mill MM200 (Retsch) Micro-dismembrator; (2) DNA extraction according to the procedure Y2 from Gamba et al.[51]; (3) USER (NEB) enzymatic treatment[30]; (4) DNA library construction from double-stranded DNA templates DNA libraries in which two internal indexes are added during adaptor ligation and one external index is added during polymerase chain reaction (PCR) amplification; and (5) PCR amplification, purification and quantification on the TapeStation 4200 (D1000 HS) instrument before pooling for Illumina DNA sequencing on MiniSeq, NovaSeq and/or HiSeq4000 instruments (paired-end mode). Sequencing pools were prepared to represent each of the three individual indexes only once.

FASTQ sequencing reads demultiplexing, trimming and collapsing was carried out using AdapterRemoval2 (v.2.3.0)[52] disregarding reads shorter than 25 bp. The resulting collapsed and uncollapsed read pairs were processed through the Paleomix bam_pipeline (v.1.2.13.2)[53] for Bowtie2 (ref. 54) alignment against the nuclear and mitochondrial horse reference genomes[55,56], appended with the 751 Y-chromosome contigs from ref. 45, using the parameters recommended in ref. 57, removing PCR duplicates and requiring minimal mapping quality scores of 25. The presence of DNA fragmentation and nucleotide misincorporation patterns indicative of post-mortem DNA damage was assessed on the basis of 100,000 random mapped reads using mapDamage2 (v.2.0.8)[58]. Overall, we obtained sequence data from 390 DNA libraries for a total of 124 ancient horse specimens, resulting in genome characterization at an average depth of coverage of 0.288-to-10.925-fold (median 1.40-fold; Supplementary Table 1), as estimated using Paleomix coverage (--ignore-readgroups). The sequence data from 352 ancient and 81 modern genomes were processed following the same procedures to provide a comparative genome panel that included 4 donkeys[59], 2 *Equus ovodovi*[60] and 2 Late Pleistocene North American horses[61] that were used as outgroups, plus 550 horses representing all lineages previously characterized at the genome level (Supplementary Table 1).

### Genome rescaling and trimming, error rates and single nucleotide polymorphism variation

Sequencing errors and nucleotide misincorporations resulting from post-mortem DNA damage were reduced by subjecting alignments to a five-step procedure: (1) PMDtools (v.0.60)[62] identification and separation of those reads affected (--threshold 1; DAM) or not (--upperthreshold 1; NODAM) by post-mortem DNA damage, (2) 5 bp end-trimming of NODAM-aligned reads, (3) rescaling of DAM read alignments using mapDamage2 with default parameters (v.2.0.8)[58], (4) 10 bp trimming of rescaled read alignments and (5) merging of processed NODAM and DAM categories to obtain final Binary Alignment Map (BAM) sequence alignments. Error rates were estimated following Librado et al.[2] as the excess of private mutations, relative to a high-quality modern genome considered to be error-free (P5782_Ice_Modern; Supplementary Table 1). Single nucleotide polymorphisms (SNPs) were identified following the procedures from ref. 2, entailing data pseudo-haploidization with ANGSD (v.0.917)[63] for those sites covered by two reads or more (base quality scores greater than or equal to 30), and disregarding sites uncovered in 30% or more of the samples. A further filter included the random selection of one transversion SNP only, in cases where two successive transversions occurred in adjacent genomic positions. Overall, our final dataset retained 9,099,487 high-quality nucleotide transversions spread across the 31 horse autosomes. Alleles were polarized considering the allele common to the three outgroup lineages as ancestral. A second dataset of 7,092,366 variants was generated to mitigate for possible bias introduced by uneven sequencing depths by repeating the procedure described above, but following the downsampling of BAM alignment files to the median value of the average depth-of-coverage values found across all specimens (that is, 2.02-fold). Subsequent analyses were replicated on both variant datasets.

### Population graph modelling and population structure

Population graph modelling was carried out using the Markov chain Monte Carlo (MCMC) framework implemented in AdmixtureBayes[64], and in Admixtools2 (ref. 4), considering a pre-selection of 14 and 10 genetically homogeneous population groups, respectively, all represented by a minimum of two specimens. This was key for Admixtools2 analyses[4], to avoid biasing f3-statistics[4] in the presence of population groups comprising a single pseudo-haploid genome. AdmixtureBayes analyses involved three independent runs, each containing 163 MCMC chains recording 200 million iterations. The final space of population graphs was obtained using a burn-in of 90% and thinning one every 40 iterations. The genomic make-up of CWC horses was further investigated through the qpAdm rotating scheme[65] (Supplementary Table 2), and using a threshold of 0.01 for statistical significance. The geographic origins of CWC horses were also predicted using the Locator methodological framework based on deep neural networks[21]. To achieve this, we considered genomic window sizes of 10 Mb and the panel of 148 ancient horses predating the radiocarbon date of CWC horses. Genetic ancestries' decomposition and multidimensional scaling were carried out using the Struct-f4 package[24], grouping together 272 ancient and modern DOM2 horses to decrease computational costs. The first analytical step (assuming no admixture) consisted of 100 million MCMC iterations, whereas the second one (assuming admixture) involved 500 million iterations, until strict convergence. Default parameters were used otherwise, and the analyses were repeated assuming $K = 8$ to $K = 10$ admixture edges.

### Inbreeding

Per-genome inbreeding levels were estimated applying the methodology from ref. 59 to individual BAM alignment files. This methodology does not require prior knowledge of population allele frequencies; it involves instead the random sampling of two reads

per nucleotide transversion position and considering the density of sites within 1-cM-long genomic windows where the same allele was sampled twice (pseudo-homozygosity), versus two different alleles (pseudo-heterozygosity). Physical distances were converted into genetic distances using the recombination map from ref. 66, interpolating recombination rates linearly between two successive positions on the map. Windows showing pseudo-heterozygosity rates lower than 0.005 were considered to represent ROHs, with their cumulative span providing an inbreeding proxy. Close-kin mating was assessed through the total genome span encompassing long ROHs (that is, greater than or equal to 15 Mb).

## Demographic trajectories

A total of 28 genomes from unrelated Botai horses were pseudo-haploidized for transversion sites, all with a maximum missingness of 10%. The demographic dynamics was reconstructed using GONE[26] and patterns of linkage disequilibrium along all autosomes, excepting chromosomes 7, 11, 12 and 20. The parameter PHASE was turned to 0 to account for pseudo-haploid data; default parameters were applied otherwise. Confidence intervals for effective size variation were estimated from 500 bootstrap pseudo-replicates. The same procedure was repeated considering a selection of 24 ancient horse genomes dating back to an average of about 1850 BCE, which represents the earliest high-quality set of DOM2 genomes characterized.

## Generation times

Generation times and their potential variation were measured from the temporal accumulation of mutations present in a given genome relative to an ancestral sequence (reconstructed based on three outgroup species; that is, mutation clock) and from the linkage disequilibrium between pairs of derived mutations (that is, recombination clock). The proportion of derived mutations present in a given genome provided a direct proxy for the distance separating the sample considered from the ancestral sequence. This proportion was converted into an estimate of number of generations, assuming the mutation rate from ref. 29, rescaled for transversions, which provided our mutation clock estimate of generations elapsed from the ancestral sequence.

Our 'recombination clock' estimate is based on the average probability to find, in a given genome, a pair of SNPs separated by milliMorgans, and both carrying a derived allele. This probability was normalized by the proportion of derived mutations detected in the genome considered to mitigate potential bias resulting from depth-of-coverage and/or error rate variations across individuals, providing a direct measurement of the number of generations from the MRCA to all Eurasian horses present in our dataset. The 'mutation clock'-based estimate was derived from all 31 autosomes, whereas chromosomes 7, 11, 12 and 20 were masked to obtain the 'recombination clock' estimate, owing to limitations in the recombination map now available for horses in relation to unaccounted structural variation, local misassemblies and the presence of neocentromeres. The 'recombination clock' estimate depends on three unknown parameters that were optimized through least square optimization ($T$, the total genealogical length in the whole sample set averaged across loci; $t_i$, the genealogical length from the MRCA to horse specimen $i$ averaged across its loci; and a constant $p_i$ capturing sample-specific variation in demography and haplotype sizes).

Our methodology was validated using the serial coalescent simulation framework provided by fastsimcoal v.2.702 (ref. 67) and considering 10 demographic scenarios, consisting of constant population sizes, population contractions and population expansion of various magnitudes and times, followed or not by population recovery (Extended Data Fig. 10). Individual genomes were simulated as 31 autosomes of 75 Mb each, using $10^{-8}$ recombination events and $2.3 \times 10^{-8}$ mutation events per base pair and generation, respectively. A total of 20 simulated individuals were sampled along the genealogy every 100 generations, starting 900 generations ago, to cover the entire temporal range of horse domestication. Simulated as haploid, the 20 individuals sampled in each time bin, except the most recent, were then randomly paired to simulate diploid data under random mating, and were further subjected to pseudo-haploidization to mimic the data processing carried out on real data. The 20 individuals sampled for the most recent time period were paired with themselves before pseudo-haploidization to account for the increased inbreeding levels found in modern horse populations[68].

The real genome dataset was filtered to exclude the IBE, LPSFR, ELEN and Vert311 population groups, which contain significant ancestry affinities with Late Pleistocene specimens from North America (LPNAMR). This prevented biasing the generation time estimates as a result of DNA introgression from divergent population groups, related to lineages used to polarize alleles as ancestral or derived. Ancient specimens not associated with direct radiocarbon dating were also disregarded, except at Botai, where the archaeological context is similar across all samples. This left 483 specimens delivering both 'mutation clock' and 'recombination clock' estimates for the number of generations elapsed from the ancestral sequence and since the time to the MRCA of Eurasian horses, respectively. Temporal shifts in generation times were identified on the basis of the downsampled dataset (Fig. 3), and using a generalized additive model (GAM), as implemented in the R mgvc package. Radiocarbon dates, the first five coordinates of the Struct-f4 multidimensional scaling analysis to capture the underlying population structure and a parameter, $p_i$, controlling for the depth of coverage of each individual genome were the model covariates. Standard errors for the dependent variable were calculated by jack-knifing, leaving one chromosome out at a time, and the inverse of the resulting variances were used as regression weights. Regression models in which radiocarbon dates were linearly related to the number of generations received significantly lower support than those allowing relaxing linearity through cubic spline transformation of radiocarbon dates (adjusted $R^2$ (adj. $R^2$) = 0.803 for the linear versus 0.894 for the GAM regression; analysis of variance $P < 2.2 \times 10^{-16}$). Finally, we used the derivative function of the R gratia package and time bins of 1,000 years to measure temporal changes in generation times.

## Reporting summary

Further information on research design is available in the Nature Portfolio Reporting Summary linked to this article.

## Data availability

All collapsed and paired-end sequence data for samples sequenced in this study are available in compressed FASTQ format through the European Nucleotide Archive under accession number PRJEB71445, together with rescaled and trimmed BAM sequence alignments against the nuclear horse reference genomes. Previously published ancient data used in this study are available under accession numbers PRJEB7537, PRJEB10098, PRJEB10854, PRJEB22390, PRJEB31613 and PRJEB44430, and detailed in Supplementary Table 1. The genomes of 78 modern horses, publicly available, were also accessed as indicated in their corresponding original publications, and in Supplementary Table 1. The maps presented in Fig. 1 were generated using QGIS 3.36 software (available at https://www.qgis.org/en/site/) and using free raster images obtained from Natural Earth (https://www.naturalearth-data.com/). The maps in Extended Data Fig. 3c,d were automatically generated through the R scripts embedded in the Locator software package (https://github.com/kr-colab/locator).

## Code availability

The software to calculate generation time changes based on the recombination clock is available without restriction at Bitbucket (https://bitbucket.org/plibradosanz/generationtime/src/master/) and Zenodo

(https://doi.org/10.5281/zenodo.10842666 or https://zenodo.org/records/10842666)[69].

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

**Acknowledgements** We thank A. Fromentier and C. Gamba for preliminary ancient DNA work; all members of the Archaeology, Genomics, Evolution and Societies group at the Centre for Anthropobiology and Genomics of Toulouse for support with laboratory organization and discussions; J. Karlsson and the Archaeological Collections at the National Historical Museums in Stockholm, Sweden, for advice and access to their collections; E. Willerslev for facilitating the sampling of ancient horse remains; A. Abedi, H. Davoudi, A. Mohaased, J. Nokandeh and H. Omrani for facilitating access to archaeological material from Iran; and A. Choyke and P. Csippán at the Aquincum Museum, and J. Dani of the Deri Museum for access to, and assistance with, the collections in Hungary. We thank A. Marangoni for initiating collaborative work with some of our Italian partner institutions; V. Zaibert, who dedicated his life excavating at Botai, for accessing material from the site. This work was supported by France Génomique National infrastructure, funded as part of the 'Investissement d'avenir' programme managed by Agence Nationale de la Recherche (ANR-10-INBS-09); RYC2021-031607 funded by MCIN/AEI/10.13039/501100011033 and by the European Union NextGenerationEU/Recovery, Transformation and Resilience Plan (PRTR); the consolidated research group SGR2021-00337-Seminari d'Estudis i Recerques Prehistòriques (SERP-UB) and SGR2021-00501- Archaeology of Social Dynamics (ASD, IMF-CSIC); the Ministerio de Economía y Competitividad of Spain (project HAR2017-87695-P); the Swiss National Science Foundation (project 178834); the Departament de Cultura of the 'Generalitat de Catalunya' (project ARQ001SOL-178-2022); the 'Ginnerup and the End of Northern Europe's First Farming Culture' project, funded by the Aage og Johanne Louis-Hansen Foundation and the Augustinus Foundation; the 'Cultural and economic centres of the Late Bronze Age of the southern part of the Middle Irtysh region' (AP23488815); the Innovation Fund of the Austrian Academy of Sciences (ÖAW) (grant agreement IF_2015_17); the Russian Science Foundation (project 22-18-00470); the CNRS International Research Project AnimalFarm; the University Paul Sabatier IDEX Chaire d'Excellence (OURASI); and the France Génomique Appel à Grand Projet (ANR-10-INBS-09-08, BUCEPHALE and MARENGO projects). C.A. and A. Outram were funded by the Arts and Humanities Research Council, UK (AH/S000380/1). Y.R.H.C. was supported by the European Union's Horizon 2020 research and innovation programme under the Marie Skłodowska-Curie (grant agreement 890702-MethylRIDE). K.K. was supported by a National Science Foundation Doctoral Dissertation Improvement Grant (no. 0833106) and a Wenner-Gren Foundation Dissertation Fieldwork Grant. A.K.K. is supported by government research (project FMZF-2022-0013). R. Kyselý is supported by the Czech Academy of Sciences (RVO:67985912). P.F.K. is supported by the Russian Science Foundation (project RSF 22-18-00194). P.M. is supported by the National Science Centre (NSC), Poland, under the grant number 2023/49/B/HS3/00825. I.M. and V.M. are supported by the Ministry of Science and Education of the Republic of Kazakhstan (project BR18574223). V.V.P. is supported by the Russian Science Foundation (projects 21-18-00457P RNF and 24-68-00031 RNF) and government research (project FMZF-2022-0012). M.V.S. is supported by grant agreement 075-15-2021-1069. M. Szeliga is supported by the NSC, Poland, under grant number 2015/19/B/HS3/01720. This project has received funding from the European Research Council (ERC) under the European Union's Horizon 2020 research and innovation programme (grant agreements 834616-ARCHCAUCASUS, 101117101-anthropYXX, 681605-PEGASUS and 101071707-Horsepower).

**Author contributions** P.L. and L.O. conceptualized the project. K.M.G., M.D.J., K.C., L.C.H., M.P., E.P., H.V., M.N., A.R., P.T., S. Reiter, G.B., C.S., E.B., C.R., C. Degueurce, L.K.H., L.K., U.R., J.K., N.N.J., D.M., P.M., M. Szeliga, V.I., V.R., J.R., V.E.M., M. Verdugo, D.G.B., J.L.C., M.J.V., M.T.A., C.A., R.T., A. Ludwig, M. Marzullo, O.P., G.B.G., U.T., J.G., A.S., S.D.E., M.S.M., N. Boulbes, A.G., C.M., H.J.D., M. Vicze, P.A.K., R. Kyselý, L.P., T.O.C., E.A., I. Shevnina, A. Logvin, A.A.K., T.O.I., M.V.S., P.K.D., A.S.G., I.M., V.M., A.K.K., V.V.P., V.O., A. Öztan, B.S.A., H.M., G.R., R. Khaskhanov, S. Demidenko, A. Kadieva, B.A., M. Sundqvist, G.L., F.J.L.C., S.A., T.T.V., A.R.P., M.B., P.R.S., J.W., D.A.V., F.C., C.G.D., J.M.D.L., J.P., G.D.P., J. Sanmartí, N. Kallala, J.R.T., B.M.T., M.C.B.F., S.V.L., A.Z., S.L., S. Duchesne, A.A., J.B., J.L.H., N. Bayarkhuu, T.T., E.C., I. Shingiray, M. Mashkour, N.Y.B., D.S.K., A.B., A. Kalmykov, J.P.D., S. Reinhold, S.H., B.W., N.R., P.F.K., A.A.T., K.K., A. Outram and L.O. were responsible for sample collection. G.T., L.C., A.F., N. Khan, S.S., L.C.T., M.A.K., C.G., X.L., S.W., C.D.S., A.S.O., A.P., J.M.A., J. Southon, B.S., O.B., C. Donnadieu, Y.R.H.C., P.W. and L.O. were responsible for data generation. P.L. and L.O. carried out data analyses. P.L. developed the method. Y.R.H.C., A.S.O., L.K., P.M., M. Szeliga, R. Kyselý, M.V.S., I.M., V.M., A.K.K., V.V.P., F.J.L.C., S.A., S.H., B.W., P.F.K., A.A.T., K.K., A. Outram and L.O. acquired funding. L.O. co-ordinated the project. P.L. and L.O. wrote the original draft. L.O. reviewed and edited the paper, with input from all co-authors.

**Competing interests** The authors declare no competing interests.

**Additional information**
**Correspondence and requests for materials** should be addressed to Pablo Librado or Ludovic Orlando.

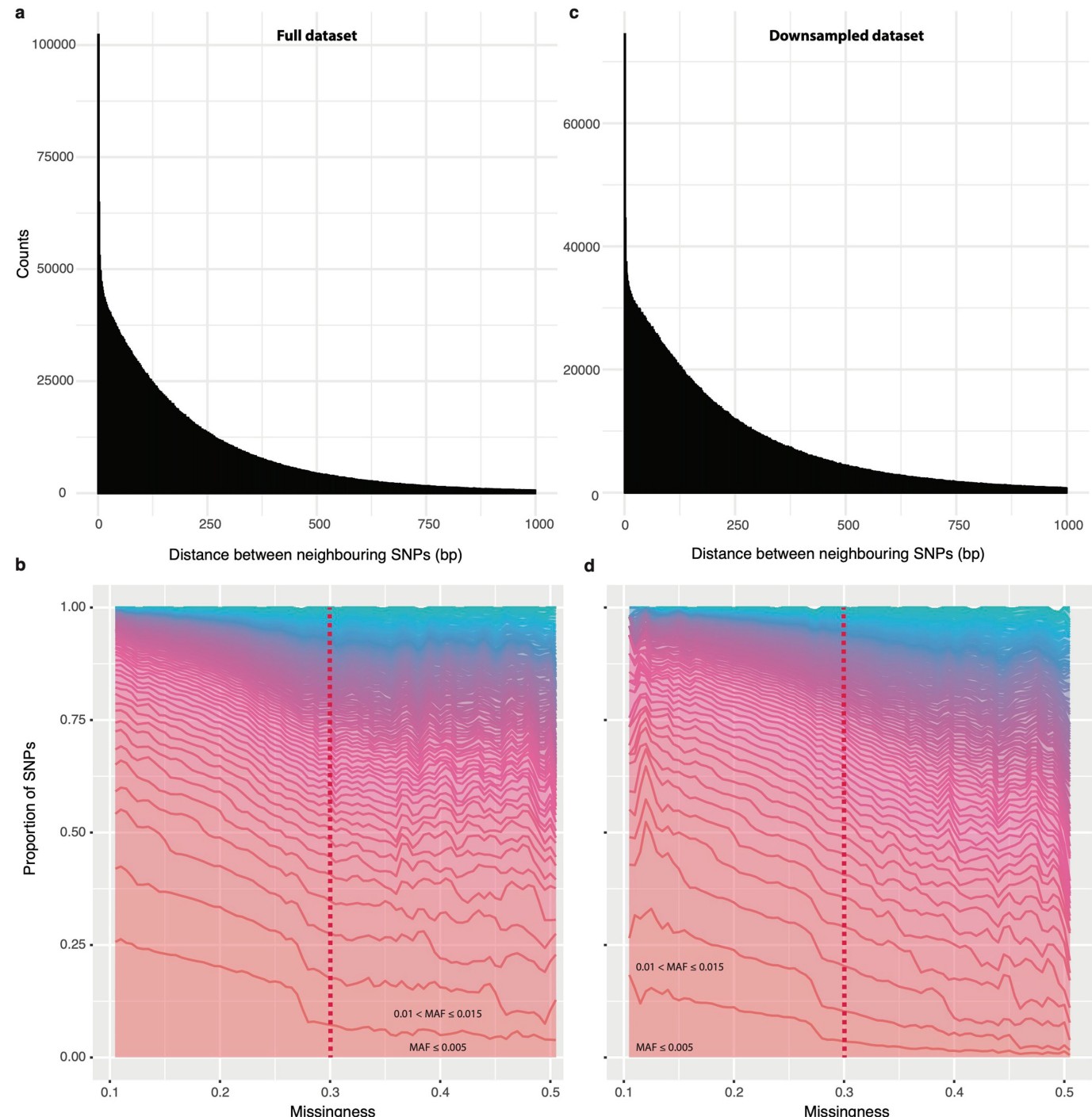

**Extended Data Fig. 1 | QC filtering. a**) Histogram showing the distance between adjacent nucleotide transversions, if separated by less than 1Kbp. This revealed an excess of mutations at contiguous genomic positions (*ie.* 1 bp away). Although these could correspond to true single nucleotide polymorphism (SNPs) or multiple nucleotide variants (MNVs), they could also be enriched for spurious variants resulting from mis-mapping around small DNA insertions and deletions. **b**) Proportion of mutations within pre-defined MAF bins (Minor Allele Frequency), as a function of missingness across the specimens. Pre-defined MAF bins range from low- (pink) to high-frequency variants (green). The dashed line delimits the positions included (left) or excluded (right) from the analyses. The identifiability of low-frequency variants decreases with greater missingness, as expected. **c**) Same as panel a), for the ~7.1 M nucleotide transversions of the downsampled data set. **d**) Same as panel b), for the ~7.1 M nucleotide transversions of the downsampled data set.

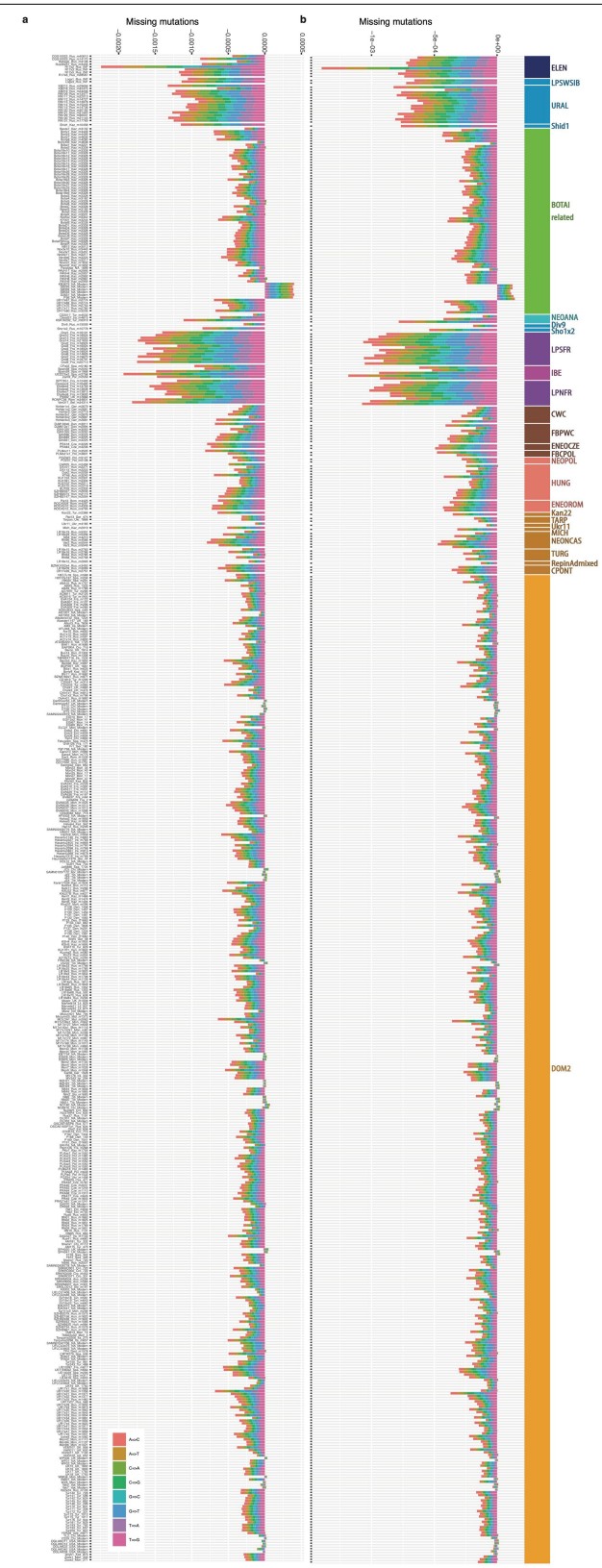

**Extended Data Fig. 2 | Relative error rates.** Missing mutations per site in a test genome (y-axis), relative to a modern Icelandic horse (P5782_Ice_Modern) used as high-quality reference. **a**) for the full data set and SNP_pval 0. **b**) for the downsampled data set and SNP_val 0.

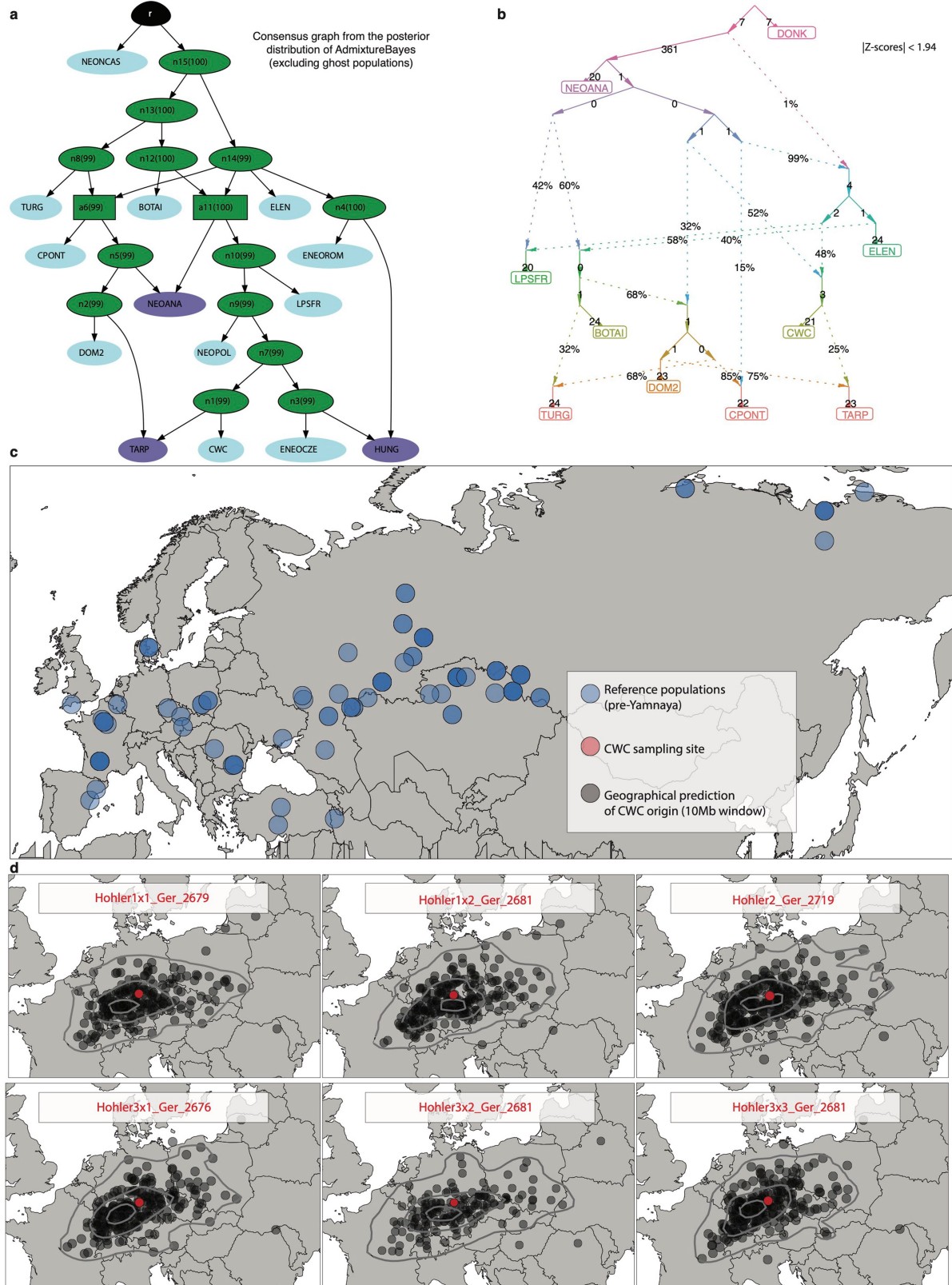

**Extended Data Fig. 3 | On the origins of CWC horses. a**) Consensus admixture graph generated from the posterior distribution of AdmixtureBayes[64], when applied to the same horse populations considered in Extended Data Fig. 4. The values between brackets summarize the proportion of graphs sampled from the posterior distribution that support a split or admixture node. Admixture from unsampled (ghosts) populations is not represented, in contrast to Extended Data Fig. 4. **b**) Best Admixtools2[4] population model assuming 8 migration edges. The drift and admixture estimates are based on our extended dataset. **c**) Reference panel used for modeling pre-CWC clines of genetic diversity. **d**) Geospatial projection of the six CWC horse genomes analyzed in this study, in 10Mb-long windows.

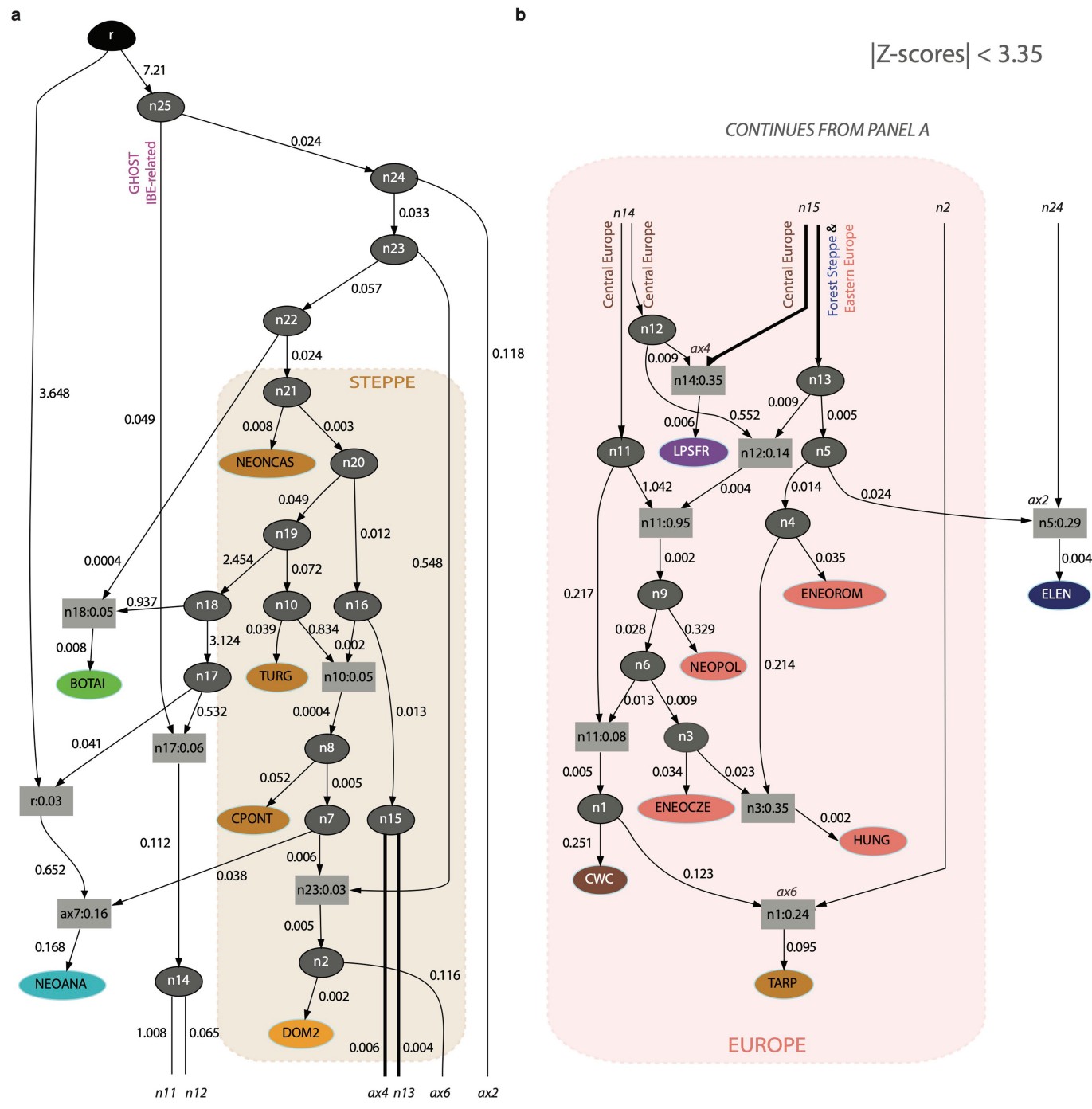

**Extended Data Fig. 4 | Most supported population graph.** This graph summarizes the evolutionary history of pre- and post-domestication horse lineages, with CWC horses not receiving any direct genetic contribution from the steppe. The model is split into 2 panels for clarity. The numbers reported within boxes reflect the admixture contributions from the nodes specified, while those adjacent to arrows indicate the amount of genetic drift leading to individual nodes. Population groups are detailed in Table S1 and colors are according to Fig. 1a.

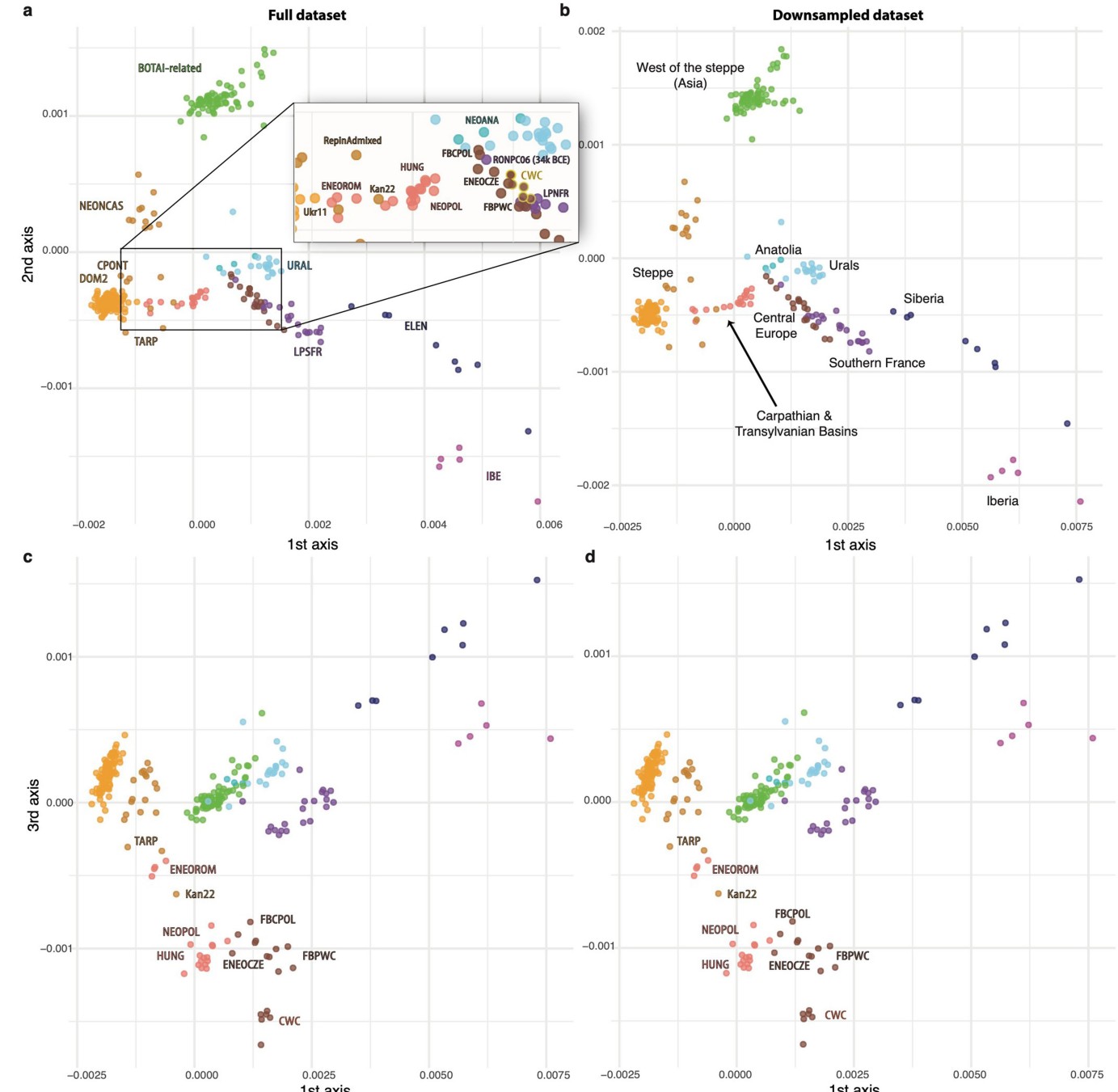

**Extended Data Fig. 5 | Visual embedding of Struct-f4 affinities. a**) The two first dimensions of a Metric MultiDimensional Scaling (MDS) analysis, summarizing the genomic affinities between horses, based on Struct-f4. To improve visualization, this excludes the five outgroup specimens. Samples are color-coded following Fig. 1a, and population groups are labelled accordingly. Horses projecting intermediate to large population groups reflect ancient clines of ancestry, stretching from the East (closer to Botai) to the West (closer to Europe). CPONT individuals, from the Central Steppe, are the closest to DOM2 horses. **b**) Same as a) for the downsampled dataset. **c**) First and third dimension of the same MDS analysis, which reveals CWC horses as the most distant European horses to DOM2 horses. **d**) Same for the downsampled dataset.

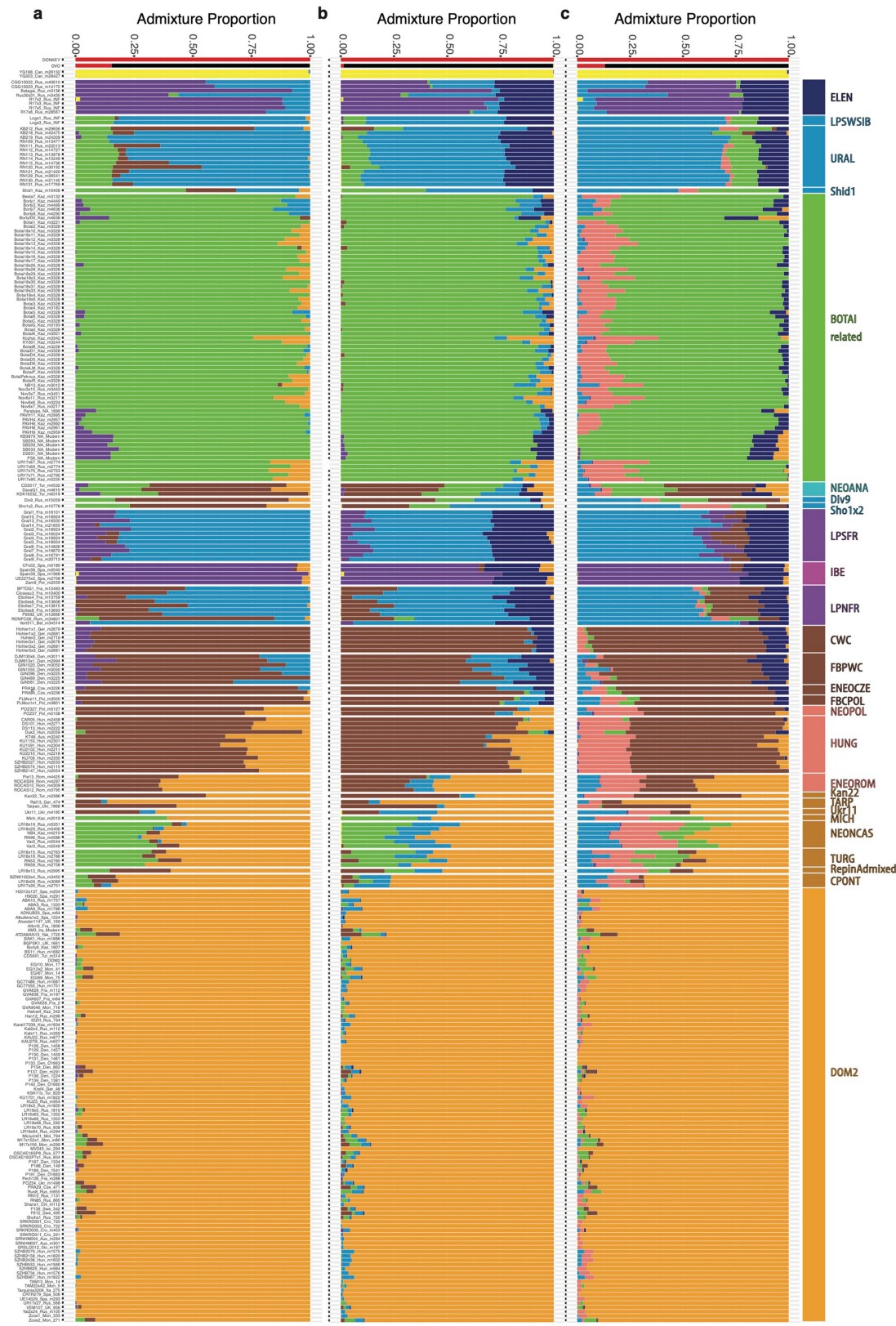

**Extended Data Fig. 6 | Struct-f4 ancestry profiles.** Ancestry proportions for the 558 individuals considered in this study, assuming from K = 8 (left) to K = 10 (right) components. A total of 272 horses previously identified as DOM2 were merged into a single population (DOM2), including all modern breeds, to reduce computational costs. CWC horses show the typical ancestry profile of pre-domestication Europe.

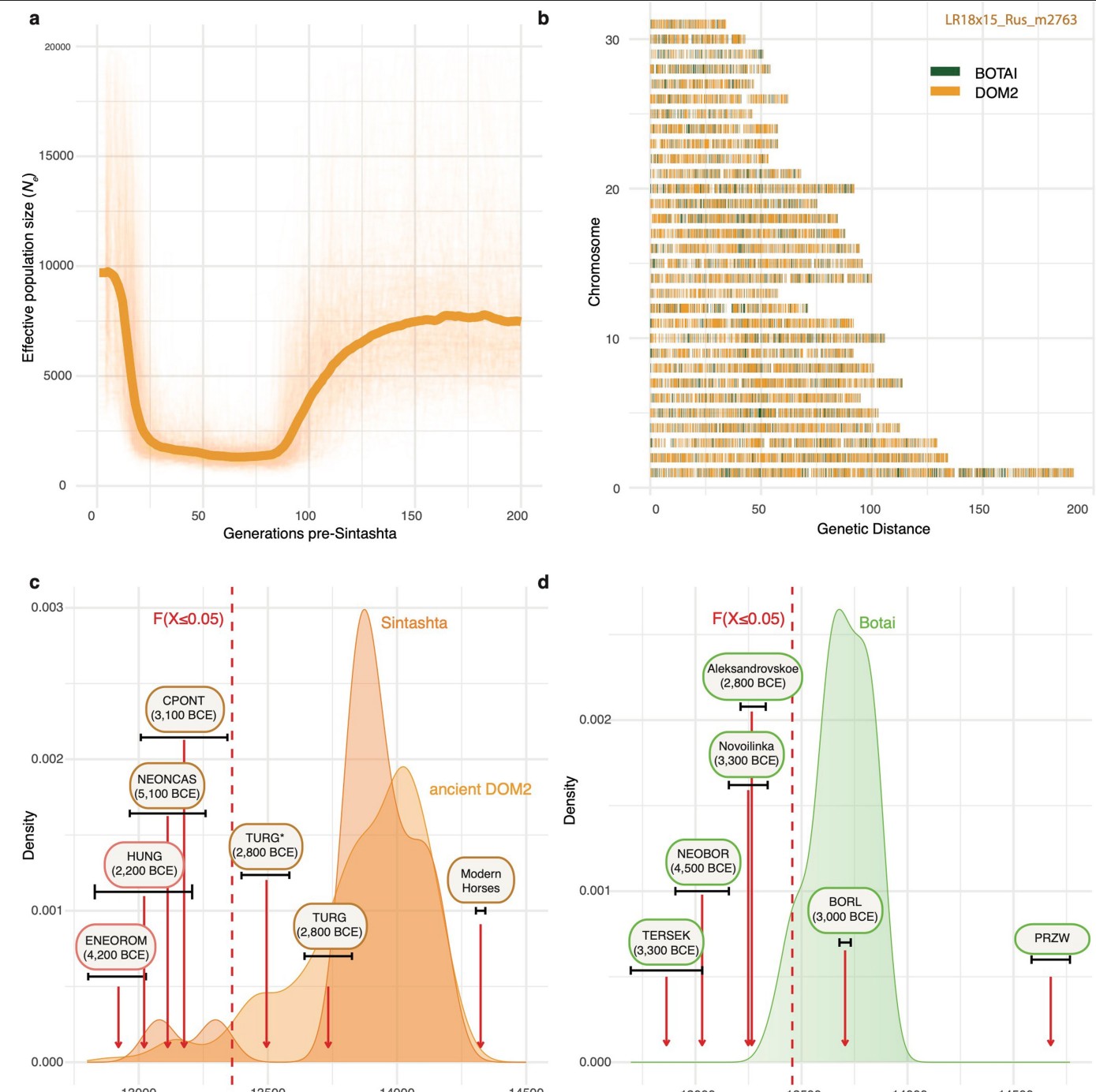

**Extended Data Fig. 7 | GONE demographic reconstruction.** Effective population size ($N_e$) estimated from the patterns of linkage disequilibrium (LD) present in a nearly contemporaneous population of 14 horses affiliated to the Sintashta culture, up to 200 generations before their existence. **b**) Example of local ancestry for a TURG horse genome (LR18x15_Rus_m2763), modeled with Admixfrog as a mixture of Botai and early DOM2 horses. **c**) Raw generation time estimates for ancient horses from the steppe, the Carpathian and Transylvanian Basins, without correcting for population structure and uneven sequencing depths (Supplementary Information). TURG* represents the group of TURG horses, after masking their genomes for tracts introgressed from Botai horses. **d**) Same for Botai horses, which involved more generations than past and contemporaneous horses from the region, with the exception of BORL and Przewalski's horses (PRZW), previously inferred to descend from Botai and saved from extinction through captive management. The dates reported correspond to rounded means of the different samples present in each group.

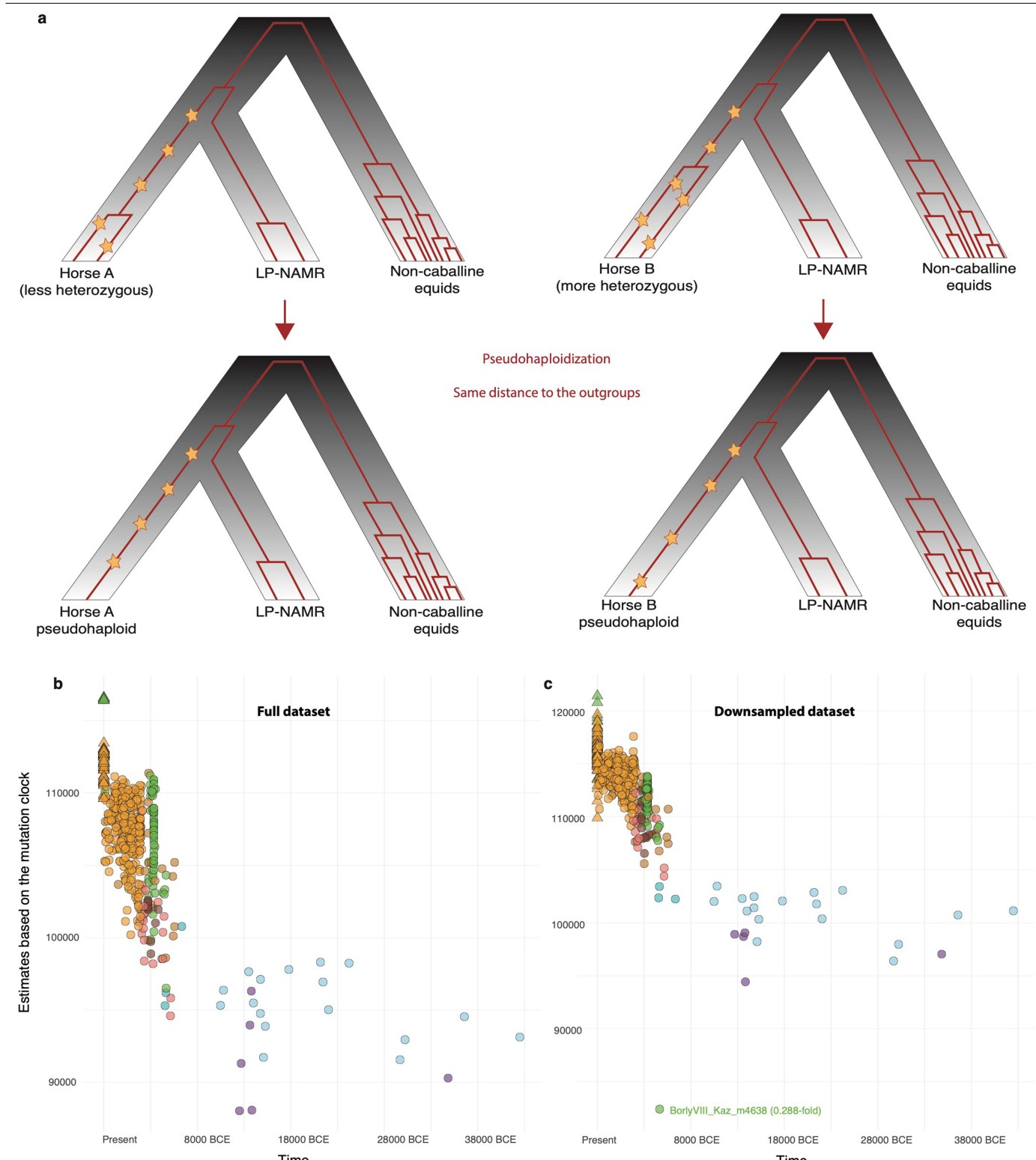

**Extended Data Fig. 8 | Mutation clock estimates. a)** Relationship of the ingroup Eurasian horses to the outgroups considered in this study, including non-caballine equids (*E. ovodovi* and the donkey) and ancient horses from North America (LP_NAMR). Leveraging this topology, we counted the number of mutations (represented as stars) that occurred in the branch leading to every single Eurasian horse. Following pseudohaploidization, positions that are truly heterozygous in Eurasian horses become ancestral or derived, and both outcomes are expected at equal probabilities. This approach is, thus, insensitive to the underlying heterozygosity of the sample, and, hence, to their demographic history. **b)** Estimates of the number of generations evolved from the outgroups, based on the full data set. **c)** Estimates based on the downsampled dataset.

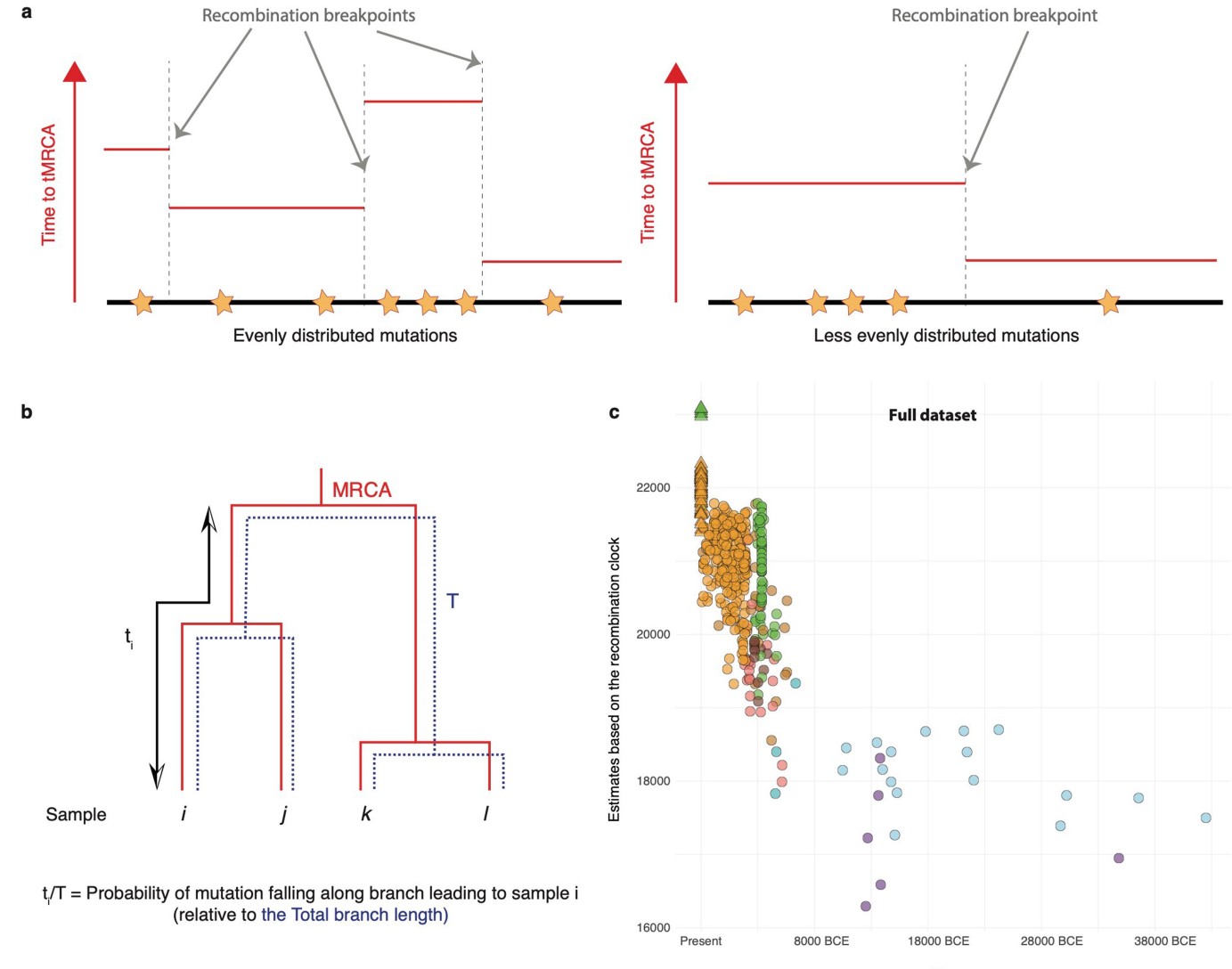

**Extended Data Fig. 9 | Recombination clock estimates. a**) Schematic representation that illustrates the expectation that the variance along the genome is greater in an older specimen (left) as the result of more generations of evolution and, hence, more recombination events than in younger specimens with regards to the time to the most common recent ancestor (MRCA) of the whole sample set. It is thus expected that the distribution of mutations (stars) is less even in the younger specimen (right), which underwent fewer recombination events, and thus carry longer haplotype blocks, in which mutations are equally likely to have occurred or not. **b**) Schematic visualization of the $t_i$ (time to the MRCA) and $T$ (total length of the genealogy) parameters constituting the recombination clock model, for an illustrative sample of four genomes. **c**) Number of generations evolved from the MRCA, as estimated by applying the recombination clock model to the full data set.

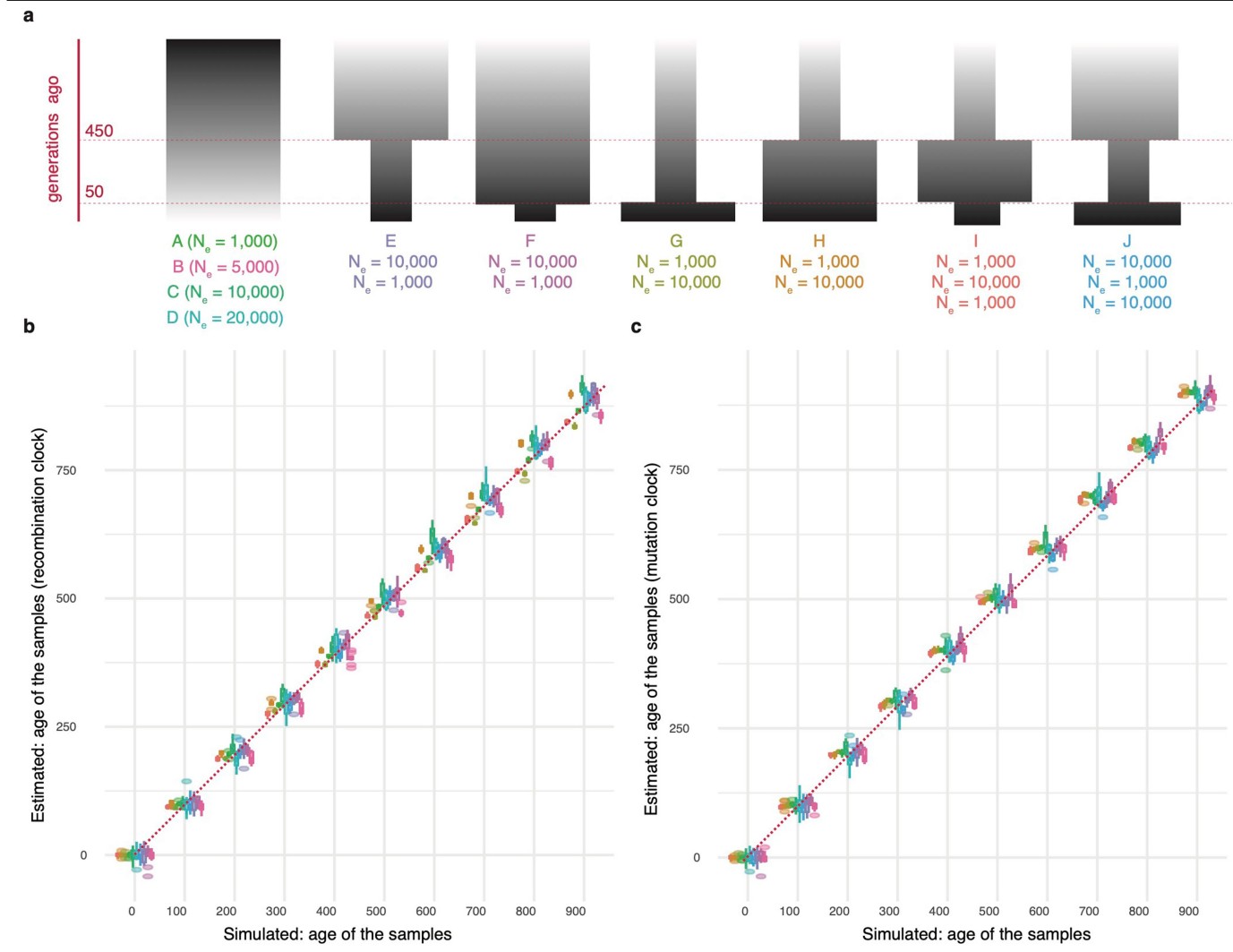

**Extended Data Fig. 10 | Coalescent simulations to validate both methods.**
**a)** Illustration of the 10 simulated scenarios (A-J), together with their underlying parameters. **b)** Each boxplot summarizes the estimates obtained from n = 10 diploid samples, when using the method relying on the recombination clock (in generations of evolution from the MRCA). Boxplots are comprised of their corresponding centres (median), box boundaries (interquantile ranges), and whiskers (1.5 times the interquantile ranges). The estimated age of the samples perfectly correlates with the simulated age of sampling (Pearson correlation; $r = 0.999$; two-tailed $p$-value = 0). **c)** Same as b) for the mutation clock (Pearson correlation; $r = 0.999$; two-tailed $p$-value = 0).

# Reporting Summary

## Statistics

For all statistical analyses, confirm that the following items are present in the figure legend, table legend, main text, or Methods section.

| n/a | Confirmed | |
|---|---|---|
| ☐ | ☒ | The exact sample size (*n*) for each experimental group/condition, given as a discrete number and unit of measurement |
| ☒ | ☐ | A statement on whether measurements were taken from distinct samples or whether the same sample was measured repeatedly |
| ☐ | ☒ | The statistical test(s) used AND whether they are one- or two-sided<br>*Only common tests should be described solely by name; describe more complex techniques in the Methods section.* |
| ☐ | ☒ | A description of all covariates tested |
| ☐ | ☒ | A description of any assumptions or corrections, such as tests of normality and adjustment for multiple comparisons |
| ☐ | ☒ | A full description of the statistical parameters including central tendency (e.g. means) or other basic estimates (e.g. regression coefficient) AND variation (e.g. standard deviation) or associated estimates of uncertainty (e.g. confidence intervals) |
| ☐ | ☒ | For null hypothesis testing, the test statistic (e.g. *F*, *t*, *r*) with confidence intervals, effect sizes, degrees of freedom and *P* value noted<br>*Give P values as exact values whenever suitable.* |
| ☐ | ☒ | For Bayesian analysis, information on the choice of priors and Markov chain Monte Carlo settings |
| ☒ | ☐ | For hierarchical and complex designs, identification of the appropriate level for tests and full reporting of outcomes |
| ☐ | ☒ | Estimates of effect sizes (e.g. Cohen's *d*, Pearson's *r*), indicating how they were calculated |

*Our web collection on statistics for biologists contains articles on many of the points above.*

## Software and code

Policy information about availability of computer code

| Data collection | *Provide a description of all commercial, open source and custom code used to collect the data in this study, specifying the version used OR state that no software was used.* |
|---|---|
| Data analysis | QGIS 3.36 (https://www.qgis.org/en/site/about/index.html)<br>Angsd v0.927 (https://www.popgen.dk/angsd/index.php/ANGSD)<br>Plink v1.9 (https://www.cog-genomics.org/plink/1.9/)<br>Locator (https://github.com/kr-colab/locator)<br>Admixtools2 (https://uqrmaie1.github.io/admixtools/articles/admixtools.html)<br>Admixtools v7.0.2 (https://github.com/DReichLab/AdmixTools)<br>AdmixtureBayes (https://github.com/avaughn271/AdmixtureBayes)<br>Struct-f4 (https://bitbucket.org/plibradosanz/structf4/src/master/)<br>generation_time (https://bitbucket.org/plibradosanz/generationtime/src/master/)<br>GONE (https://github.com/esrud/GONE/) |

For manuscripts utilizing custom algorithms or software that are central to the research but not yet described in published literature, software must be made available to editors and reviewers. We strongly encourage code deposition in a community repository (e.g. GitHub). See the Nature Portfolio guidelines for submitting code & software for further information.

## Data

Policy information about availability of data

All manuscripts must include a data availability statement. This statement should provide the following information, where applicable:

- Accession codes, unique identifiers, or web links for publicly available datasets
- A description of any restrictions on data availability
- For clinical datasets or third party data, please ensure that the statement adheres to our policy

All collapsed and paired-end sequence data for samples sequenced in this study are available in compressed FASTQ format through the European Nucleotide Archive under accession number PRJEB71445, together with rescaled and trimmed bam sequence alignments against both the nuclear horse reference genomes. Previously published ancient data used in this study are available under accession numbers PRJEB7537, PRJEB10098, PRJEB10854, PRJEB22390, PRJEB31613, and PRJEB44430, and detailed in Supplementary Table 1. The genomes of 78 modern horses, publicly available, were also accessed as indicated in their corresponding original publications, and in Supplementary Table 1.

## Research involving human participants, their data, or biological material

Policy information about studies with human participants or human data. See also policy information about sex, gender (identity/presentation), and sexual orientation and race, ethnicity and racism.

| | |
|---|---|
| Reporting on sex and gender | *Use the terms sex (biological attribute) and gender (shaped by social and cultural circumstances) carefully in order to avoid confusing both terms. Indicate if findings apply to only one sex or gender; describe whether sex and gender were considered in study design; whether sex and/or gender was determined based on self-reporting or assigned and methods used. Provide in the source data disaggregated sex and gender data, where this information has been collected, and if consent has been obtained for sharing of individual-level data; provide overall numbers in this Reporting Summary. Please state if this information has not been collected. Report sex- and gender-based analyses where performed, justify reasons for lack of sex- and gender-based analysis.* |
| Reporting on race, ethnicity, or other socially relevant groupings | *Please specify the socially constructed or socially relevant categorization variable(s) used in your manuscript and explain why they were used. Please note that such variables should not be used as proxies for other socially constructed/relevant variables (for example, race or ethnicity should not be used as a proxy for socioeconomic status). Provide clear definitions of the relevant terms used, how they were provided (by the participants/respondents, the researchers, or third parties), and the method(s) used to classify people into the different categories (e.g. self-report, census or administrative data, social media data, etc.) Please provide details about how you controlled for confounding variables in your analyses.* |
| Population characteristics | *Describe the covariate-relevant population characteristics of the human research participants (e.g. age, genotypic information, past and current diagnosis and treatment categories). If you filled out the behavioural & social sciences study design questions and have nothing to add here, write "See above."* |
| Recruitment | *Describe how participants were recruited. Outline any potential self-selection bias or other biases that may be present and how these are likely to impact results.* |
| Ethics oversight | *Identify the organization(s) that approved the study protocol.* |

Note that full information on the approval of the study protocol must also be provided in the manuscript.

# Field-specific reporting

Please select the one below that is the best fit for your research. If you are not sure, read the appropriate sections before making your selection.

☐ Life sciences   ☐ Behavioural & social sciences   ☒ Ecological, evolutionary & environmental sciences

For a reference copy of the document with all sections, see [nature.com/documents/nr-reporting-summary-flat.pdf](http://nature.com/documents/nr-reporting-summary-flat.pdf)

# Ecological, evolutionary & environmental sciences study design

All studies must disclose on these points even when the disclosure is negative.

| | |
|---|---|
| Study description | We generate 124 new ancient genomes, and combined them with 434 previously available to identify when domestic horses were extensively used for long-distance mobility |
| Research sample | The 550 horse genomes used in this study were used to provide a comprehensive representation of the horse genetic diversity prior and during domestication across Eurasia. The rationale was to identify shifts in the evolutionary trajectory of horses induced by humans, following their domestication. Eight outgroups were also included to polarise alleles as ancestral or derived. A full description of each new sample is provided in Table S1. |
| Sampling strategy | In the field of ancient DNA, sampling strategies are conditioned by the levels of endogenous DNA preservation in ancient remains. We attempted to (and succeeded) generate the largest genomic time-series for a non-human species. This was larger than our own |

| | |
|---|---|
| | previous studies (eg. Librado et al. 2021), where the corresponding sampling size was already proven more than sufficient to perform evolutionary analyses |
| Data collection | Fossil remains were collected from across Eurasia, from coauthors of this study. Sequencing of these remains was performed at the dedicated facilities of CAGT (Toulouse). The contribution of each coauthor is detailed in the corresponding section of the main manuscript, and the whole process was registered in our internal database. |
| Timing and spatial scale | Ancient DNA samples were processed for whole genome sequenced as they arrived to our laboratory, shipped by our coauthors. No particular strategy was followed in this regard. |
| Data exclusions | All data included in this study was analysed, with the only exception pertaining to a few samples showing signatures of introgression from basal lineages (true outliers) or not radiocarbon-dated in the regression analyses, as openly explained in the supplementary information. |
| Reproducibility | All our experiments were found to be highly reproducible. Those experiments with lower reproducibility, pertaining to complex statistical inference, were repeated multiple times with different starting values and parameters to check for concordance between runs (eg. Locator runs). Only solid analyses are reported. |
| Randomization | We followed the population group assignment from Librado et al. (2021; Nature) |
| Blinding | Blinding was not relevant in our ancient DNA study. |

Did the study involve field work? ☒ Yes ☐ No

## Field work, collection and transport

| | |
|---|---|
| Field conditions | Fossil remains were collected across Eurasia during years, if not decades, during a diversity of conditions. These do not impact our conclusions as fossils have been buried for hundreds to thousands of years before being sampled during field work. |
| Location | All relevant parameters, including radiocarbon dates and GPS coordinates for each new ancient sample sequenced in this study, are provided in Table S1 |
| Access & import/export | All fossils were collected strictly following the highest standards in ancient DNA research, and in close coordination with the archaeologists responsible for the material and the corresponding excavations, with all local and international permits in place. All these archaeologists are coauthors in our study. |
| Disturbance | This study caused no disturbance |

# Reporting for specific materials, systems and methods

We require information from authors about some types of materials, experimental systems and methods used in many studies. Here, indicate whether each material, system or method listed is relevant to your study. If you are not sure if a list item applies to your research, read the appropriate section before selecting a response.

### Materials & experimental systems

| n/a | Involved in the study |
|---|---|
| ☒ | ☐ Antibodies |
| ☒ | ☐ Eukaryotic cell lines |
| ☐ | ☒ Palaeontology and archaeology |
| ☒ | ☐ Animals and other organisms |
| ☒ | ☐ Clinical data |
| ☒ | ☐ Dual use research of concern |
| ☒ | ☐ Plants |

### Methods

| n/a | Involved in the study |
|---|---|
| ☒ | ☐ ChIP-seq |
| ☒ | ☐ Flow cytometry |
| ☒ | ☐ MRI-based neuroimaging |

## Palaeontology and Archaeology

| | |
|---|---|
| Specimen provenance | The samples that were analyzed in this study were collected from a range of archaeological contexts, as detailed in the Supplementary Information and summarized in Supplementary Table S1. As this involved sampling from across Eurasia and different procedures between countries and institutions, key contact persons were identified in each country so as to access relevant material and coordinate legal authorization to sample material for DNA analysis and radiocarbon dating. Samples were collected with permission from the organizations holding the collections and documented through official agreement letters provided by the named archaeologists and/or curators and/or directors of relevant institutions, named below. As DNA and radiocarbon dating techniques are partially destructive, we sought every opportunity to access samples as part of collaborations with other research projects so as to both save resources and avoid double sampling, and, thus, ultimately minimize destruction. The following list provides the sites and |

names of those key contact persons, who granted access to the corresponding material, with reference to letters and permits where appropriate:

-Albufeira (Silo 1, Rua Henrique Calado), Portugal. Sample: Albufeira1x2_Spa_1224. Key contacts: Maria João Valente, and Luís Paulo (Museu Municipal de Arqueologia, Albufeira). Collection of the Museu Municipal de Arqueologia, Albufeira.

-Alorda Park, Calafell, Spain. Sample: H9020_Spa_m291 (SU 9020 - Bottom level of the filling of the ditch surrounding the fortified aristocratic residence). Key contact: Silvia Valenzuela-Lamas (Archaeology of Social Dynamics, Institució Milà i Fontanals - Consejo Superior de Investigaciones Científicas (IMF-CSIC), C/ Egipcíaques 15, 08001 Barcelona, Spain).

-Arzhan-2, Russia. Sample: Rus8_Rus_m855. Key contacts: Aleksei K. Kasparov, Vladimir V. Pitulko (Institute of Material Culture, Russian Academy of Sciences). Sampled through the project 21-18-00457 from the Russian Science Foundation, with permission and all proper authority (confirmation letter nb 14102/33-772.4-263).

-At Daban, Yakutia. Sample: ATDABAN13_Yak_1725. Key contact: Éric Crubézy (Centre d'Anthropobiologie et de Génomique de Toulouse, CNRS UMR 5288, Université Paul Sabatier, Faculté de Médecine Purpan, 37 Allées Jules Guesde, 31000 Toulouse, France).

-Bakonszeg-Kádárdomb, Hungary. Sample: BAK1_Hun_m1686. Key contacts: Lajos Lakner, Dani János, Katherine Kanne (Department of Archaeology and History, University of Exeter, Exeter EX4 4QE, UK). Exported in 2008-2009 under National Science Foundation Dissertation Improvement Grant nb 0833106, Wenner-Gren Foundation for Anthropological Research Dissertation Fieldwork Grant nb 7896). Collections Institution: Déri Museum.

-Berettyóújfalu-Szilhalom, Hungary. Sample: BS11_Hun_m1682. Key contacts: Lajos Lakner, Dani János, Katherine Kanne (Department of Archaeology and History, University of Exeter, Exeter EX4 4QE, UK). Exported in 2008-2009 under National Science Foundation Dissertation Improvement Grant nb 0833106, Wenner-Gren Foundation for Anthropological Research Dissertation Fieldwork Grant nb 7896). Collections Institution: Déri Museum.

-Biluut 2, Zeerdegchingiin Khoshuu, Zunii Gol, Zuunkhangai, Ulaan Tolgoi, Mongolia: Samples M17x152x1_Mon_m80 and M17x156_Mon_m299. Key contact: Jamsranjav Bayarsaikhan (Institute of Archaeology, Mongolian Academy of Science, Ulaanbaatar 13330, Mongolia), and Will T. T. Taylor (Museum of Natural History, University of Colorado Boulder, Boulder, CO 80309, USA; Fulbright US Student research award nb 34154234, National Geeographic Young Explorer's grant nb 9713-15), National Science Foundation Doctoral Dissertation Improvement Grant nb 1522024). Exported in 2015 and 2015 under research agreement nb 20150315. Collections from the National Museum of Mongolia, Ulaanbaatar

-Bitozeves, Czechia. Sample: PRA29_Cze_471. Key contact: René Kyselý, Institute of Archaeology of the Czech Academy of Sciences, Prague. Excavated by Věra Sušická, ÚAPPSZČ, Most, Czechia.

-Bleachfield Street, Alcester, Warwickshire, United Kingdom. Key contact: Jacobo Weinstock (Faculty of Arts and Humanities, Department of Archaeology, University of Southampton, UK). Collection from the site curated by Warwickshire Museum.

-Bled, Pristava necropolis, Slovenia. Samples: SRSLO012_Slo_m197. Key contact: Peter Turk (Narodni muzej Slovenije, Prešernova 20, SI-1000 Ljubljana, Slovenia). Samples were made available through the Innovation Fund of the Austrian Academy of Sciences (ÖAW) (Grant agreement IF_2015_17).

-Borly IV, Kazakhstan. Sample: BorlyXIII_Kaz_m4638 and Borly9_Kaz_1807 (SQ 7G (33-44cm)). Key contact: Alan Outram (Department of Archaeology and History, University of Exeter, Exeter EX4 4QE, UK). Collections from the A. Kh. Margulan Joint Archaeological Research Centre Toraighyrov University (Director: Viktor K. Merts), sampled with permission and all proper authority from the Acting Deputy Chairman of the Board for Academic Work (confirmation letter 0605-2021 nb. 107-1232).

-Botai, Kazakhstan. Samples: Botai18x30_Kaz_m3328 and BotaiB_Kaz_m3228. Key contact: Prof Alan Outram (Department of Archaeology and History, University of Exeter, Exeter EX4 4QE, UK). Collections from the Al Farabi Kazakh National University, sampled with permission and all proper authority from the Dean of Faculty of History M.S. Nogaibayeva (confirmation letter 605-2021 nb. 1523-602).

-Bradgate Park, Leicestershire, United Kingdom. Sample: BGPSK1_UK_1661. Key contact: Richard Thomas (School of Archaeology and Ancient History, University of Leicester). Permission for analysis was provided by the Bradgate Park Trust. The specimen is curated at the School of Archaeology and Ancient History, University of Leicester, but will eventually be deposited with Leicestershire County Council Museums Service under the accession code XA19.2015.

-Bredholm, Denmark. Samples: P128_Den_1458, P129_Den_1457, P130_Den_1459, and P131_Den_1461. Key contact: Peter Pentz (museum inspector; National Museum of Denmark, Ny Vestergade 10, 1471 Copenhagen K., Denmark). Sampled with permission from the collection manager at the Collections of the Natural History Museum of Denmark.

-Brusyany IV, Kurgan 2, mound fill, Russia. Sample: LR18x70_Rus_608. Key contact: Pavel Kuznetsov (Department of Russian History and Archaeology, Samara State University of Social Sciences and Education, Samara, Russia). Sampled with proper authorization (confirmation letter nb 03-01-Myzea).

-Brusyany IV, Kurgan 1, Russia. Sample: LR18x68_Rus_592 (Grave 1, horse 1). Key contact: Pavel Kuznetsov (Department of Russian History and Archaeology, Samara State University of Social Sciences and Education, Samara, Russia). Sampled with proper authorization (confirmation letter nb 03-01-Myzea).

-Budapest-Királyok Útja 293, Hungary. Samples KU1153_Hun_m2301, KU1591_Hun_m2304, KU1701_Hun_m1822, KU2102_Hun_m2211, KU2210_Hun_m2218, and KU709_Hun_m2335. Key contacts: Paula Zsidi, Alice Choyke, Katherine Kanne (Department of Archaeology and History, University of Exeter, Exeter EX4 4QE, UK). Exported in 2008-2009 under National Science Foundation Dissertation Improvement Grant nb 0833106, Wenner-Gren Foundation for Anthropological Research Dissertation Fieldwork Grant nb 7896). Collections Institution: Aquincum Museum.

-Burgast, Mongolia: Samples GVA9046_Mon_716. Key contacts: Sébastien Lepetz (CNRS, Muséum national d'Histoire naturelle, Archéozoologie, Archéobotanique (AASPE), CP 56, Paris, France), Tsagaan Turbat (Archaeological Research Center and Department of Anthropology and Archaeology, National University of Mongolia, Ulaanbaatar, Mongolia), and Bayarkhuu Noost (Archaeological Research Center and Department of Anthropology and Archaeology, National University of Mongolia, Ulaanbaatar, Mongolia). Excavation campaign from 2016, Program MEAE – Institut of Archaeology, Mongolian Academy of Sciences, Ulaanbaatar, Mongolia.

-Çadır Höyük, Yozgat, Türkiye. Sample: CD5041_Tur_m314. Key contact: Benjamin Arbuckle (Department of Anthropology, Alumni Building, University of North Carolina at Chapel Hill, Chapel Hill, NC, USA). Exported in 2013 to Benjamin Arbuckle via the Yozgat Müze Müdürlüğü under grant (NSF BCS-1311551).

-Can Roqueta-Torre Romeu, Spain. Sample: CRTR279_Spa_506 (Barcelona, Spain - Late Roman, structure CRTR-279). Key contact: Silvia Albizuri (Institut d'Arqueologia, Universitat de Barcelona).

-Chekon settlement, Russia. Sample: KUZ3_Rus_m954. Key contact: Pavel Kuznetsov (Department of Russian History and Archaeology, Samara State University of Social Sciences and Education, Samara, Russia). Sampled with proper authorization (confirmation letter nb 03-01-Myzea).

-Dava Goz, Iran. Sampled: DavaG1_Ira_m4615. Key contact: Marjan Mashkour (CNRS, Muséum national d'Histoire naturelle, Archéozoologie, Archéobotanique (AASPE), CP 56, Paris, France).

-Derkul, Russia. Sample: NB4_Kaz_m4210. Key contacts: Pavel Kosintsev (Institute of Plant and Animal Ecology, Ural Branch of the Russian Academy of Sciences, Ekaterinburg, Russia), and Mélanie Pruvost (UMR5199, PACEA, Université de Bordeaux, France).

-Dunakeszi-Székesdűlő, Hungary. Samples: DS101_Hun_m2271 and DS113_Hun_m2222. Key contacts: Paula Zsidi, Alice Choyke, Katherine Kanne (Department of Archaeology and History, University of Exeter, Exeter EX4 4QE, UK). Exported in 2008-2009 under National Science Foundation Dissertation Improvement Grant nb 0833106, Wenner-Gren Foundation for Anthropological Research Dissertation Fieldwork Grant nb 7896). Collections Institution: Aquincum Museum.

-Eketorp, Sweden. Sample: F612_Swe_495. Key contacts: Key contacts: Johnny Karlsson (curator, The Swedish History Museum), Marie Sundquist (Östra Greda Research Group, 38791 Borgholm, Sweden), and Gabriella Lindgren (Department of Animal Breeding and Genetics, Swedish University of Agricultural Sciences, Uppsala, Sweden). Loan agreement: 331-2017-958.

-Egying Gol, Mongolia. Samples: EGI10_Mon_17 (Tomb 10), EGI12x2_Mon_41 (Tomb 12.2), EGI67_Mon_14 (Tomb 67), and EGI69_Mon_75 (Tomb 69). Key contacts: Éric Crubézy (Centre d'Anthropobiologie et de Génomique de Toulouse, CNRS UMR 5288, Université Paul Sabatier, Faculté de Médecine Purpan, 37 Allées Jules Guesde, 31000 Toulouse, France), and Tsagaan Turbat (Archaeological Research Center and Department of Anthropology and Archaeology, National University of Mongolia, Ulaanbaatar, Mongolia).

-El Graell (Vic), Spain. Sample: ADNUB33_Spa_m64 (Ibero Roman, Spain; structure E-42). Key contacts: F. Javier López-Cachero (Can Roqueta Project Manager, Ref: ARQ001SOL-178-2022), and Silvia Albizuri (Institut d'Arqueologia, Universitat de Barcelona).

-Filippovka II, Kurgan 1, Grave 2, Russia. Sample: LR18x84_Rus_m294. Key contact: Pavel Kuznetsov (Department of Russian History and Archaeology, Samara State University of Social Sciences and Education, Samara, Russia). Sampled with proper authorization (confirmation letter nb 03-01-Myzea).

-Følenslev Mose, Denmark. Sample: P133_Den_O1663. Key contact: Peter Pentz (museum inspector; National Museum of Denmark, Ny Vestergade 10, 1471 Copenhagen K, Denmark). Sampled with permission from the collection manager at the Collections of the Natural History Museum of Denmark.

-Gáborján-Csapszékpart, Hungary. Samples: GC77486_Hun_m1681 and GC77550_Hun_m1751. Key contacts: Lajos Lakner, Dani János, Katherine Kanne (Department of Archaeology and History, University of Exeter, Exeter EX4 4QE, UK). Exported in 2008-2009 under National Science Foundation Dissertation Improvement Grant nb 0833106, Wenner-Gren Foundation for Anthropological Research Dissertation Fieldwork Grant nb 7896). Collections Institution: Déri Museum.

-Ginnerup, Denmark. Samples: GIN1020_Den_m3000 (structure A4), GIN1055_Den_m3000 (structure A4), GIN396_Den_m3000 (structure A1), GIN489_Den_m3000 (structure A1), GIN561_Den_m3000 (structure A1). Key contact: Lutz Klassen (Museum Østjylland, Randers, Denmark).

-Gørlev, Denmark. Samples P187_Den_1334. Key contact: Peter Pentz (museum inspector; National Museum of Denmark, Ny Vestergade 10, 1471 Copenhagen K, Denmark). Sampled with permission from the collection manager at the Collections of the Natural History Museum of Denmark.

-Halvay, Kazakhstan. Sample: Halvai4_Kaz_342. Key contact: Prof Alan Outram (Department of Archaeology and History, University of Exeter, Exeter EX4 4QE, UK). Sampled with permission and all proper authority from the Acting Vice-Rector on Science, Internationalization and Digitalization Gulshat Shaikamal (confirmation letter 11.05.2021 nb. 15-20-09/1052). Collections from the A. Baitursynov Kostanay State University, KSU (Kostanay).

-Hereuet, Seró, Spain. Sample: H2012x137_Spa_m204 (SU 2012 - Filling of silo SJ-8). Key contact: Silvia Valenzuela-Lamas (Archaeology of Social Dynamics, Institució Milà i Fontanals - Consejo Superior de Investigaciones Científicas (IMF-CSIC), C/ Egipcíaques 15, 08001 Barcelona, Spain).

-Hjortspringkobbel, Denmark. Sample: P137_Den_m291. Key contact: Collections of the Natural History Museum of Denmark). Key contact: Kristian M. Gregersen (former collection manager of Quaternary Zoology, Natural History Museum of Denmark, Gothersgade 130, 1123 Copenhagen K., Denmark). Sampled with permission from the collection manager at the Collections of the Natural History Museum of Denmark.

-Hovmarken, Denmark. Sample: P138_Den_1224. Key contact: Kristian M. Gregersen (former collection manager of Quaternary Zoology, Natural History Museum of Denmark, Gothersgade 130, 1123 Copenhagen K., Denmark). Sampled with permission from the collection manager at the Collections of the Natural History Museum of Denmark.

-Hungate, United Kingdom. Sample: VEM107_UK_956. Key contact: Terry O'Connor (Department of Archaeology, University of York, c/o Kings Manor, York YO1 7EP, UK ).Excavated by York Archaeological Trust between 2006 and 2011 from whom permission was granted to Terry O'Connor, University of York, for ancient DNA study in 2013 under exit documentation X0903 and X0937.

-Husiatyn, Ukraine. Sample: POZ54_Ukr_m1498 (Double burial). Key Contact: Daniel Makowiecki (Institute of Archaeology, Faculty of History, Nicolaus Copernicus University, Toruń, Poland). Two horse skeletons were discovered during rescue excavations (2015) of barrow by Vasyl Ilchyshyn (Zaliztsi Museum, Ternopil Region, Ukraine, and the Security Archaeological Service of the Institute of Archeology, National Academy of Sciences of Ukraine, Kyiv, Ukraine). Samples were collected during zooarchaeological research (2021) by Daniel Makowiecki and Przemysław Makrowicz (Faculty of Archaeology Adam Mickiewicz University, Poznań), and stored since at the Institute of Archaeology, Nicolaus Copernicus, Torun. One of the two horses was sampled for the DNA and radiocarbon analysis presented in this study.

-Idzhil, Russia. Sample: IDZH_Rus_734. Marjan Mashkour (CNRS, Muséum national d'Histoire naturelle, Archéozoologie, Archéobotanique (AASPE), CP 56, Paris, France).

-Industriya, Russia. Sample: KAU27B_Rus_m627. Key contacts: Sabine Reinhold and Svend Hansen (Eurasia Department of the German Archaeological Institute, Berlin, Germany). Excavation carried out by Dr. D. S. Korobov (Institue of Archaeology, Russian Academy of Sciences, Moscow, Russia, licence nb 2001-868), with material curated by Ltd. 'Nasledie'. Exported in 2016 with proper authorization to the German Archaeological Institute, Berlin, Germany.

-Ipatovo 3, Russia. Sample: KAU22_Rus_m877 (Kurgan 2, Animal Complex 13). Key contacts: Sabine Reinhold and Svend Hansen (Eurasia Department of the German Archaeological Institute, Berlin, Germany). Excavation carried out by Dr. A.B. Belinskij (Stavropol, excavation Ltd. 'Nasledie' & DAI, Eurasia-Department, license nb 1998-177). Exported in 2013 with proper authorization to the German Archaeological Institute, Berlin, Germany.

-Katanda II, Russia. Samples: Kat2x4_Rus_m112 and Katx11_Rus_m269. Key contact: Alexey A. Tishkin (Department of Archaeology, Ethnography and Museology, Altai State University, Prospekt Lenina, 61, 656049 Barnaul, Russia). Sampled with proper permission and authority by Alexey A. Tishkin from auxiliary collection of the Department of Archaeology, Ethnography and Museology of the Altai State University, under the framework of the Russian Science Foundation project "The world of ancient nomads of Inner Asia: interdisciplinary studies of material culture, sculptures and economy" (No. 22-18-00470).

-Karatomar Burial ground, Kazakhstan (Kurgan 1). Sample: Karat17039_Kaz_m1834. Key contact:
Prof Alan Outram (Department of Archaeology and History, University of Exeter, Exeter EX4 4QE, UK). Sampled with permission and all proper authority from the Acting Vice-Rector on Science, Internationalization and Digitalization Gulshat Shaikamal (confirmation letter 11.05.2021 nb. 15-20-09/1052). Collections from the A. Baitursynov Kostanay State University, KSU (Kostanay).

-Khankarinsky dol, Russia. Sample: Han12_Rus_m296. Key contact: Alexey A. Tishkin (Department of Archaeology, Ethnography and Museology, Altai State University, Prospekt Lenina, 61, 656049 Barnaul, Russia). Sampled with proper permission and authority by Alexey A. Tishkin from auxiliary collection of the Department of Archaeology, Ethnography and Museology of the Altai State University, under the framework of the Russian Science Foundation project "The world of ancient nomads of Inner Asia: interdisciplinary studies of material culture, sculptures and economy" (No. 22-18-00470).

-Kittsee settlement, Steinfeldäcker, Austria. Sample: KT46_Aus_m3240 (campaign 1997, pit 289). Key contact: Christian Mayer (Federal Monuments Authority Austria, Department for Digitalization and Knowledge Transfer, Vienna, Austria). Excavation documentation and excavated material accessible through the Federal Monuments Authority Austria, Department of Archaeology.

-Køge A ved Spanager, Denmark. Sample: P134_Den_862. Key contact: Kristian M. Gregersen (former collection manager of Quaternary Zoology, Natural History Museum of Denmark, Gothersgade 130, 1123 Copenhagen K., Denmark). Sampled with permission from the collection manager at the Collections of the Natural History Museum of Denmark.

-Krasnosamarskoe settlement, Russia. Sample: RN85_Rus_865. Key contact: Pavel Kuznetsov (Department of Russian History and Archaeology, Samara State University of Social Sciences and Education, Samara, Russia). Sampled with proper authorization (confirmation letter nb 03-01-Myzea).

-Krasnosamarskoe IV, kurgan cemetery, Kurgan 7, Russia. Sample: RN10_Rus_1131 (horse bone in the mound fill layer). Key contact: Pavel Kuznetsov (Department of Russian History and Archaeology, Samara State University of Social Sciences and Education, Samara, Russia). Sampled with proper authorization (confirmation letter nb 03-01-Myzea).

-Krasny Gorodok settlement, Russia. Sample: ABA3_Rus_1333. Key contact: Pavel Kuznetsov (Department of Russian History and Archaeology, Samara State University of Social Sciences and Education, Samara, Russia). Sampled with proper authorization (confirmation letter nb 03-01-Myzea).

-Krefeld-Gellep, Germany. Sample Kref4_Ger_48. Key contact: Sabine Deschler-Erb and Monika Schernig Mráz (Integrative Prehistory and Archaeological Science, University Basel). Sample made available through Project HumAnimAl: Swiss National Science Foundation 178834. Sample reference NI 2017/0030 2708-12, Museum Burg Linn, Germany.

-Le Cendre – Gondole, France: Samples GVA629_Fra_m112, and GVA636_From_m197. Key contact: Sébastien Lepetz (Muséum national d'Histoire naturelle, Archéozoologie, Archéobotanique (AASPE), CP 56, Paris, France). Excavation campaign from 2003.

-Langhøj, Denmark. Samples P189_Den_1541. Key contact: Kristian M. Gregersen (former collection manager of Quaternary Zoology, Natural History Museum of Denmark, Gothersgade 130, 1123 Copenhagen K., Denmark). Sampled with permission from the

collection manager at the Collections of the Natural History Museum of Denmark.

-Maison Alfort Museum of the Veterinarian School, France (7 Av du Général De Gaulle, 94704 Maisons Alfort, France). Sample: Alfort3_Fra_1806: Key contacts: Christophe Degueurce, and Céline Robert (Ecole Nationale Vétérinaire d'Alfort, 7 Avenue du Général De Gaulle, 94704 Maisons-Alfort, France). Sampled with permission from the collections of the Maison Alfort Museum of the Veterinarian School.

-Miciurin (Odaia), Moldavia. Sample: Miciurin01_Mol_794. Key contact: Arne Ludwig (Leibniz-Zentrum für Archäologie (LEIZA), Ludwig-Lindenschmit-Forum 1, 55116 Mainz, Germany).

-Noviye Kluchi III cemetery, Bronze Age Pokrovka Culture, Kurgan 1, sacrificial complex, Russia. Sample: LR18x3_Rus_m1820. Key contact: Pavel Kuznetsov (Department of Russian History and Archaeology, Samara State University of Social Sciences and Education, Samara, Russia). Sampled with proper authorization (confirmation letter nb 03-01-Myzea).

-Nytorv, Denmark. Sample: P139_Den_1381. Key contact: Kristian M. Gregersen (former collection manager of Quaternary Zoology, Natural History Museum of Denmark, Gothersgade 130, 1123 Copenhagen K., Denmark). Sampled with permission from the collection manager at the Collections of the Natural History Museum of Denmark.

-Orcet – La Roche Blanche – L'Enfer, France: Samples GVA637_Fra_m64 and GVA639_Fra_2. Key contact: Sébastien Lepetz (CNRS, Muséum national d'Histoire naturelle, Archéozoologie, Archéobotanique (AASPE), CP 56, Paris, France). Excavation campaign from 2003.

-Pech Maho, France. Sample: Pech126_Fra_m288. Key contact: Armelle Gardeisen (CNRS, Archéologie des Sociétés Méditerranéennes, Archimède IA-ANR-11-LABX-0032-01, Université Paul Valéry, Montpellier 34090, France).

-Puig de Sant Andreu, Spain (Ullastret). Sample: UE14029_Spa_m293 (SU14029 - abandoned layer of the main residence; zona 14 of this urban site of more than 10Ha). Key contact: Silvia Valenzuela-Lamas (Archaeology of Social Dynamics, Institució Milà i Fontanals - Consejo Superior de Investigaciones Científicas (IMF-CSIC), C/ Egipcíaques 15, 08001 Barcelona, Spain).

-Rathewitz 13, Burgenland district, Saxony-Anhalt, Central Germany. Sample: Rat13_Ger_474 (deposited in grave 13 of a burial ground from the Migration period, 5th/6th century). Key contact: Hans-Jürgen Döhle (former curator of State Museum of Prehistory, Halle (Saale), Germany). Stored in the State Office for Heritage Management and Archaeology Saxony-Anhalt - State Museum of Prehistory, Halle (Saale), Germany.

-Rislev, Denmark. Sample:  P188_Den_149. Key contact: Kristian M. Gregersen (former collection manager of Quaternary Zoology, Natural History Museum of Denmark, Gothersgade 130, 1123 Copenhagen K., Denmark). Sampled with permission from the collection manager at the Collections of the Natural History Museum of Denmark.

-Roseldorf, Austria. Samples: SRNHM007_Aus_m301 and SRNHM004_Aus_m294. Key contact: Erich Pucher (Naturhistorisches Museum Wien, Austria). Samples were made available through the Innovation Fund of the Austrian Academy of Sciences (ÖAW) (Grant agreement IF_2015_17).

-Sadgorod IV, Kurgan 2, sacrificial complex 2, Russia. Sample: LR18x5_Rus_1810. Key contact: Pavel Kuznetsov(Department of Russian History and Archaeology, Samara State University of Social Sciences and Education, Samara, Russia). Sampled with proper authorization (confirmation letter nb 03-01-Myzea).

-Sarengrad-Klopare, Croatia. Samples: SRKRO001_Cro_720 and SRKRO002_Cro_722. Key contact: Mario Novak (Centre for Applied Bioanthropology, Institute for Anthropological Research, Ljudevita Gaja 32, 10 000 Zagreb, Croatia), and Andrea Rimpf. (Ilok Town Museum, Šetalište o. Mladena Barbarića 5, 32236 Ilok, Croatia). Samples were made available through the Innovation Fund of the Austrian Academy of Sciences (ÖAW) (Grant agreement IF_2015_17).

-Százhalombatta-Földvár, Hungary. Samples: SZHB2027_Hun_m2033, SZHB2074_Hun_m2115, SZHB2079_Hun_m1575, SZHB2147_Hun_m2054, SZHB2158_Hun_m1820, SZHB2438_Hun_m1602, SZHB553_Hun_m1566, SZHB625_Hun_m984, SZHB734_Hun_m1576, and SZHB967_Hun_m1822. Key Contact: Magdolna Vicze, Katherine Kanne (Department of Archaeology and History, University of Exeter, Exeter EX4 4QE, UK). Exported in 2008-2009 under National Science Foundation Dissertation Improvement Grant nb 0833106, Wenner-Gren Foundation for Anthropological Research Dissertation Fieldwork Grant nb 7896). Collections Institution: Matrica Museum.

-Sepphoris, Israel. Sample: MV243_Isr_294. Key contact: Liora Kolska Horwitz (National Natural History Collections, Edmond J. Safra Campus, Givat Ram, The Hebrew University; Jerusalem 9190401, Israel) Sampled with permission and all proper authority from Zeev Weiss (Eleazar L. Sukenik Professor of Archaeology, Institute of Archaeology, The Hebrew University of Jerusalem), archaeologist in charge of site.

-Shanmava, China. Sample: Shanx1_Chi_m112. Key contact: Alexey A. Tishkin (Department of Archaeology, Ethnography and Museology, Altai State University, Prospekt Lenina, 61, 656049 Barnaul, Russia). Sampled with proper permission and authority by Alexey A. Tishkin from auxiliary collection of the Department of Archaeology, Ethnography and Museology of the Altai State University, under the framework of the Russian Science Foundation project "The world of ancient nomads of Inner Asia: interdisciplinary studies of material culture, sculptures and economy" (No. 22-18-00470).

-Shahr-i-Qumis, Iran. Sample: AM3_Ira_Modern. Key contact: Marjan Mashkour (CNRS, Muséum national d'Histoire naturelle, Archéozoologie, Archéobotanique (AASPE), CP 56, Paris, France).

-Shohidon, Tajikistan. Sample: Shohx1_Rus_720 (Grave 20). Key contact: Alexey A. Tishkin (Department of Archaeology, Ethnography and Museology, Altai State University, Prospekt Lenina, 61, 656049 Barnaul, Russia). Sampled with proper permission and authority by Alexey A. Tishkin from auxiliary collection of the Department of Archaeology, Ethnography and Museology of the Altai State University, under the framework of the Russian Science Foundation project "The world of ancient nomads of Inner Asia:

interdisciplinary studies of material culture, sculptures and economy" (No. 22-18-00470).

-Shumaevo I, Kurgan 5, ditch, Russia. Samples: LR18x65_Rus_1352 (SE sector, skull 5) and LR18x66_Rus_1353 (NE sector, Skull 1, depth -123). Key contact: Pavel Kuznetsov (Department of Russian History and Archaeology, Samara State University of Social Sciences and Education, Samara, Russia). Sampled with proper authorization (confirmation letter nb 03-01-Myzea).

-Skedemosse, Sweden. Sample: F139_Swe_342. Key contacts: Johnny Karlsson (curator, The Swedish History Museum), Marie Sundquist (Östra Greda Research Group, 38791 Borgholm, Sweden), and Gabriella Lindgren (Department of Animal Breeding and Genetics, Swedish University of Agricultural Sciences, Uppsala, Sweden). Loan agreement: 331-2017-958.

-Sosnovka 1, Russia. Samples: UR17x27_Rus_568. Key contact: Pavel Kosintsev (Institute of Plant and Animal Ecology, Ural Branch of the Russian Academy of Sciences, Ekaterinburg, Russia).

-Tamiryn Ulaan Khoshuu, Mongolia. Samples: TAM13_Mon_14 (Tomb 13) and TAM22xA2_Mon_5 (Tomb 22). Key contacts: Éric Crubézy (Centre d'Anthropobiologie et de Génomique de Toulouse, CNRS UMR 5288, Université Paul Sabatier, Faculté de Médecine Purpan, 37 Allées Jules Guesde, 31000 Toulouse, France), and Tsagaan Turbat (Archaeological Research Center and Department of Anthropology and Archaeology, National University of Mongolia, Ulaanbaatar, Mongolia).

-Tarquinia monumental complex, Italy. Sample: Tarquinia3206_Ita_275. Key contact: Giovanna Bagnasco Gianni (Dipartimento di Beni Culturali E Ambientali Etruscologia, Universita Degli Studi di Milano, Italy).

-Tominy, Poland. Samples: POZ327_Pol_m5127 (EQ_288, EQ_To6_07; No inv P18/09 (ob. 108), and POZ37_Pol_m5108 (No inv. 55/09; feature 115, layer 163). Key contact:  Daniel Makowiecki (Institute of Archaeology, Faculty of History, Nicolaus Copernicus University, Toruń, Poland). Samples are from animal remains excavated (2006 – 2016) by Marcin Szeliga (Institute of Archaeology, Maria Curie-Skłodowska University, Lublin, Poland), and were collected during zooarchaeological research (2018) by Daniel Makowiecki, and stored since at the Institute of Archaeology, Nicolaus Copernicus, Torun.

-Tuse Skole, Denmark. Sample: P135_Den_1806. Key contacts: Kirsten Christensen (museum inspector; Museum West Zealand, Forten 10, 4300 Holbaek, Denmark), and Lone Claudi-Hansen (museum inspector; Museum West Zealand, Forten 10, 4300 Holbaek, Denmark). Sampled with permission from the collection manager at the Collections of the Natural History Museum of Denmark.

-Tyubyak, Russia. Samples: ABA9_Rus_m1786 and ABA10_Rus_m1757. Key contact: Pavel Kosintsev (Institute of Plant and Animal Ecology, Ural Branch of the Russian Academy of Sciences, Ekaterinburg, Russia).

-Ulvehøj, Denmark. Sample: P191_Den_O1663. Key contact: Kristian M. Gregersen (former collection manager of Quaternary Zoology, Natural History Museum of Denmark, Gothersgade 130, 1123 Copenhagen K., Denmark). Sampled with permission from the collection manager at the Collections of the Natural History Museum of Denmark.

-Vejen, Denmark. Sample: P140_Den_O1660. Key contact: Kristian M. Gregersen (former collection manager of Quaternary Zoology, Natural History Museum of Denmark, Gothersgade 130, 1123 Copenhagen K., Denmark). Sampled with permission from the collection manager at the Collections of the Natural History Museum of Denmark.

-Vinkovci, Na-ma, Croatia. Samples: SRKRO009_Cro_m453 and SRKRO011_Cro_331. Key contact: Hrvoje Vulic (Vinkovci Municipal Museum, Trg bana Josipa Šokčevića 16, 32 100 Vinkovci, Croatia). Samples were made available through the Innovation Fund of the Austrian Academy of Sciences (ÖAW) (Grant agreement IF_2015_17).

-Yaloman-II, Russia. Sample: Yal2x24_Rus_m105. Key contact: Alexey A. Tishkin (Department of Archaeology, Ethnography and Museology, Altai State University, Prospekt Lenina, 61, 656049 Barnaul, Russia). Sampled with proper permission and authority by Alexey A. Tishkin from auxiliary collection of the Department of Archaeology, Ethnography and Museology of the Altai State University, under the framework of the Russian Science Foundation project "The world of ancient nomads of Inner Asia: interdisciplinary studies of material culture, sculptures and economy" (No. 22-18-00470).

-Yenikapi, Türkiye. Sample: KSK11b_Tur_829. Key contact: Vedat Onar (Osteoarchaeology Practice and Research Center and Department of Anatomy, Faculty of Veterinary Medicine, Istanbul University-Cerrahpaşa, Istanbul 34320, Turkey).

-Zayukovo 3, Russia. Samples: OSCAE16SP7x1_Rus_604 and OSCAE16SP8_Rus_577. Key contact: Anna Kadieva (State Historical Museum, Department of Archaeological Monuments, Moscow, Red Square 1, Moscow 109012, Russian Federation).

-Zoolongiyn am, Mongolia. Samples: Zoox1_Mon_333 and Zoox2_Mon_271. Key contact: Alexey A. Tishkin (Department of Archaeology, Ethnography and Museology, Altai State University, Prospekt Lenina, 61, 656049 Barnaul, Russia). Sampled with proper permission and authority by Alexey A. Tishkin from auxiliary collection of the Department of Archaeology, Ethnography and Museology of the Altai State University, under the framework of the Russian Science Foundation project "The world of ancient nomads of Inner Asia: interdisciplinary studies of material culture, sculptures and economy" (No. 22-18-00470).

Specimen deposition | Ancient remains were 3D-scanned in our facilities to ensure future morphometric studies, if needed, and stored in our laboratory facilities unless the bone fragment was inevitably destroyed during ancient DNA extraction. Based on the detailed archaeological metadata provided in Table S1, scholars are encouraged to contact Prof. Ludovic Orlando if aim to obtain further sample information.

Dating methods

> 140 samples were radiocarbon-dated. Their calibrated and uncalibrated dates, 95% confidence intervals, and relevant laboratory codes are provided in Table S1.

☒ Tick this box to confirm that the raw and calibrated dates are available in the paper or in Supplementary Information.

Ethics oversight

> *Identify the organization(s) that approved or provided guidance on the study protocol, OR state that no ethical approval or guidance was required and explain why not.*

Note that full information on the approval of the study protocol must also be provided in the manuscript.

# Plants

Seed stocks

> *Report on the source of all seed stocks or other plant material used. If applicable, state the seed stock centre and catalogue number. If plant specimens were collected from the field, describe the collection location, date and sampling procedures.*

Novel plant genotypes

> *Describe the methods by which all novel plant genotypes were produced. This includes those generated by transgenic approaches, gene editing, chemical/radiation-based mutagenesis and hybridization. For transgenic lines, describe the transformation method, the number of independent lines analyzed and the generation upon which experiments were performed. For gene-edited lines, describe the editor used, the endogenous sequence targeted for editing, the targeting guide RNA sequence (if applicable) and how the editor was applied.*

Authentication

> *Describe any authentication procedures for each seed stock used or novel genotype generated. Describe any experiments used to assess the effect of a mutation and, where applicable, how potential secondary effects (e.g. second site T-DNA insertions, mosiacism, off-target gene editing) were examined.*

