## [Peer Review File · Nature]

Manuscript Title: Widespread horse-based mobility arose around 2,200 BCE in Eurasia

Reviewer Comments & Author Rebuttals

Reviewer Reports on the Initial Version:

Referees' comments:

Referee #1 (Remarks to the Author):

Librado and colleagues' manuscript titled "Widespread horse-based mobility did not arise in Eurasia before 2,200 BCE" presents another important chapter in the history of horse domestication.

The large international team presents new genome data from 86 ancient horses and 93 new radiocarbon dates, which upon integration with published genome data from ancient horses and modern-day domestic breeds accounts for a very powerful time series.

A general point at the start:

Proper formatting, i.e., line spacing and use of line numbers would have made reviewing a more pleasant experience. I normally tend to provide more detailed feedback on smaller or minor issues, providing line and page numbers, but this is not possible here.

The three main aspects covered in the manuscript include a reanalysis of the CWC-associated genomes with respect to a potential involvement in the expansion of steppe pastoralists, the demographic trajectories of DOM2 and Botai horses, and lastly the evidence for controlled reproduction, such as signs of inbreeding and accelerated generation times.

The first aspect is addressed as a response to alternative admixture graph modelling results, which were seen as a better fit and used to weave the CWC-associated horses into a narrative in line with the 3rd millennium expansion of steppe pastoralists (Maier et al. 2023).

There are intrinsic problems with this scenario. First, the underlying assumptions expand on the simplistic equation of pots = people (= languages) = horses, which has already been shown to not be so

simple. Second, while the assumption of such a scenario could be deemed legitimate within a hypothesis testing framework, I remain unconvinced that the current dataset is capable of resolving the involvement of the CWC in the increased horse-based mobility that is said to be emerging in the mid-3rd millennium BC.

In particular, I wonder how representative the six genomes associated with the CWC horizon really are. The association of humans, their genetic ancestry and material culture is already problematic enough. A strong assignment of domesticated animals, which can be traded and exchanged or “appropriated”, to a particular archaeological culture, as done here, would be much more convincing if the data came from more than a single tentatively assigned CWC site. And even then, I would expect to see some variation unless there are in parallel strong indicators of breeding practices, for example as later shown for the DOM2 lineages.

In fact, while it is clear from the original analyses in the 2021 manuscript and the re-analyses shown here that the six “CWC-associated, or better “Final Neolithic”, genomes show a distinct ancestry profile that is more akin to central/northcentral Late and Final Neolithic genomes, it is the data from Czechia, southern Poland and Hungary (ENEOCZE, FBCPOL, HUNG) that show evidence for early gene flow from the Eneolithic horse lineages of the West Eurasian steppe and forest steppe as indicated by the negative f_4 -statistics in Fig. 1A.

I am wondering whether an AdmixtureBayes model was attempted that included one of the following groups of ENEOCZE, FBCPOL or HUNG in direct comparison to CWC. The data quality for ENEOCZE should suffice. In case there was a well-supported admixture node to ENEOCZE and none to CWC would further strengthen the argument against an introduction via a direct Yamnaya-associated lineage. However, more CWC or Bell Beaker-associated horse finds would certainly be favorable in a future study to say anything definite about CWC-associated horses.

As a consequence, as the role of the CWC cannot be solved in this manuscript without having sampled more broadly from early 3rd millennium BC cultural groups north of the Carpathian (the authors state so themselves on pg. 4), the ‘problem’ around CWC horses (or not?) is merely a side issue that should not take away the attention from the main findings by occupied a good third of the paper’s word limit.

Instead, on the basis of the available horse data from the chronological window ranging from ~3500-2500 BC, it could be stressed that critical developments are already underway in southeastern Europe

during the late 4th and the early 3rd millennium BC, involving a number of Eneolithic, Final Neolithic and Early Bronze Age groups and a mosaic of horse ancestries.

Analogously, a strict fixation on Yamnaya and whether a finding is compatible with its timeline is a strong simplification and becoming a hindrance in the recent literature. As far as horizons of genetic transformations, of contact, exchange and/or admixture on the human side are concerned, the “Yamnaya-related steppe migrations” (pg. 5) are a misnomer or nuisance parameter. The period from ~3300 to ~2800 BC is clearly not the only critical time period during which some human groups, pastoralists in particular, expanded. In fact, this period is bracketed in between time periods that describe the prelude and aftermath in many regions and time periods that are not or no longer linked to Yamnaya (see e.g., Olalde et al. 2018, Nature).

More importantly, the newly presented results on the estimated bottleneck and generation time estimates are way more interesting and convincing, and clearly moving the focus away from the Yamnaya-centered obsession that has manifested itself since the 2015 papers on human genetic transformations. This is a great achievement, but what I am missing is the contextualization of the other players post-dating the Yamnaya time period. For example, if the upper boundary of the DOM2 breeding in the native steppe homeland was set to 2,726 BCE as estimated in the paper, this would put the efforts of creating a strong domestication bottleneck in the hands of people associated with the Catacomb culture and contemporaneous groups. Catacomb-associated individuals are genetically virtually indistinguishable from preceding Yamnaya-associated individuals and are culturally and economically so similar that the naming categories we apply become more of a nuisance than a useful scheme to follow. In this sense, and as cultures usually do not fall from the sky but instead build on each other, as already argued above, I would recommend a less strict adherence to the seemingly monolithic cultural blocks and suggest using a broader interpretation in the light of the developments over the course of the 3rd millennium BCE. For example, the intensification of herding practices (Scott et al. 2022), increasing aridisation and/or overexploitation of the steppe environment might have caused the need to continuously expand the grazing areas, and thus further facilitated the use of domesticated horses to overcome larger geographic distances.

The in-depth evaluation of effective population sizes, lengths of generations, inbreeding and recombination clocks in DOM2 and Botai horses are highly interesting. Regarding the side-by-side comparison and the points addressed in the discussion, I recommend integrating a few critical statements and references from the SOM part, such as “The breeding of domestic animal can involve the reproduction within close kins, otherwise called inbreeding, as a practice aimed at propagating desirable traits⁸⁷.” See also first sentence on pg. 20.

This will help the general reader to understand which parameters are critical to distinguish between 'breeding/selection of specific traits' and 'management/husbandry as food source' of animals. Along the same line, a GONE graph in which the effective population sizes of both DOM2 and Botai are shown side-by-side could help to illustrate that some human intervention can lead to same effects (e.g., a drop in N_e), but that the differences in order of magnitude and other lines of evidences are very important for the interpretation of this signal.

The true merits of the paper lie clearly in the methodological developments of the latter two aspects, the observations of N_e bottlenecks and acceleration of generation times via mutation and recombination clocks. Both are described and delineated well in the SOM and I applaud the additional validation via coalescent simulations.

Additional (minor) points:

SOM, pg. 18

Typo in the equation: pemutations > permutations

SOM, pg. 18

The paragraph "DOM2 mobility on page 18-19 in the SOM file requires cross-references to the figures/tables, where the IBD results and the Mantel test are shown (Figure 2, I suppose?).

SOM, pg. 19 and Fig. 3D

"While Botai horses were found the least inbred (Fig. 3D),..."

The three-dimensional perspective is very hard to read. As the z-axis is the crucial factor in this argument, I suggest displaying the Mb windows as separate panels above each other the same x-axis time. That way, the fraction of the genome within each inbreeding category is much easier to compare across the split y-axis (i.e., turn z- to y-axis).

Fig. 3D

Furthermore, as this is a critical point in support of one of the main aspects, I suggest highlighting or labelling the "The earliest instances of inbreeding within DOM2 horses are provided by ancient horses

associated with Sintashta archaeological contexts, namely UR17x47_Rus_m1856 (0.107), RN03_Rus_m1851 (0.176) and UR17x31_Rus_m1845 (0.122),...” in the 15Mb+ window.

SOM, pg. 20

“Considering a realistic transition:transversion rate of 2.2 implies that one every 3.2 mutations actually corresponds to a transversion.” Change to “...one in every 3.2 mutations corresponds to a transversion.”

Referee #2 (Remarks to the Author):

The authors present a study on horse domestication and early spread using ancient genomes. It represents state-of-the-art if this field of science with a very impressive sampling and data generation effort. The paper is concise and well-written. I think the figure legends could do with some more detail on what the figure actually show.

The findings can be summarized into two main topics, which I cover in turn:

1. The spread of the “DOM2” ancestry profile associated with modern domestic horses
2. Changes in horse life history and genetic structure as a result of human management.

Topic 1: The key message is in the title of the paper and is the claim that the DOM2 horse ancestry profile did not spread widely until after 2200 BC. In particular, key to this claim is that horses associated with the Corded Ware Culture (CWC), who were basically the first western/central-European offshoot of the Yamnaya steppe herders, did not have any DOM2 ancestry. The important implication of the claim is that the DOM2 expansion would post-date the expansion of Yamnaya and the Indo-European language family, which often has been thought to have been horse-driven.

This finding was recently reported by the same research group in Librado et al. 2021. Nature. “The origins and spread of domestic horses from the Western Eurasian steppes”. The finding itself is thus not

a new one to this paper. Librado et al. 2021 wrote e.g.: “Our results reject the commonly held association between horseback riding and the massive expansion of Yamnaya steppe pastoralists into Europe around 3000 BC”, “The globalization stage started later, when DOM2 horses dispersed outside their core region, first reaching Anatolia, the lower Danube, Bohemia and Central Asia by approximately 2200 to 2000 BC”. This new paper can thus be seen as following up on, and providing additional evidence for, the 2200 BC claim. But the claim is the same one, and I imagine some readers would view this new paper as in some sense publishing the same finding twice.

The motivation to revisit the 2200BC claim comes primarily from a recent paper challenging that claim - Maier et al. 2023 (<https://elifesciences.org/articles/85492>) revisited the analyses that make use of so called ‘admixture graphs’ – population history models with gene flow - and argued that the finding that CWC horses did not have any DOM2 ancestry is not robust. A ~third of current paper can thus be viewed as a rebuttal to Maier et al. 2023.

The critique from Maier et al. 2023 and the response from the authors gets deep into the weeds of admixture graphs, and it’s challenging for outsiders to follow all the details here. Nonetheless, here is my attempted summary:

- Maier et al. point out that the original graph in Librado 2021 fig 3b actually fits the data very poorly, even though it was the best graph found by the algorithm employed. I think this is an important point – typically admixture graphs are considered to “fit” the data if all f-statistics residuals are smaller than $Z=3$. This original graph has its worst residual at $Z=24$. With such a poor fit, I think this original graph should not really be used to draw any major conclusions.

- Maier et al. expand the search for graphs by allowing for a much larger number of admixture events (8 or 9, compared to just 3 in the original graph). Doing so, they identify 16 graphs that do fit the data much better, at $Z<4$. With more admixture events, there is a larger number of degrees of freedom, so it’s not surprising that better fits are achieved – this is always expected when increasing the number of admixture events. Maier et al. then find that 13 out of 16 of these graphs do not support the idea that CWC horses do not have any DOM2 admixture. In other words, there are possible models out there that are compatible with CWC horses having some amount of DOM2 ancestry. In the best-fitting graph highlighted by Maier et al., CWC receive 20% DOM2-related ancestry, but there are other possibilities too.

- Maier et al. further state that “We are not aware of other lines of evidence in the paper (apart from the fitted admixture graph) that support the claim of no Yamnaya horse impact on CWC horses.”. I think Librado et al. would disagree with this point, and argue that they had multiple lines of evidence for the claim, not just the admixture graph. In my view, I think Librado 2021 do have other lines of evidence for their claim, and overall rely more strongly on the model-free clustering analyses (Fig 1e in Librado 2021), in which the CWC do not show much if any of the DOM2-related ancestry component.

My overall view would be that the analyses presented by Maier et al. do show that the admixture graph presented in Librado 2021 Fig 3b should not be taken as much, if any, evidence for the claim that CWC horses do not have any DOM2 ancestry. And I think a further takeaway is probably also that admixture graphs are not the right tool to address this question at all. Maier et al. also state that even though they identify graphs that fit the data, they don't necessarily think these correspond to the truth. The space of possible graphs is very large, some well-fitting graphs are consistent with claim while some are not, and it's not clear how to deal with this even on a conceptual level.

Getting back to the current paper then, where the authors revisit the question of DOM2 ancestry in CWC horses. They do this in several ways:

Extended Data 5: A model-free clustering approach based on f_4 -statistics, conceptually similar to PCA. The CWC horses cluster not with DOM2 horses, rather with wild, local European horses. (Though the labelling could be improved – I'm assuming the light blue dots are some wild European horses, but this is not actually indicated. And why are FBPWC and CWC the same colour? Spelling out abbreviations, and possibly some more annotations, would help many readers here). However, I don't think this qualitative analysis could rule out a smaller amount of DOM2 ancestry in the CWC horses. Some readers might even look at this figure and perceive that the CWC horses are slightly shifted towards the orange DOM2 cluster, relative to the light blue horses. Though relative to FBCPOL and FBPWC they are not DOM2-shifted.

Extended Data 6: Another model-free clustering approach. I assume this is basically the same thing as in Librado 2021 Fig 1e. The authors argue, as in Librado 2021, that CWC horses do not show much if any of the DOM2 ancestry profile, but rather look like local European horses. I had difficulties navigating this figure – I'm not sure which horses are CWC horses. Only Ural, Botai and DOM2 clusters are labelled on the side of the figure. Digging through the supplementary table I think Hohler3x3_Ger_m2681 is a CWC horse – it does indeed show very little DOM2 component. Is this the only CWC horse available? Some labelling would help this figure. In any case, as with STRUCTURE/ADMIXTURE clustering, it's difficult to draw very firm conclusions from these kind of exploratory clustering analyses that do not come with a way to assess goodness of fit.

Fig 1a: Direct f_4 -statistics tests, asking if DOM2 and other steppe horses share more with CWC horses or other European horses. Generally, DOM2 share more with other European horses (negative values in this figure), including e.g. horses in Poland, Czechia and Denmark that lived just before the formation of the Corded Ware (ENEOCZE, FBCPOL and FBPWC in the figure). This does seem to argue against a

particular steppe-Corded Ware connection. I suppose one could imagine these results reflecting a pre-domestic cline across Europe such that steppe horses are genetically closer to European horses further east (e.g. Poland) than those further west (e.g. Germany) just due to geography. But otherwise, Funnel Beaker Poland or Denmark seem like pretty good proxies for pre-CWC horses in western/central Europe, and relative to those there does not appear to be a steppe-shift in the Corded Ware horses. This is a much more simple and robust analysis than others presented. Nonetheless, I suppose it's still possible that the unsampled pre-CWC horses in Germany at e.g. 5000BCE were in fact slightly less steppe-shifted than the CWC horses at 3000BCE, but it wouldn't seem very likely to me.

Fig 2a: A LOCATOR analysis attempting to predict the geographical origin of CWC horses, which clearly point to western/central Europe rather than the steppe. However, as far as I understand this is a uni-modal analysis, and will simply point to where the majority of CWC horse ancestry is from. I doubt it's an analysis that would be able to detect let's say 10% or 20% DOM2 ancestry, as the majority ancestry would still dominate the result. As such, presenting this as a main text figure does not seem like the strongest thing that could be done here.

Fig 1b,c: More admixture graphs, using the alternative algorithm AdmixtureBayes. The new graphs identified here differ quite a bit from the one originally published in Librado 2021, which itself demonstrates how tricky and unstable admixture graph analyses are. The authors find that none of the best graphs identified feature DOM2-related gene flow into CWC horses. These results are not thus at odds with those presented by Maier et al. I think it's important to provide some indication on the absolute goodness of fit of the new graphs. I.e. how many outliers statistics $> |Z|$, and what is the worst residual Z-score? If the new graphs are not good absolute fits to the data either, I think that should be clarified in the main text. And if they are not good absolute fits, I would question if they should be presented as main figures and as strong evidence of anything. Extended Data Figure 4 do show the residual Z-scores, and eyeballing this it looks like there are dozens of non-fitting f-stats, and a worst residual of >10 .

The admixture graph re-fit of the Maier et al. graph presented in Extended Data Figure 4, with worst residuals of >10 , is seemingly very different from the fit presented by Maier et al, where the worst residual is 3.4. The legend of Extended Data Figure 4 does specify exactly what is being evaluated here, but based on the supplements I'm guessing this is a refit using the admixturegraph R package using only f4-statistics. It's thus fit in a different way compared to Maier et al., making it yet harder to compare and evaluate. "This revealed the model from Maier and colleagues incompatible with the data (Extended Data Fig. 3B)," – I don't see how EDF3B supports this claim, as EDF3B is about the authors' own new graph. In any case, if Extended Data Figure 4 is to be used as the metric here, then all graphs, including those favoured by the authors, are incompatible with the data.

One specific question on these admixture graphs would be why more proximate wild European horses are not included – e.g. FBPWC or ENEOCZE? The only proxy for pre-domestic European ancestry in the graphs is LPSFR, which is very old at 20kya, and thus not necessarily an optimal proxy. It seems to me that if an origin of CWC horses from populations similar to FBPWC or ENEOCZE is proposed, surely it would be useful to include those populations in the graph?

Given the many complexities of admixture graphs and their fitting, it's hard to evaluate these results. While the authors present some arguments for this in the supplements, it's not clear if the graphs identified here are better fits than those identified by Maier et al. This feels like a potentially endless admixture graph rabbit hole. As such, I don't think presenting these admixture graphs as main text figures is necessarily the strongest way to go, especially if they are actually poor fits to the data in an absolute sense (the authors themselves also seem to express some scepticism about the usefulness of admixture graphs in this context).

Overall, my interpretation of these results is that they demonstrate a largely western/central-European ancestry in CWC horses, but I'm not necessarily fully convinced that there is 0% steppe/DOM2-related ancestry in the CWC horses. I don't think the analyses presented here can firmly rule out minority contributions, e.g. on the order of 5-20%. I suspect the authors are probably right in their conclusion, but I don't currently see the kind of slam-dunk argument that would make this reaffirmation of a previously published result of great interest to a wide audience.

It seems to me, that the best approach to this problem is probably qpWave/qpAdm. Can CWC horses be modelled using only pre-CWC local European sources, with steppe horses placed in the outgroup/reference list? This gets around having to model all the relationships between sources, as in admixture graphs. If using qpWave/qpAdm the authors find e.g. that a single source-FBPWC model fits CWC ancestry, to the exclusion of all steppe horses, they could then still go on to fit a two-source FBPWC+DOM2 model to quantify the maximum bounds of a DOM2 contribution. Perhaps they would find that adding DOM2 does not help the fit, and that the maximum contribution would come out at a few %. Some care would be needed when selecting the reference populations list. This kind of analysis would seem much more robust than admixture graphs to me, and would probably provide the strongest line of evidence.

Actually, the authors do briefly present a qpWave analysis in the supplements. They find that the western/central European populations left=(CWC, FBPWC, ENEOCZE, FBCPOL) are a clade relative to the steppe populations right=(DOM2, TURG, C-PONT, UKR11, NEONCAS). I think this is probably the most

robust evidence against steppe/DOM2 gene flow, in principle. However, the qpWave results are seemingly at odds with the results in Fig 1A. The qpWave results imply that all f_4 -statistics of the form (left,left;right,right) should be consistent with zero. But Fig 1A shows that many of these very f_4 -statistics are quite strongly departing from 0. So in fact, we can conclude that in this case qpWave fails to reject the cladal relationship due to a lack of power. Perhaps due to using too few SNPs? This is an issue with qpWave/qpAdm – if too few SNPs are available, power is low and models are not rejected. Some more care might thus be needed in running and interpreting these analyses – for example excluding very low-coverage samples that reduce the total SNP overlap, and/or inspecting larger sets of f_4 -statistics directly (as in Fig 1B).

Topic 2: This concerns changes in horse life history and genetic structure following domestication, along three separate lines of evidence: a sharp bottleneck followed by a later rapid expansion; the appearance of horses with high inbreeding; and a reduction in average times between generations. These results paint a quite detailed picture of horse domestication, which overall represent by far the most in-depth understanding of the genetics of domestication in any species. The methodology concerning the generation times is also very novel, and potentially quite transformative. However, their novelty also make them a bit more challenging to review.

First, some of the results here concerns the collapse of genetic structure in horses after the spread of the DOM2 lineage, though this is basically what was already reported in Librado 2021. Figure 2B here is basically the same thing as Librado 2021 Extended Data Fig 3d, and conceptually the same results are also expressed in Librado 2021 Fig 2. The sentences presenting the results in Fig 2B here should probably cite Librado 2021 at some point.

Figure 3A presents an inference of effective population sizes (N_e) in the history of domestic horses. The method employed is GONE, which is based on linkage disequilibrium. It's important to note that the curve is extrapolated into the past from domestic horses that lived ~1.9 BCE – it is not based on direct observations in ancient genomes. The authors state there are not enough ancient genomes in the relevant time period to look at this directly. If there still are some genomes in that period, it would still be interesting to see what their diversity levels are like – indeed just a basic heterozygosity-over-time plot would be interesting in general? Because it is indirect, the inference is sensitive to the strengths and weaknesses of the particular inference method employed. GONE is not yet very widely used in population genomics, and at least to me it's not necessarily clear how accurate it is. In general, N_e methods based on linkage disequilibrium have not really taken off in usage. But it's hard to say whether

the curve inferred here is accurate or not. Given methodological caveats, I think it's certainly an interesting result.

The authors use the inferred time of the bottleneck from the GONE curve (~2,726 BCE – surely an overly precise number?) as a constraint on when intensive breeding (and perhaps domestication itself?) started. But as above then, I think it's important to note that this constraint is based on an inference extrapolating into the past, rather than direct ancient DNA evidence. It's also sensitive to the choice of generation time made by the authors (seemingly 8 years here).

The inference of generation times using the recombination clock is quite innovative – I'm not aware of previous work doing this. The idea of using the recombination clock to estimate generation times itself is not entirely novel, it was used by Moorjani et al. 2016 (doi: 10.1073/pnas.1514696113) to estimate generation times in humans – but that was more ad-hoc and dependent on dating Neanderthal admixture in those populations that have it (but perhaps a citation at least in the supplements might be appropriate). The method proposed here is more general, and could potentially be very useful across ancient genomics more broadly. Part of me wants to express some scepticism along the lines of “if this works so well, why hasn't it been done before?”, but that would not really be a fair critique! I am not really able to evaluate the equations in the supplements. The authors show some results on simulated data and there the method performs very well. The results also seem realistic in the sense that there is no other obvious reason why domestic horses would behave differently in these analyses.

The authors also use the mutation clock to independently infer generation times. This is slightly more conventional, but still hasn't really been done in this general, large-scale way on ancient genome before, to my knowledge. As above, I'm a bit surprised it seems to work as well as it does, given the typically low quality of ancient genomes and the challenges of identifying low-frequency mutations. As above, the authors show good performance on simulated data (though simulations will not capture all of the quality issues of ancient DNA data).

An interesting question is how well the recombination and mutation clock estimates correlate with each other. The authors state in the supplements that “both estimators correlate almost linearly (Pearson correlation; $r = 0.996$)” – this is an extremely high correlation, it's difficult to believe that any two estimators applied to noisy ancient DNA data would correlate so strongly. Or am I misunderstanding what this correlation is measuring?

The results implied by the generation time inferences are quite extraordinary and would to my knowledge represent the first time accelerated generation times are observed in early domestic populations across an ancient genomic time series. The results on the Botai horses are particularly interesting, as they contribute to the evidence of domestication, or at least human management, of this otherwise dead-end population. This would be the first time such genetic evidence for management is provided for a population that is not the ancestors of present-day domestic populations.

Author Rebuttals to Initial Comments:

Referees' comments:

Referee #1 (Remarks to the Author):

POINT R1.1. Referee #1 (Remarks to the Author): Librado and colleagues's manuscript titled "Widespread horse-based mobility did not arise in Eurasia before 2,200 BCE" presents another important chapter in the history of horse domestication. The large international team presents new genome data from 86 ancient horses and 93 new radiocarbon dates, which upon integration with published genome data from ancient horses and modern-day domestic breeds accounts for a very powerful time series.

RESPONSE TO R1.1. We thank reviewer #1 for their constructive suggestions. We sincerely believe they helped improve our study, and the resulting manuscript. We encourage reviewer #1 to read the responses to reviewer #2, as many of the points raised in fact overlapped. We explicitly point to relevant responses that we believe could be complementary, which will hopefully facilitate the reviewing experience by highlighting the complementarity of the arguments presented.

POINT R1.2. A general point at the start:

Proper formatting, i.e., line spacing and use of line numbers would have made reviewing a more pleasant experience. I normally tend to provide more detailed feedback on smaller or minor issues, providing line and page numbers, but this is not possible here.

RESPONSE TO R1.2. We apologize for this inconvenience and have included, in our revised manuscript, both line numbers and followed the line spacing recommended as per the 'instructions for authors' of the journal website.

POINT R1.3. The three main aspects covered in the manuscript include a reanalysis of the CWC-associated genomes with respect to a potential involvement in the expansion of steppe pastoralists, the demographic trajectories of DOM2 and Botai horses, and lastly the evidence for controlled reproduction, such as signs of inbreeding and accelerated generation times. The first aspect is addressed as a response to alternative admixture graph modelling results, which were seen as a better fit and used to weave the CWC-associated horses into a narrative in line with the 3rd millennium expansion of steppe pastoralists (Maier et al. 2023). There are intrinsic problems with this scenario.

RESPONSE TO R1.3. We agree that the scenario portraying the steppe migration from ~5,000 years ago as an expansion of riders (and their horses) into Europe faces multiple problems, that we briefly summarize below.

First, we would like to emphasize that, while revising our initial work and using admixtools2 to further model the population history, we have found an admixture graph that is significantly better (i.e. shows a better goodness of fit) than that presented by Maier and colleagues (eLife 2023). Significance was assessed using the very mathematical function provided by those authors, i.e. *qgraph_resample_multi*, which is based on bootstrapping SNP data and contrasting the fit of both competing models to these bootstrap pseudo-replicates ($P < 0.01$). The underlying graph (Extended Fig. 4) supports our original conclusions, including the absence of DOM2 ancestry into CWC horses, in contrast to the models favored by Maier et al. Therefore, following the same line of argument that these authors used in their original work, we conclude that the data do not support a mirrored shift of genetic ancestry in humans and horses, in relation to the ~5,000 years-old steppe expansion.

The difficulty that both Maier et al. and us experienced in fitting a complex population graph reveals in fact severe convergence issues with admixtools2. To discover our significantly better graph, we had to use the graph returned by AdmixtureBayes as a starting input for admixtools2. This graph was not returned by the admixtools2 automated procedure otherwise, despite its improved statistical fit. This demonstrates that Maier et al. converged to a local minimum, despite their massive optimization attempts (see RESPONSE TO R2.3). We believe that it is critical to caution the scientific community that such issues can arise when using this software, as it is designed and promised to become part of the standard toolkit of paleogenomicists. Using this tool as a black box to automatically identify graphs best-fitting genetic data holds the potential for reaching misleading conclusions. This is why we have decided to present the details of our analyses in the Supplemental Information file, despite de-emphasizing the refuting of the model from Maier et al. in our main text. The late expansion of modern domestic horses, originally proved by only single CWC archaeological site from Germany, now emerges with our revisions as a valid observation for the whole region of Central Europe, and the Carpathian and Transylvanian Basins.

There are, in fact, additional issues related to using admixtools2 with three population groups comprised of a pseudohaploid sample only (as done in the original work from Maier et al. 2023). One example of such groups involved a specimen of *Equus somaliensis* as a single donkey outgroup, the only Tarpan specimen ever sequenced in 2021 (our revision now presents a second, older specimen), and NEOANA (they removed a second low-coverage genome from the population group originally defined by Librado et al. (2021)). As acknowledged by the authors themselves, f3 permutations calculated from pseudohaploid target population groups are biased, which implies that any resulting graph fitting such permutations could be wrong. It is therefore surprising that they decided to not follow their own recommendations but extensively discussed instead the implications of their horse graph(s) (see also RESPONSE TO R2.15). Our new data set, now extended to 558 genomes, includes several donkey outgroups, an additional Tarpan and a third NEOANA individual to complete the original population groups with at least two pseudohaploid individuals per group. This served to compute unbiased f3 permutations, which confirmed our population model as best based on the same number of migration events ($m=8$). We note that the model presented in our originally-submitted manuscript was found using AdmixtureBayes (full covariance matrix), and validated using AdmixtureGraph (f4-statistics), instead of using f3-statistics as admixtools2. This made hard, to cross-compare our own results to those presented by Maier et al. as pointed out by reviewer #2 (see RESPONSE TO R2.15), due to the use of different methodologies. Finding now a better graph using their own methodology (admixtools2), as presented in our revisions (Supplementary Information), provides unambiguous rejection of their horse population model.

We also employ, in our revisions, the qpAdm framework to validate the CWC ancestry as a two-way mixture of pre-CWC European horses ($P = 0.109$). The major source was best represented by FBPWC horses from present-day Denmark (67.6%; excavated from a layer dating to ~5,050-4,950 years ago) from present-day Czechia, while the minor source involved ENEOCZE horses from present-day Denmark (32.4%; ~5,230 years-old). Considering that the human steppe ancestry only reached Denmark 4,600-4,800 years ago (see the following BiorXiv preprint doi: 10.1101/2022.05.04.490594), FBPWC horses are also pre-Yamnaya, even if considering their lower temporal boundary of ~4,950 ya. From a geographic standpoint, therefore, CWC horses can be modeled as a mixture of nearby local populations, located North and South of the true CWC sampling location (Hohler); the genetic variation present in their genomes, thus, does not require additional streams of ancestry e.g. from the steppes.

Our qpAdm results appear highly consistent with the results returned by LOCATOR while attempting to predict the geographic origins of CWC horses (these analyses are carried out after splitting the genome of each CWC horse into 10-Mb long windows, and projecting the origin of each window based on a georeferenced panel of 154 ancient horse genomes; see R2.12). All 10-Mb windows of all CWC horses are projected in the vicinity of the location where CWC horses were excavated (Extended Fig. 4C). Combined, all these analyses show that the ancestry of the six CWC horses (and not just one, as in presented in the manuscript previously submitted) are highly unlikely to derive from an expansion from the steppes.

POINT R1.4. First, the underlying assumptions expand on the simplistic equation of pots = people (= languages) = horses, which has already been shown to not be so simple.

RESPONSE TO R1.4. We cannot agree more with the reviewer, and feel that it is still critical to challenge this simplistic and yet recurrent equation, as it is too often the core hypothesis of many studies investigating the human past, including Maier et al. (2023).

POINT R1.5. Second, while the assumption of such a scenario could be deemed legitimate within a hypothesis testing framework, I remain unconvinced that the current dataset is capable of resolving the involvement of the CWC in the increased horse-based mobility that is said to be emerging in the mid-3rd millennium BC. In particular, I wonder how representative the six genomes associated with the CWC horizon really are. The association of humans, their genetic ancestry and material culture is already problematic enough.

RESPONSE TO R1.5. As detailed in RESPONSES TO R1.3 and R1.4, the genetic ancestry of these six CWC horses is not controversial, although we understand that they may not represent the whole CWC horizon, as originating from a single archaeological site. Our attempts to identify additional horse remains associated with the CWC culture and showing DNA preservation levels compatible with genome characterization by shotgun sequencing have failed. We have, therefore, extended our search to e.g. horses from the Carpathian basin, where Trautman et al. (2023, *Sci Adv*) have tentatively identified Yamnaya horse riders. These new data show the persistence of local genomic makeup until ~4,150 years ago, and the replacement by DOM2 horses thereafter (Fig. 1CD). Therefore, all available horse genetic data set reject massive horse mobility before ~4,200 years ago, which is too recent in time to possibly involve the Yamnaya phenomenon. This mirrors the evidence reported in previous work for the absence of a replacement in Asia (some Botai-related specimens persisted at least until ~4,700 years ago at Aleksandrovscoe), and for DOM2 replacement in both Iberia and Anatolia at the transition between the third and second millennia BCE (~4,000 years). Combined, all the evidence available to date highlights that DOM2-like horses were mostly confined onto the Western Eurasian steppe by the end of the 3rd mill BCE, but suddenly emerged in North, South, East and West of that location afterwards.

POINT R1.6. A strong assignment of domesticated animals, which can be traded and exchanged or appropriated to a particular archaeological culture, as done here, would be much more convincing if the data came from more than a single tentatively assigned CWC site. And even then, I would expect to see some variation unless there are in parallel strong indicators of breeding practices, for example as later shown for the DOM2 lineages. In fact, while it is clear from the original analyses in the 2021 manuscript and the re-analyses shown here that the six CWC-associated, or better Final Neolithic genomes show a distinct ancestry profile that is more akin to central/northcentral Late and Final Neolithic genomes, it is the data from Czechia, southern Poland and Hungary (ENEOCZE, FBCPOL, HUNG) that show evidence for early gene flow from the Eneolithic horse lineages of the West Eurasian steppe and forest steppe as indicated by the negative f_4 -statistics in Fig. 1A.

RESPONSE TO R1.6. The figure 1A of the manuscript originally submitted showed less steppe ancestry in CWC horses relative to other local horse groups from Europe. This, however, did not necessarily mean that ENEOCZE (local horse populations from present-day Czechia), FBCPOL (Poland) and HUNG (Hungary) horses received gene flow from Eneolithic horse lineages of the West Eurasian steppe. A negative $f_4(\text{CWC}, \text{Europe}; \text{Steppe}, \text{Donkey})$ statistics could equally indicate introgression from a basal lineage into CWC, for example, from the divergent IBE lineage described by Fages et al. 2019 (*Cell*) (Extended Data. Fig 6A; Response R2.19).

Our new analyses further clarify the population history of European horses. We find support for early gene flow from the Western Eurasian steppe into Eastern Europe (HUNG and ENEOROM), however, not into the only CWC horses available. ENEOROM horses (a ~6,200 years-old lineage from present-day Romania), for example, were already described in Librado et. al 2021 as carrying significant steppe ancestry (~54%), despite being archaeologically identified as

wild. The oldest ENEOROM horse, known as Pie13 (~6,500 years old), was in fact excavated from the Pietrele archaeological site, in which human individuals were also analyzed at the DNA level by Penske and colleagues (Nature 2023). In contrast to horses, Pietrele humans carried no steppe ancestry but retained Neolithic genetic profiles, suggesting that the horse steppe ancestry was already in place before the arrival of human groups originating from the steppes. Likewise, we also detect ~29% steppe ancestry in a horse living 5,300 years ago in eastern Austria (KT46), at the entrance of the Carpathian basin, which predates in time the development of the CWC culture. This proportion is greater than that carried by those Central European horses living nearby (HUNG, present-day Hungary) until 4,200 years ago (~17%). In fact, significant proportions of steppe ancestry (~11.2-19.2%) were already present in ~7,100 years-old horses excavated from present-day Poland (NEOPOL), which is prior to any documented east-to-west human migration from the steppe, but also any archaeological evidence for horse domestication. This demonstrates the natural diffusion of steppe-like ancestry prior to the Yamnaya phenomenon, following the model proposed by Librado et al. (2021), in which neighboring populations mixed locally, creating patterns of isolation by distance and continuous gradients of ancestry. Interestingly, those horses excavated from more recent contexts in Poland showed no steppe ancestry at all (FBCPOL, 5,800-5,500 years-old), in line with a virtually null influence of steppe ancestry further west than eastern Europe during the third millennium BCE. The spread of forest ecosystems in this region could well have represented an effective barrier against horse dispersion from the steppe into Central and Western Europe between 10,000 and 4,000 years ago. Finally, the ancient genomes added during the revisions show that Carpathian horses, which carried a limited fraction of steppe ancestry (~17% in the HUNG lineage) relative to KT46 (~29%), only started to be replaced by DOM2 horses, and their genomic makeup dominated with steppe ancestry, no earlier than ~4,150 years ago.

All these arguments are now developed in their own section, which replaces our original focus on refuting Maier et al. (2023). It reads as follows (lines 324-338):

To investigate this, we mapped the genetic ancestry characteristic of horse populations living across the steppe before the expansion of DOM2 (C-PONT, TURG, and NEONCAS; ~5,616-2,636 BCE). Around ~17.2% of this ancestry was present in the Carpathian Basin during the fourth and third millennia BCE (~3,364-1,971 BCE). However, we find it also in Austria ~3,300 BCE (28.9%, KT46), and in the Transylvanian Basin ~4,200 BCE (54.5%, ENEOROM), at the Pietrele site where the genomic makeup of human populations is inconsistent with steppe contact⁷. In fact, the steppe-related genetic ancestry is found in even earlier horse populations spanning a broad geographic range, including Poland (NEOPOL, ~5,210-5,006 BCE), Anatolia (NEOANA, ~6,396-4,456 BCE), Iberia (IBE, ~5,299-1,900 BCE), and as far back in time as in the Upper Paleolithic of France (LPNFR, ~13,969-12,090 BCE, and; LPSFR, ~21,909-14,646 BCE). This is consistent with the best-fitting population graph showing ENEOROM horses receiving steppe genetic material from an ancestor that also contributed to LPSFR populations (Fig. 2). Therefore, the spread of steppe-related horse genetic ancestry into Europe must predate ~14,646 BCE, which is considerably earlier than any claimed evidence for horse husbandry⁴, and most likely developed in the aftermath of the Last Glacial Maximum (~24,000-17,500 BCE)²².

POINT R1.7. I am wondering whether an AdmixtureBayes model was attempted that included one of the following groups of ENEOCZE, FBCPOL or HUNG in direct comparison to CWC. The data quality for ENEOCZE should suffice. In case there was a well-supported admixture node to ENEOCZE and none to CWC would further strengthen the argument against an introduction via a direct Yamnaya-associated lineage.

RESPONSE TO R1.7. Considering the new genomes generated, we pursued a population model including Eastern and Central European horses, as suggested by the reviewer. Our model now adds ENEOCZE to the population groups previously considered, in addition to NEOPOL, HUNG and ENEOROM from further eastern locations in Europe. We also incorporated NEONCAS, as the most ancient reference for steppe horses in our data set.

The resulting graph is presented in Fig. 2 (copied-pasted below), and fits tightly the allele frequency covariance estimated from the SNP variation (see RESPONSE TO R2.13.) It portrays a population history in which CWC horses carried no DOM2 ancestry, and Tarpan consisted of a mixture of CWC and DOM2 horses, in line with the analysis from Librado et al 2021, and in contrast with the model proposed by Maier et al. (2023). CWC horses, and LPSFR (~17,000-23,600 years-old from southern France; close to Iberia), received introgression from a basal, divergent lineage, possibly related to IBE, as elaborated in RESPONSE TO R1.3 (Extended Data. Fig 6A; R2.19). ENEOROM horses are modelled as descending from a lineage that left the steppe, and also contributed ancestry to LPSFR (Upper Paleolithic) and NEOPOL (~7,100 years ago, from present-day Poland). This supports the existence of an out-of-the steppe migration, which occurred at least 14,646 BCE (the lower ^{14}C boundary for the most recent LPSFR horse). This migration possibly occurred in relation to the Last Glacial Maximum, and before the establishment of woodlands in Europe, much earlier than any evidence for horse domestication and before any herbivore was even domesticated (MacHugh et al. 2017, *Annual Rev Animal Biosciences*).

Figure 2. Most supported population graph summarizing the evolutionary history of pre- and post-

domestication horse lineages. CWC horses do not receive any direct genetic contribution from the steppe. The model is split into 2 panels for clarity. The numbers reported within boxes reflect the admixture contributions from the nodes specified, while those adjacent to arrows indicate the amount of genetic drift leading to individual nodes. Population groups are detailed in Table S1 and colors are according to Fig. 1A.

POINT R1.8. However, more CWC or Bell Beaker-associated horse finds would certainly be favorable in a future study to say anything definite about CWC-associated horses. As a consequence, as the role of the CWC cannot be solved in this manuscript without having sampled more broadly from early 3rd millennium BC cultural groups north of the Carpathian (the authors state so themselves on pg. 4), the problem around CWC horses (or not?) is merely a side issue that should not take away the attention from the main findings by occupied a good third of the paper's word limit.

RESPONSE TO R1.8. Following the reviewer #1 and editor's suggestions, we have drastically reduced the focus on CWC horses to reach out to the most general implications of our findings regarding the domestication history of the horse, especially demonstrating that the late DOM2 replacement of local population groups was not limited to one single CWC site only, but valid across Central Europe, and Carpathian and Transylvanian basins (see Fig. 1CD).

POINT R1.9. Instead, on the basis of the available horse data from the chronological window ranging from ~3500-2500 BC, it could be stressed that critical developments are already underway in southeastern Europe during the late 4th and the early 3rd millennium BC, involving a number of Eneolithic, Final Neolithic and Early Bronze Age groups and a mosaic of horse ancestries.

RESPONSE TO R1.9. Together with the GONE demographic reconstructions (Fig. 3AB), inbreeding patterns (Fig. 3C) and generation time changes (Fig. 4 and Extended Figs. 8-9), all pointing to massive genomic shifts no earlier than ~4,200 years ago), the new admixture graph (Fig. 2) and the late population replacement in the Carpathian basin (~4,150 years ago) reported in the revised manuscript (Fig. 1CD), suggest that the horse steppe ancestry present in Eastern Europe during the late 4th and the early 3rd millennia BCE was decoupled from human migrations and developments within the region (Penske and colleagues, *Nature* 2023). See RESPONSES TO R1.6 and R1.7.

POINT R1.10. Analogously, a strict fixation on Yamnaya and whether a finding is compatible with its timeline is a strong simplification and becoming a hindrance in the recent literature. As far as horizons of genetic transformations, of contact, exchange and/or admixture on the human side are concerned, the Yamnaya-related steppe migrations (pg. 5) are a misnomer or nuisance parameter. The period from ~3300 to ~2800 BC is clearly not the only critical time period during which some human groups, pastoralists in particular, expanded. In fact, this period is bracketed in between time periods that describe the prelude and aftermath in many regions and time periods that are not or no longer linked to Yamnaya (see e.g., Olalde et al. 2018, *Nature*).

RESPONSE TO R1.10. We fully agree. Scholars tend to associate human migrations (especially Yamnaya, but not only) with horses (eg. Heggarty et al. 2023, *Science*), while the blunt truth is that humans already expanded all over the globe before horses were domesticated. Our previous version was over-focused on the Yamnaya narrative, due to our decision to introduce readers to the scientific debate developed in the most recent papers published by Maier et al. and Trautmann et al. In fact, our findings debunk any involvement of horses into large-distance human migrations before 4,200 years ago, Yamnaya and others alike. We believe that the new narrative presented in our revisions will incentivize researchers to seek for alternative mechanistic explanations of past human migrations, including the spread of cultures, goods and diseases. This is an important take-away message from our work, which emphasizes the need to run away from simplistic views about the past. We, in fact, especially discuss the potential incentive underpinning horse domestication. This is, for example, addressed in the sections below (lines 417-431):

Intensified herding practices⁹, growing aridity (the so-called 4.2BP aridification event⁴¹), and/ or over-exploitation of the steppe may have heightened the demand for expanding grazing areas, potentially facilitated by horse-mediated mobility. Domestic horses and spoke-wheeled chariots^{4,39} may also have facilitated the conquest and defense of larger geographic areas in the face of uprising violence and social conflicts^{32,33}.

Our work does not reject the possibility of equestrianism developing in the Pontic steppe or the Carpathian Basin before ~2,200 BCE. However, in such a scenario, the associated breeding practices would not have involved close kin mating or accelerated generation times. The phenomenon would also have remained confined in scale, both demographically and geographically, excluding mobility as the primary domestication incentive. Our research strengthens the case for recognizing Botai as one such locations in the Central Asian steppe where horsemanship developed before large-scale horse-based mobility. There, the domestication process did not aim at global production but remained regional. It is aligned with the expectations of the prey pathway³⁸, in which a settled group of humans developed husbandry through corralling and reproductive control, in the form of shortened generation times, but not close kin mating, to ensure access to an otherwise depleting resource¹².

POINT R1.11. More importantly, the newly presented results on the estimated bottleneck and generation time estimates are way more interesting and convincing, and clearly moving the focus away from the Yamnaya-centered obsession that has manifested itself since the 2015 papers on human genetic transformations. This is a great achievement, but what I am missing is the contextualization of the other players post-dating the Yamnaya time period. For example, if the upper boundary of the DOM2 breeding in the native steppe homeland was set to 2,726 BCE as estimated in the paper, this would put the efforts of creating a strong domestication bottleneck in the hands of people associated with the Catacomb culture and contemporaneous groups. Catacomb-associated individuals are genetically virtually indistinguishable from preceding Yamnaya-associated individuals and are culturally and economically so similar that the naming categories we apply become more of a nuisance than a useful scheme to follow. In this sense, and as cultures usually do not fall from the sky but instead build on each other, as already argued above, I would recommend a less strict adherence to the seemingly monolithic cultural blocks and suggest using a broader interpretation in the light of the developments over the course of the 3rd millennium BCE. For example, the intensification of herding practices (Scott et al. 2022), increasing aridisation and/or overexploitation of the steppe environment might have caused the need to continuously expand the grazing areas, and thus further facilitated the use of domesticated horses to overcome larger geographic distances.

RESPONSE TO R1.11. We agree with this potential interpretation, which we find a fascinating and constructive hypothesis to test in the future. We thank the reviewer for his/her suggestion, and have rephrased the discussion to incorporate it. We also provide more contextual information on the major societal developments of the 3rd millennium BCE, which definitely helps moving the focus away from the Yamnaya obsession. More specifically, the first paragraph of the discussion now reads as (lines 409-421):

This study tackles crucial debates regarding horse domestication with major implications for both horse and human history. It reveals that the horse genomic makeup remained entirely local in Central Europe as well as the Carpathian and Transylvanian Basins until the end of the third millennium BCE. This timeline postdates the human steppe migrations starting ~4,500 BCE⁷, including those potentially spreading proto-IE languages into Europe with the Yamnaya phenomenon ~3,000 BCE. The dramatic spread of DOM2 horses immediately followed the foundation of this new bloodline, and marked a new era of widespread horse-based mobility from ~2,200 BCE. It mirrors the archaeological record, which witnesses a massive spread of horses in the Near East and Asia during the transition between the third and second millennium BCE^{1,39,40}. Intensified herding practices⁹, growing aridity

(the so-called 4.2BP aridification event⁴¹), and/or over-exploitation of the steppe may have heightened the demand for expanding grazing areas, potentially facilitated by horse-mediated mobility. Domestic horses and spoke-wheeled chariots^{4,39} may also have facilitated the conquest and defense of larger geographic areas in the face of uprising violence and social conflicts^{32,33}.

The concluding sentence of one of the paragraphs in the Results section also reads as (lines 351-356):

Restricting analyses to horses from Sintashta contexts, which are associated with the spread of spoke-wheeled chariots in Asia, returns similar demographic profiles and time estimates (~2,100 BCE (2,200-2,075 BCE); Extended Data Fig. 7). These timelines not only coincide with the radiocarbon dating of the earliest DOM2 horses outside the steppe, but also with the earliest horse images in Akkadian art^{30,31}, and with major evidence of conflicts, crises and political disruption, from the Balkans to Egypt and the Indus valley^{32,33}.

POINT R1.12. The in-depth evaluation of effective population sizes, lengths of generations, inbreeding and recombination clocks in DOM2 and Botai horses are highly interesting. Regarding the side-by-side comparison and the points addressed in the discussion, I recommend integrating a few critical statements and references from the SOM part, such as Fig 1C; The breeding of domestic animal can involve the reproduction within close kin, otherwise called inbreeding, as a practice aimed at propagating desirable traits; See also first sentence on pg. 20. This will help the general reader to understand which parameters are critical to distinguish between breeding/selection of specific traits; and management/husbandry as food source of animals.

RESPONSE TO R1.12. We have now calculated inbreeding tracts in cM, and not in Mb (as in the manuscript that was first submitted), and the shift in inbreeding patterns originally reported appears even more clearly (Fig. 3C). Horses that lived prior to 4,200 years ago could be sporadically inbred due to long-term small population sizes yielding to short and less frequently long runs of homozygosity. In contrast, inbred horses that lived after 4,200 years ago, including early DOM2 horses, only carried long runs of homozygosity, indicating mating between close kin, a breeding practice that is avoided by wild animals, and never seen in the previous 40 millennia of horse evolution. Only Przewalski horses, representing an almost-extinct lineage that has been recently recovered from 12-16 captive individuals, show levels of close kin matting comparable to early DOM2 horses. We have clarified our results as suggested by the reviewer. The corresponding Result paragraph now reads as follows (lines 361-366):

The onset of the domestication bottleneck also marks the beginning of DOM2 breeding. Interestingly, the first evidence for horses carrying long runs of homozygosity only (≥ 15 cM), which is indicative of close kin mating, is found in some of the earliest DOM2 sequenced (Fig. 3C), including in the steppes of Central Asia and Anatolia. This indicates that the reproductive control underlying early DOM2 spread involved some levels of inbreeding, which is avoided in the wild but represents a common practice for breeding animals with desirable traits³⁴.

POINT R1.13. Along the same line, a GONE graph in which the effective population sizes of both DOM2 and Botai are shown side-by-side could help to illustrate that some human intervention can lead to same effects (e.g., a drop in N_e),

but that the differences in order of magnitude and other lines of evidences are very important for the interpretation of this signal.

RESPONSE TO R1.13. Figs. 3AB now show the GONE demographic reconstructions side-by-side, for both Sintashta and Botai horses.

POINT R1.14. The true merits of the paper lie clearly in the methodological developments of the latter two aspects, the observations of N_e bottlenecks and acceleration of generation times via mutation and recombination clocks. Both are described and delineated well in the SOM and I applaud the additional validation via coalescent simulations.

RESPONSE TO R1.14. We thank the reviewer for his/her kind words.

POINT R1.15. Additional (minor) points: SOM, pg. 18

- Typo in the equation: permutations; permutations SOM, pg. 18

RESPONSE TO R1.15. Fixed.

POINT R1.16. – The paragraph ‘DOM2 mobility’ on page 18-19 in the SOM file requires cross-references to the figures/tables, where the IBD results and the Mantel test are shown (Figure 2, I suppose?).

RESPONSE TO R1.16. This figure is no longer presented in the revised version of the manuscript, as a result of de-emphasizing the refuting of Maier and colleagues.

POINT R1.17. - SOM, pg. 19 and Fig. 3D: While Botai horses were found the least inbred (Fig. 3D), the three-dimensional perspective is very hard to read. As the z-axis is the crucial factor in this argument, I suggest displaying the Mb windows as separate panels above each other the same x-axis time. That way, the fraction of the genome within each inbreeding category is much easier to compare across the split y-axis (i.e., turn z- to y-axis).Fig. 3D

RESPONSE TO R1.17. We have modified Figure 3D, as suggested for reviewer #1 (it is now shown as Figure 3C).

POINT R1.18. - Furthermore, as this is a critical point in support of one of the main aspects, I suggest highlighting or labelling the “The earliest instances of inbreeding within DOM2 horses are provided by ancient horses associated with Sintashta archaeological contexts, namely UR17x47_Rus_m1856 (0.107), RN03_Rus_m1851 (0.176) and UR17x31_Rus_m1845 (0.122); in the 15Mb+ window.SOM, pg. 20

RESPONSE TO R1.18. All relevant samples cited in the text or SOM are now labeled in Fig 3C. See R1.17.

POINT R1.19. - Considering a realistic transition:transversion rate of 2.2 implies that one every 3.2 mutations actually corresponds to a transversion; Change to “one in every 3.2 mutations corresponds to a transversion.

RESPONSE TO R.1.19. Fixed.

Referee #2 (Remarks to the Author):

POINT R2.1. The authors present a study on horse domestication and early spread using ancient genomes. It represents state-of-the-art in this field of science with a very impressive sampling and data generation effort. The paper is concise and well-written. I think the figure legends could do with some more detail on what the figure actually show. The findings can be summarized into two main topics, which I cover in turn:

1. The spread of the DOM2 ancestry profile associated with modern domestic horses
2. Changes in horse life history and genetic structure as a result of human management.

RESPONSE TO R2.1. We would like to thank reviewer #2 for his/her precious suggestions, as they clearly encouraged us to find simpler approximations to refute Maier and colleagues, and thus to focus the manuscript on our novel discoveries. We encourage reviewer #2 to also read the responses to reviewer #1, as many of the points raised in fact overlapped.

POINT R2.2.

-----Topic 1:

The key message is in the title of the paper and is the claim that the DOM2 horse ancestry profile did not spread widely until after 2200 BC. In particular, key to this claim is that horses associated with the Corded Ware Culture (CWC), who were basically the first western/central-European offshoot of the Yamnaya steppe herders, did not have any DOM2 ancestry. The important implication of the claim is that the DOM2 expansion would post-date the expansion of Yamnaya and the Indo-European language family, which often has been thought to have been horse-driven. This finding was recently reported by the same research group in Librado et al. 2021. Nature “The origins and spread of domestic horses from the Western Eurasian steppes”. The finding itself is thus not a new one to this paper. Librado et al. 2021 wrote e.g.: “Our results reject the commonly held association between horseback riding and the massive expansion of Yamnaya steppe pastoralists into Europe around 3000 BC. “The globalization stage started later, when DOM2 horses dispersed outside their core region, first reaching Anatolia, the lower Danube, Bohemia and Central Asia by approximately 2200 to 2000 BC”. This new paper can thus be seen as following up on, and providing additional evidence for, the 2200 BC claim. But the claim is the same one, and I imagine some readers would view this new paper as in some sense publishing the same finding twice. The motivation to revisit the 2200 BC claim comes primarily from a recent paper challenging that claim.

RESPONSE TO R2.2. We understand and agree that the previous version of the manuscript was too much focused on Yamnaya and CWC horses. We have now shortened this section refuting Maier et al. 2023, also following the editor and reviewer #1 suggestions (see RESPONSES TO E1 and R1.3). Yet, we strongly believe that our study -as a whole- represented more than a mere extension of Librado et al. 2021, as the topic remains extremely debated and even regained considerable interest after Penske and colleagues documented pre-Yamnaya steppe migrations into southeastern Europe (Nature 2023), and after Trautmann et al. 2023 (Science Advances) reported human osteological lesions in skeletons from Yamnaya contexts, and located in the Carpathian basin, that they interpreted as evidence for horse riding. The extended panel of 558 ancient genomes presented in our revisions demonstrates that a minor fraction of steppe ancestry naturally reached eastern Europe prior to any human migration from the steppe, and prior to any evidence for horse domestication

(See R1.6). Those local horses were almost fully replaced by their domestic counterparts from the steppe ~4,150 years ago (Fig. 1CD). This not only rules out that the limited steppe-related ancestry found pre-4,150 years ago is related to human movement, hence, horse riding, but also demonstrates that horse-related mobility only rose after ~4,150 years ago, as per the title of our original study. It also helps us moving focus away from the single CWC site previously analysed, and demonstrates that the timeline for the horse replacement was not limited to present-day Germany (CWC) but extended to Central Europe as well as the Carpathian and Transylvanian Basins, all of which represent critical regions for discussing the diffusion of the Yamnaya phenomenon.

POINT R2.3. - Maier et al. 2023 (<https://elifesciences.org/articles/85492>) revisited the analyses that make use of so called 'admixture graphs' population history models with gene flow - and argued that the finding that CWC horses did not have any DOM2 ancestry is not robust. A ~third of current paper can thus be viewed as a rebuttal to Maier et al. 2023. The critique from Maier et al. 2023 and the response from the authors gets deep into the weeds of admixture graphs, and it's challenging for outsiders to follow all the details here. Nonetheless, here is my attempted summary:

RESPONSE TO R2.3. The rebuttal has been now shortened and simplified to only a couple of sentences. This is perhaps most apparent in the first paragraph following the description of our genome and radiocarbon date data set, which immediately extends the findings regarding CWC horses to Central Europe, and the Carpathian and Transylvanian Basins:

Rebuttal (lines 229-235): However, we predict that the geographic origins of CWC horses is exclusively within Europe (Extended Data Fig. 4C), and identify population graphs that fit extant and new data significantly better (p -value <0.01 ; Supplementary Information). No such graphs support DOM2 genetic contribution into CWC horses (Fig. 2, Extended Data Figs. 3 and 4AB), with the most comprehensive placing CWC horses close to pre-Yamnaya populations from Central Europe

POINT R2.4. - Maier et al. point out that the original graph in Librado 2021 fig 3b actually fits the data very poorly, even though it was the best graph found by the algorithm employed. I think this is an important point; typically admixture graphs are considered to fit; the data if all f -statistics residuals are smaller than $Z=3$. This original graph has its worst residual at $Z=24$. With such a poor fit, I think this original graph should not really be used to draw any major conclusions.

RESPONSE TO R2.4. This remark summarizes quite well the rationale presented Maier and colleagues, but we have to disagree. First, there may be multiple graphs that fit the data better with 8 migration edges, but this does not imply that they are significantly better than a graph including fewer migration events. Second, parameter complexity matters, and using the worst residual (WR) as the sole criterion does not provide a formal test for model selection, not to claim for poor fit. The WR criterion, for example, ignores the overall likelihood of each population model. In fact, the model presented in the original publication from Librado et al. (2021, *Nature*) explained >99% of the genetic variance, and the only high WR involved two populations known to carry ghost ancestry, namely NEOANA and ELEN (incidentally, this was discussed in full transparency both in the main text and the Supplementary Information). All of the remaining population pairs, in contrast, showed reasonable residual values. Given that ghost populations are not, by definition, represented in the set of genomes sequenced, we originally decided to not attempt to fit the data to more complex models, including more migration edges, as this could unavoidably lead to spurious and uncertain topologies. Obviously, Maier and colleagues made different decisions, with the consequence that they obtained best fits for models that contradict basic analyses and are, in fact, not supported (see below).

As an exercise, we can rely on a well-established method for model choice in phylogenetics, namely the Bayesian Information Criterion Criterion (BIC), to contrast two of the models presented by Maier and colleagues (<https://cdn.elifesciences.org/articles/85492/elif-85492-fig3-data7-v2.pdf>). The first assuming $M=7$ migration edges

(Fig. 3h at page 10 of their publication, left panel) vs. the second assuming $M=8$ migration events (Fig 3j, page 13, left panel). The latter corresponds to the graph reported in their main article. The former, however, has a lower BIC ($BIC_7 = 633$ vs. $BIC_8 = 678$), and should be theoretically preferred. It shows no introgression from DOM2 into CWC, which immediately contradicts the model that they choose to discuss the role that horses may have played on the Yamnaya expansion. Their analysis also appears contradictory as they reported better likelihood scores for $M=7$ ($\log_likelihood = -36$) than for the supposedly best model fit ($M=8$, $\log_likelihood = -37$). We would, however, expect $M=8$ to provide a likelihood score at least as good as $M=7$, due to the increased model complexity. These observations, although not discussed in their article, led us to consider that the graph reported in their main article, could represent a suboptimal solution.

As a matter of fact, our revised article now reports an admixtools2 population graph, also based on $m=8$ migrations, that contradicts the model from Maier et al. and fully supports our original conclusions ($|Z\text{-score}| < 1.89$ and $LL = -14$ in the original data set, with unmodified population groups and $diag = 1e^{-8}$). This model is statistically significantly better than that of Maier and colleagues, according to the statistical function that they developed to model fit ($P < 0.01$, *qpgraph_resample_multi*; see RESPONSE TO R1.3). To find this improved graph, we had to use the population model returned by AdmixtureBayes as a starting input for the *find_graphs* function in admixtools2. If admixtools2 was capable to automatically identify the space of models that best fit the data (this is presented by Maier et al. as the most important improvement of admixtools2 vs. previous versions of admixtools), no better models than the ones originally reported by those authors could have been found. This reveals severe optimization problems in admixtools2, which adds to the limitations already described above, including 3 population groups based on a single pseudohaploid sequence that provide biased f_3 -values, and, thus, affect model fit and the resulting graph scores (see RESPONSE TO R1.3). Even more concerning, the original graph presented by Librado et al. (2021), which was built by OrientAGraph (Molloy et al. 2021; Bioinformatics), ended up more similar to the new admixtools2 model presented in our revised manuscript than the one reported by Maier et al., and aligned with the results of the other data analyses presented (e.g. qpAdm modelling, see RESPONSE TO R2.17).

The whole line of argumentation above is developed with details in a new Supplementary Information section, entitled '*Rebuttal to Maier et al. (2023)*'.

POINT R2.5. - Maier et al. expand the search for graphs by allowing for a much larger number of admixture events (8 or 9, compared to just 3 in the original graph). Doing so, they identify 16 graphs that do fit the data much better, at $Z < 4$. With more admixture events, there is a larger number of degrees of freedom, so it's not surprising that better fits are achieved; this is always expected when increasing the number of admixture events. Maier et al. then find that 13 out of 16 of these graphs do not support the idea that CWC horses do not have any DOM2 admixture. In other words, there are possible models out there that are compatible with CWC horses having some amount of DOM2 ancestry. In the best-fitting graph highlighted by Maier et al., CWC receive 20% DOM2-related ancestry, but there are other possibilities too.

RESPONSE TO R2.5. See RESPONSE TO R2.4. With the same number of admixture events ($M=8$), our new admixtools2 graph is significantly better than their best graph, which disqualifies all of their statements. No DOM2 nor steppe ancestry is found within CWC horses, and the Tarpan is indeed modelled as a mixture of CWC and DOM2 horses, as originally presented in the Librado et al. (2021) publication. This is now confirmed with the addition of a second tarpan genome, generated in preparation for the revisions (sample Rat13_Ger_474).

POINT R2.6. - Maier et al. further state that "We are not aware of other lines of evidence in the paper (apart from the fitted admixture graph) that support the claim of no Yamnaya horse impact on CWC horses. I think Librado et al. would disagree with this point, and argue that they had multiple lines of evidence for the claim, not just the admixture graph. In my view, I think Librado 2021 do have other lines of evidence for their claim, and overall rely more strongly on the model-

free clustering analyses (Fig 1e in Librado 2021), in which the CWC do not show much if any of the DOM2-related ancestry component.

RESPONSE TO R2.6. Thanks. We appreciate this fair representation of the analyses that we originally reported.

POINT R2.7. My overall view would be that the analyses presented by Maier et al. do show that the admixture graph presented in Librado 2021 Fig 3b should not be taken as much, if any, evidence for the claim that CWC horses do not have any DOM2 ancestry. And I think a further takeaway is probably also that admixture graphs are not the right tool to address this question at all. Maier et al. also state that even though they identify graphs that fit the data, they don't necessarily think these correspond to the truth. The space of possible graphs is very large, some well-fitting graphs are consistent with claim while some are not, and it's not clear how to deal with this even on a conceptual level.

RESPONSE TO R2.7. We disagree with the statement that Librado 2021 Fig 3b should not be taken as much evidence for the claim that CWC horses do not have any DOM2 ancestry (see R1.3, R2.4 and R2.5), but fully agree with the fact that admixture graphs are not the right tools to address these types of questions; strong patterns of isolation-by-distance create spatial clines of ancestry, as opposed to sharp discontinuity between discrete population demes. This was openly discussed in Librado et al. 2021, for example, in the main text: "*Identifying discrete populations and modelling admixture as single unidirectional pulses, however, was highly challenging given the extent of spatial genetic connectivity.*". Our new analyses, including the Multi-Dimensional Scaling plots shown as Extended Fig. 5ABCD, confirm the presence of natural clines of genetic ancestry across the steppe and the Carpathian and Transylvanian Basins, predating any known human migration from the steppe.

It remains, nonetheless, that some complex admixture graphs are better approximations of the data than others, and some statistical methods to identify these can show better performance than others. AdmixtureBayes, for example, implements a Bayesian MCMC framework that prevents local minima, in contrast to admixtools2. Briefly, MCMC can eventually transit solutions with worst likelihood scores, to find other alternative paths to the optimum solution. The ultimate idea underlying MCMC frameworks is that it is often recommendable to step back, gain perspective, and explore alternative paths to the optimum. Admixtools2 proposes instead a limited number of alternative graphs every iteration, and only the graph with the best score is retained. All other possible models, which may not be the best solution but still represent an improvement relative to the previous iteration, are disregarded. No temporarily step backwards is allowed. This implies that the optimization procedure can get trapped in local optima. AdmixtureBayes can also summarize all graphs forming the posterior distribution, naturally capturing the uncertainty associated with graph inference (Extended Fig. 4B), beyond the best single graph returned (Fig. 2).

POINT R2.8. Getting back to the current paper then, where the authors revisit the question of DOM2 ancestry in CWC horses. They do this in several ways:

1. Extended Data 5: A model-free clustering approach based on f4-statistics, conceptually similar to PCA. The CWC horses cluster not with DOM2 horses, rather with wild, local European horses. (Though the labelling could be improved; I'm assuming the light blue dots are some wild European horses, but this is not actually indicated. And why are FBPWC and CWC the same colour? Spelling out abbreviations, and possible some more annotations, would help many readers here). However, I don't think this qualitative analysis could rule out a smaller amount of DOM2 ancestry in the CWC horses. Some readers might even look at this figure and perceive that the CWC horses are slightly shifted towards the orange DOM2 cluster, relative to the light blue horses. Though relative to FBCPOL and FBPWC they are not DOM2-shifted.

RESPONSE TO R2.8. In the revised version of the manuscript, we have extended our panel of genomes and radiocarbon dates, for an updated total of 558 genomes and 401 dates, respectively. The MDS plots summarizing the genome-wide variation (Extended Figs. 5ABCD), as generated by Struct-f4, places CWC on top of other pre-Yamnaya horses from Europe, especially ENEOCZE (Czechia) and FBPWC (Denmark), relatively close to Upper Paleolithic horses from northern France (LPNFR) and Romania (RONPC06_Rom_m34801). This, and keeping consistency with Librado et. al 2021, which colored FBPWC and CWC the same way as they formed a

monophyletic cluster supported with 100% bootstrap values, justifies the selection of colors. While preparing the revisions, we have, nonetheless, paid attention to improve labeling and coloring as much as possible to enhance the figures readability, both in the main article and the Extended Figures. The new version of the figure presenting MDS plots is copied below:

Data
same
Fig
that
not
any
but
local
sure
are
Only
are

2. **POINT R2.9.** Extended 6: Another model-free clustering approach. I assume this is basically the thing as in Librado 2021 1e. The authors argue, as in Librado 2021, CWC horses do show much if of the DOM2 ancestry profile, rather look like European horses. I had difficulties navigating this figure; I'm not which horses CWC horses. Ural, Botai and DOM2 clusters labelled on the

side of the figure. Digging through the supplementary table I think Hohler3x3_Ger_m2681 is a CWC horse; it does indeed show very little DOM2 component. Is this the only CWC horse available? Some labelling would help this figure.

RESPONSE TO R2.9. We apologize for overlooking this. We have improved the corresponding figure, which now includes the new genomes, and labelled all population groups accordingly, including CWC horses. As a stacked barplot with 558 samples would barely fit within a page size, we have grouped all horses identified as DOM2 in previous work as a single group (labeled DOM2), including also modern breeds. This provides more space for visualizing the genomic makeup of all the newly identified DOM2 horses, and of all pre-DOM2 population groups, including all local groups from Europe such as CWC and more. The corresponding figures are shown now as Fig. 1B and Extended Fig. 6, and are the underlying ancestry plots are copied below:

3. **POINT R2.10.** In any case, as with

STRUCTURE/ADMIXTURE clustering, it's difficult to draw very firm conclusions from these kind of exploratory clustering analyses that do not come with a way to assess goodness of fit.

RESPONSE TO R2.10. We would like to emphasize that this ancestry profiles were computed with Struct-f4. Being based on f4 permutations only, this methodology is devoid of the typical biases of STRUCTURE/ADMIXTURE, such as inbreeding, elevated drift or unbalanced compositions of population groups (same for the Structf4-based MDS plot compared to classical PCA). This is demonstrated in the original publication presenting the methodology (Librado & Orlando 2022, *Bioinformatics*). In robustness, Struct-f4 is comparable to qpAdm, without the need to specify pure donor (left) or reference (right) populations, or apply rotating strategies. It can infer the ancestry profiles of all samples at once, and being implemented within a MCMC framework can provide confidence intervals for each ancestry profile. Because all samples are analyzed jointly (22+ million f4 permutations in a single analysis; see supplementary methods), one can conceive that the amount

of information considered is high and, thus, confidence intervals are small (less than 1% of the best point estimate, also for all CWC horses). This is true regardless of the assumed K (K represents number of ancestral components, meaning the number of split nodes -ancestral populations- in the underlying population tree).

4. **POINT R2.11.** Fig 1a: Direct f4-statistics tests, asking if DOM2 and other steppe horses share more with CWC horses or other European horses. Generally, DOM2 share more with other European horses (negative values in this figure), including e.g. horses in Poland, Czechia and Denmark that lived just before the formation of the Corded Ware (ENEOCZE, FBCPOL and FBPWC in the figure). This does seem to argue against a particular steppe-Corded Ware connection. I suppose one could imagine these results reflecting a pre-domestic cline across Europe such that steppe horses are genetically closer to European horses further east (e.g. Poland) than those further west (e.g. Germany) just due to geography. But otherwise, Funnel Beaker Poland or Denmark seem like pretty good proxies for pre-CWC horses in western/central Europe, and relative to those there does not appear to be a steppe-shift in the Corded Ware horses. This is a much more simple and robust analysis than others presented.

RESPONSE TO R2.11. We agree, this analysis should be simple, but its negative f4 values could mislead readers, as we think happened to Reviewer #1 and Reviewer #2. See RESPONSES TO both R1.6 and R2.19, especially the following section:

“The figure 1A of the manuscript originally submitted showed less steppe ancestry in CWC horses relative to other local horse groups from Europe. This, however, did not necessarily mean that ENEOCZE (local horse populations from present-day Czechia), FBCPOL (Poland) and HUNG (Hungary) horses received gene flow from Eneolithic horse lineages of the West Eurasian steppe. A negative f4(CWC, Europe; Steppe, Donkey) statistics could equally indicate introgression from a basal lineage into CWC, for example, from the divergent IBE lineage described by Fages et al. 2019 (Cell) (Extended Data. Fig 6A, Response R2.19).”

To avoid such a confusion, we replaced the previous figure panel by the Struct-f4 ancestry profiles, and decided to present in the main text the results of a qpAdm rotation strategy, which represents a commonly used method in ancient genomics. In this analysis, we consider amongst the rotated populations all of the pre-CWC European and steppe populations. The best model explains the CWC ancestry as a simple 2-way mixture of Northern pre-CWC horses from Denmark (FBPWC), and Southern pre-CWC horses from Czechia (ENEOCZE), in line with the clines of ancestry found in Europe by that time. All 2-way qpAdm models including DOM2, NEONCAS, CPONT or TURG were rejected, and allowing for a third (neglectable) source population returned at best 1.7% of steppe ancestry into CWC horses (see RESPONSE TO R1.3 again).

5. **POINT R2.12.** Nonetheless, I suppose it's still possible that the unsampled pre-CWC horses in Germany at e.g. 5000BCE were in fact slightly less steppe-shifted than the CWC horses at 3000BCE, but it wouldn't seem very likely to me.

RESPONSE TO R2.12. The ancestry of CWC horses mirrors pre-CWC local populations, and this was proven using the same tool employed by Maier et al. (admixtools2), but also LOCATOR (Extended Fig. 4C) and qpAdm (Table S2). This view is now reinforced with our new best fit graph (Fig. 2), which rejects the presence of any steppe ancestry into the genome of CWC horses. The findings reported in the manuscript originally submitted now extend to a much broader region across Central Europe, and the Carpathian and Transylvanian basins. It appears, thus, considerably less likely that other horse bloodlines could have been missed when designing our sample set.

6. **POINT R2.13.** Fig 2a: A LOCATOR analysis attempting to predict the geographical origin of CWC horses, which clearly point to western/central Europe rather than the steppe. However, as far as I understand this is a uni-modal analysis, and will simply point to where the majority of CWC horse ancestry is from. I doubt it's an analysis that would be able to detect let's say 10% or 20% DOM2 ancestry, as the majority ancestry would still dominate the result. As such, presenting this as a main text figure does not seem like the strongest thing that could be done here.

RESPONSE TO R2.13. Our internal evaluations do indicate that a ~20% of DOM2 ancestry indeed shifts the spatial CWC projection to the East. Additionally, we have now rerun the analysis splitting the genome of CWC horses in 10Mb windows, and run LOCATOR separately for each window, allowing us to not just project the average genome-wide spatial origins of CWC ancestors, but also to scan for the homogeneity of this signal along the genome. The 10Mb-size represents a trade-off between the number of SNPs within each window, and the possibility to project local ancestry tracts. Indeed, a recent introgression from DOM2 into CWC horses should have occurred ~300 years earlier than the date of CWC horses (~4,700 years old), at most. Consequently, large

ancestry tracts should still be detectable in the low-recombining regions of the CWC horse genomes. Our analyses projected all the CWC genomic windows to have originated near the actual sampling CWC location (i.e. the very Hohler archaeological site). This analysis rules out any significant gene flow coming from the steppes as a consequence of the migration related to the Yamnaya phenomenon. This is an important result, because it is model-free and inferred by deep neural networks that can efficiently exploit the geographic gradients of genetic diversity (the patterns of isolation by distance), in a manner that admixture graphs cannot, given their general design based on the a priori definition of population groups. See also RESPONSE TO R2.17.

7. **POINT R2.14.** Fig 1b,c: More admixture graphs, using the alternative algorithm AdmixtureBayes. The new graphs identified here differ quite a bit from the one originally published in Librado 2021, which itself demonstrates how tricky and unstable admixture graph analyses are. The authors find that none of the best graphs identified feature DOM2-related gene flow into CWC horses. These results are not thus at odds with those presented by Maier et al. I think it's important to provide some indication on the absolute goodness of fit of the new graphs. I.e. how many outliers statistics $|Z|$, and what is the worst residual Z-score? If the new graphs are not good absolute fits to the data either, I think that should be clarified in the main text. And if they are not good absolute fits, I would question if they should be presented as main figures and as strong evidence of anything.

RESPONSE TO R2.14. Based on a suggestion from reviewer #1, we have now incorporated additional population groups into the graph (see RESPONSE TO R1.7), including ENEOROM, HUNG, NEOPOL and NEONCAS. AdmixtureBayes inferred a graph with 13 migration edges (Fig. 2). Unfortunately, none of the previous and current implementations of AdmixtureBayes provide model residual values yet. Following the reviewer #2's request, however, we have modified the AdmixtureBayes code (in the *likelihood* function of their posterior.py file) to print and then compare the covariance matrix estimated from the data (variable: emp_cov), to the covariance matrix predicted by the population model with 13 migration edges (par_cov + add). The maximum absolute difference between both matrices is 1.64546766e-07 per SNP, which represents a neglectable allele frequency difference. See also the resulting Figure 2, and RESPONSE TO R1.7 for a description of the graph. See also RESPONSE TO R2.15. for a further description of the AdmixtureBayes method.

POINT R2.15. Extended Data Figure 4 do show the residual Z-scores, and eyeballing this it looks like there are dozens of non-fitting f-stats, and a worst residual of 10. The admixture graph re-fit of the Maier et al. graph presented in Extended Data Figure 4, with worst residuals of 10, is seemingly very different from the fit presented by Maier et al, where the worst residual is 3.4. The legend of Extended Data Figure 4 does specify exactly what is being evaluated here, but based on the supplements I'm guessing this is a refit using the admixturegraph R package using only f4-statistics. It's thus fit in a different way compared to Maier et al., making it yet harder to compare and evaluate. "This revealed the model from Maier and colleagues incompatible with the data (Extended Data Fig. 3B); I don't see how EDF3B supports this claim, as EDF3B is about the authors' own new graph. In any case, if Extended Data Figure 4 is to be used as the metric here, then all graphs, including those favoured by the authors, are incompatible with the data. One specific question on these admixture graphs would be why more proximate wild European horses are not FBPWC or ENEOCZE? The only proxy for pre-domestic European ancestry in the graphs is LPSFR, which is very old at 20kya, and thus not necessarily an optimal proxy. It seems to me that if an origin of CWC horses from populations similar to FBPWC or ENEOCZE is proposed, surely it would be useful to include those populations in the graph?"

RESPONSE TO R2.15. The reviewer is correct: there were multiple f4 permutations with $|Z\text{-scores}|$ of 10. This was not unexpected for AdmixtureBayes, since it calculates standard errors in a different manner than admixtools and admixtools2. Briefly, AdmixtureBayes splits the SNP matrix into X window blocks, and samples with replacement X of these blocks. This is repeated 100 times. Allelic covariance matrices (the equivalent to emp_cov) are then calculated for each of the 100 bootstrap pseudo-replicates. These 100 matrices of allelic covariances are then used to calibrate the optimal number of degrees of freedom (df) for the Wishart distribution, so that the df estimated naturally captures the non-independence between adjacent SNPs (see Nielsen and colleagues 2023 for more details; *PLoS Genetics*). This offers a better statistical strategy than the leave-one-out block jackknifing implemented in admixtools and admixtools2.

What was more surprising to us was to find $|Z\text{-scores}|$ superior to 10 for the graph reported by Maier and colleagues, as the whole fitting process was conducted under the methodological framework of so-called f -statistics (in contrast to what implemented in AdmixtureBayes). It is important to bear in mind that the f4 permutations that we used to evaluate the models in AdmixtureGraph were calculated through qpDstats from admixtools (the original implementation, v7.02), using SNPs covered by at least one pseudohaploid individual in each population (no missingness at the group level).

It is also important to notice that the `find_graphs` function from `admixturetools2` is based on f_3 -statistics (instead of f_4 -statistics), as it comes with a number of important caveats. For example, the f_3 -statistics are biased in the presence of population groups comprised of a single pseudohaploid sequence (see RESPONSE TO R1.3). This was the case for the population groups NEOANA, DONKEY and TARPAN, after they excluded one of the 2 samples forming NEOANA according to Librado et al. (2021). Basically, $f_3(a,b;c) = (a-c)*(b-c)$ can never be negative if c is pseudohaploid (its allele frequencies will be always 0 or 1, thus cannot be intermediate to a or b). This bias in the f_3 calculations can seriously impact graph inference, also in the presence of two or more pseudohaploid individuals per population group (even with larger sampling sizes per group, there will always be some SNP positions that, by chance, are covered by a single pseudohaploid sequence only). This is precisely to prevent this systematic bias that Maier and colleagues developed the `minmac2` option, which removes a SNP from all populations if covered by single individual only, in any of the studied populations. It is anticipable that `minmac2` then filters out most of the SNPs. It results that the `admixturetools2` graph that they retained was at best suboptimal, and most likely biased, and based on a drastically reduced subset of SNPs, which may explain why it was inconsistent with the model fitting f_4 -statistics.

8. **POINT R2.16.** Given the many complexities of admixture graphs and their fitting, it's hard to evaluate these results. While the authors present some arguments for this in the supplements, it's not clear if the graphs identified here are better fits than those identified by Maier et al. This feels like a potentially endless admixture graph rabbit hole.

RESPONSE TO R2.16. We agree: this was indeed highly technical, and we should have made this point clearer. The extensive work underlying the preparation of our revised manuscript now presents more concise and convincing evidence, which allowed us to simplify the rebuttal of Maier and colleagues to not even a full paragraph in the main text. This evidence includes their graph being a suboptimal solution, and `qpAdm` explaining the CWC ancestry as a mixture of local European populations, all pre-Yamnaya (all `qpAdm` models with steppe ancestry, rejected). Beyond simplifying the refuting of Maier et al., we believe that presenting the new admixture graph returned by `AdmixtureBayes` is important (Fig. 2). It is not only consistent with all our analyses (i.e. those from Librado et al. (2021), and those newly presented in our new study), but also uncovers important shifts in the horse ancestry within Central Europe, the Carpathian and Transylvanian basins during time periods that do not overlap the Yamnaya phenomenon, or previous migrations from the steppe (Penske et al. 2023, Nature), in contrast to the claims from Trautman et al. (2023, *Sci Adv*).

9. **POINT R2.17.** As such, I don't think presenting these admixture graphs as main text figures is necessarily the strongest way to go, especially if they are actually poor fits to the data in an absolute sense (the authors themselves also seem to express some scepticism about the usefulness of admixture graphs in this context). Overall, my interpretation of these results is that they demonstrate a largely western/central-European ancestry in CWC horses, but I'm not necessarily fully convinced that there is 0% steppe/DOM2-related ancestry in the CWC horses. I don't think the analyses presented here can firmly rule out minority contributions, e.g. on the order of 5-20%. I suspect the authors are probably right in their conclusion, but I don't currently see the kind of slam-dunk argument that would make this reaffirmation of a previously published result of great interest to a wide audience. It seems to me, that the best approach to this problem is probably `qpWave/qpAdm`.

RESPONSE TO R2.17. We particularly thank the reviewer for suggesting this, which proved instrumental in the process of clarifying our own results and streamlining general arguments, as opposed to only refuting Maier et al.. Our revised work includes additional genome (and radiocarbon dating) data, which increases resolution into the genomic makeup of important pre-DOM2 populations from Anatolia (NEOANA), the Carpathian basin (HUNG), and the western range of the steppes (Poland, NEOPOL). Together with the other local populations from Europe present in our data set (ENEOROM, LPSFR, LPNFR, IBE, ENEOCZE, FBPWC, FBCPOL and KAN22), as well as all from the steppe (NEONCAS, CPONT, TURG and DOM2), these allowed us to set out comprehensive `qpAdm` analyses, not carried out in the version previously submitted.

After rotating all possible 2-way `qpAdm` models, the only one explaining the data involved FBPWC (67.6% ancestry) and ENEOCZE from modern-day Czechia (32.4% ancestry). The age of those two population sources predate human migrations from the steppes. All 2-way models including DOM2, NEONCAS, CPONT or TURG were rejected. It is important to bear in mind that a recent preprint from Prof Willerslev's group has established that the genomic profiles of Danish FBPWC human groups is also local and does not show any steppe influence. The latter only seems to enter the region for the first time from 2,800-2,600 BCE. Therefore, both horses and humans were local in the region at the time period considered, and the `qpAdm` models obtained for explaining the CWC horse genomic makeup only involve the mixing of their neighboring populations, as

expected in situations of isolation-by-distance forming clines of genetic ancestry. In other words, no significant contributions from geographically distant groups from the steppe is necessary to explain the genetic data.

Other qpAdm models become compatible with the data when allowing for a 3rd possible source (this was done as an exploratory exercise since the standard practice in the field is to maintain focus on those simplest models). With a $P = 0.38$, the best 3-way model includes FBCPOL (15.5%), ENEOCZE (25.1%) and FBPWC (59.5%). The following models cannot be rejected either ($0.01 < P < 0.236$), and include steppe populations (NEONCAS, NEOANA, CPONT and TURG), but with really small contributions, probably noise (<1.7%). Either case, ancient horses from pre-CWC Denmark (2,950-3,050 BCE) are the best proxy for CWC horses (~2,700 BCE). Results are reported in Supplementary Table S2.

Combined, out qpAdm results are fully compatible with the spatial projections for the origins of CWC horses returned by Locator, which predict the spatial origins of CWC horses right on their sampling site.

10. **POINT R2.18.** Can CWC horses be modelled using only pre-CWC local European sources, with steppe horses placed in the outgroup/reference list? This gets around having to model all the relationships between sources, as in admixture graphs. If using qpWave/qpAdm the authors find e.g. that a single source-FBPWC model fits CWC ancestry, to the exclusion of all steppe horses, they could then still go on to fit a two-source FBPWC+DOM2 model to quantify the maximum bounds of a DOM2 contribution. Perhaps they would find that adding DOM2 does not help the fit, and that the maximum contribution would come out at a few %.

RESPONSE TO R2.18. Exactly, thanks for the suggestion again. See RESPONSES TO R2.18 and R1.3.

11. **POINT R2.19.** Some care would be needed when selecting the reference populations list. This kind of analysis would seem much more robust than admixture graphs to me, and would probably provide the strongest line of evidence. Actually, the authors do briefly present a qpWave analysis in the supplements. They find that the western/central European populations left=(CWC, FBPWC, ENEOCZE, FBCPOL) are a clade relative to the steppe populations right=(DOM2, TURG, C-PONT, UKR11, NEONCAS). I think this is probably the most robust evidence against steppe/DOM2 gene flow, in principle. However, the qpWave results are seemingly at odds with the results in Fig 1A. The qpWave results imply that all f_4 -statistics of the form (left, left; right, right) should be consistent with zero. But Fig 1A shows that many of these very f_4 -statistics are quite strongly departing from 0. So in fact, we can conclude that in this case qpWave fails to reject the cladal relationship due to a lack of power. Perhaps due to using too few SNPs? This is an issue with qpWave/qpAdm; if too few SNPs are available, power is low and models are not rejected. Some more care might thus be needed in running and interpreting these analyses; for example excluding very low-coverage samples that reduce the total SNP overlap, and/or inspecting larger sets of f_4 -statistics directly (as in Fig 1B).

RESPONSE TO R2.19. Since our previous qpWave analysis was based on ~10 million nucleotide transversions, we can rule out a lack of statistical power. As explained in RESPONSE TO R1.6, the negative f_4 values in Figure 1A reflected introgression from an IBE-related ancestry into CWC, and not necessarily introgression from Eneolithic steppe lineages into non-CWC populations. We understand this could lead to confusion, and we have removed this panel from Figure 1A.

The design of our previous qpWave test ignored any potential influence from Iberia (IBE) into CWC. It simply compared whether Central and Eastern European population were equally related to steppe populations, obviating contributions from other geographical regions. qpWave showed that all European (left) populations were found to be equally related to steppe (right) populations, despite both groups encompassing the Yamnaya temporal horizon. This implied that no additional wave of steppe ancestry reached the most recent European populations, compared to the pre-Yamnaya ones. In our previous version of the manuscript, we also reported a second qpWave analysis, including IBE within the group of right populations, in addition to those from the steppe. qpWave results became then significant, proving that our former result was not due to a lack of statistical power, and that CWC received IBE-related ancestry, compared to other Central and Eastern European populations.

POINT R2.20.

-----Topic 2:

This concerns changes in horse life history and genetic structure following domestication, along three separate lines of evidence: a sharp bottleneck followed by a later rapid expansion; the appearance of horses with high inbreeding; and a reduction in average times between generations. These results paint a quite detailed picture of horse domestication, which

overall represent by far the most in-depth understanding of the genetics of domestication in any species. The methodology concerning the generation times is also very novel, and potentially quite transformative. However, their novelty also make them a bit more challenging to review.

First, some of the results here concerns the collapse of genetic structure in horses after the spread of the DOM2 lineage, though this is basically what was already reported in Librado 2021. Figure 2B here is basically the same thing as Librado 2021 Extended Data Fig 3d, and conceptually the same results are also expressed in Librado 2021 Fig 2. The sentences presenting the results in Fig 2B here should probably cite Librado 2021 at some point.

RESPONSE TO R2.20. We agree, and thus removed Figure 2B to reduce the part of the manuscript previously devoted to rebutting Maier et al.

POINT R2.21. Figure 3A presents an inference of effective population sizes (N_e) in the history of domestic horses. The method employed is GONE, which is based on linkage disequilibrium. It's important to note that the curve is extrapolated into the past from domestic horses that lived ~ 1.9 BCE. It is not based on direct observations in ancient genomes. The authors state there are not enough ancient genomes in the relevant time period to look at this directly. If there still are some genomes in that period, it would still be interesting to see what their diversity levels are like indeed just a basic heterozygosity-over-time plot would be interesting in general? Because it is indirect, the inference is sensitive to the strengths and weaknesses of the particular inference method employed. GONE is not yet very widely used in population genomics, and at least to me it's not necessarily clear how accurate it is.

RESPONSE TO R2.21. Thanks for the constructive suggestion. We, however, think that investigating the nucleotide diversity (π) or heterozygosity from steppe populations over time as a proxy for N_e would be not only patchy, but also potentially misleading. Basically, steppe populations maximize the major component from present-day domestic horses, but include a minority of additional ancestries, as shown in the population models presented and their underlying genetic ancestry profiles (Fig. 1B, Extended Data Fig. 6). TURG and NEONCAS (East of the steppe), for example, include Botai-like ancestry. Such hybridizations result in decoupling the effective from the census population size. Additionally, π or equivalently the heterozygosity (H) is not a good proxy for detecting recent population size changes. Even with a sudden N_e decline, H would decay over time, during many generations. More specifically, using a classical equation derived from the coalescent theory:

$$H_t = H_0 \left(1 - \frac{1}{N_e}\right)^t \approx H_0 e^{-t/N_e}$$

where H_0 is the heterozygosity at time 0, and H_t after t generations of changed N_e population size. It appears that if a population with an initial heterozygosity of 0.002 ($N_{e0} \sim 50,000$) undergoes a sudden, ten-fold decline to $N_t \sim 1,000$ at t generations, then the heterozygosity drop will follow an exponential decay process that will last for hundreds of generations. The heterozygosity, for example, would remain at 0.0018 after 100 generations, and will keep decreasing until reaching a new mutation-drift equilibrium. We would simply not have sufficient temporal resolution, and would miss the timing of the domestication bottleneck.

GONE is a widely used tool in population genomics (121 citations; Santiago et al. 2020 *MBE*), which remains surprisingly under-considered in ancient DNA research, likely due to the inherent inertia of any field to adopt tools developed in other research areas and contexts. The robustness of GONE was yet systematically evaluated by the

developers, under different scenarios, such as selection (Novo et al 2022; *PLoS genetics*), and using experimental data from a laboratory strain of *Drosophila* (Novo et al. 2023; *MER*). Using these experimental data, the authors found that the timing of the population bottlenecks could be accurately inferred, and would show only limited bias in the most unfavorable conditions. Such a bias would result in over-estimating the time period when bottlenecks occurred. Translated to the study of horse demography, this bias would, thus, return a conservative approximation for the timing of the domestication bottleneck, which could be even more recent than what currently inferred. It may be worth noticing that we followed all the developer's recommendations when applying the GONE methodology, including correspondence with the authors by email. They were highly supportive and kind enough to share additional unpublished experiments further proving the accuracy of GONE. The benefit of using GONE is that the approach infers the N_e of the major steppe ancestry, only (colored in orange), and that LD-based reconstructions of the effective population size are known to be highly informative on recent times, in the range of the ~200 generations preceding the population sampled. This time range conveniently intercepts the time of both the Yamnaya and Botai phenomena.

POINT R2.22. In general, N_e methods based on linkage disequilibrium have not really taken off in usage. But it's hard to say whether the curve inferred here is accurate or not. Given methodological caveats, I think it's certainly an interesting result. The authors use the inferred time of the bottleneck from the GONE curve (~2,726 BCE; surely an overly precise number?) as a constraint on when intensive breeding (and perhaps domestication itself?) started. But as above then, I think it's important to note that this constraint is based on an inference extrapolating into the past, rather than direct ancient DNA evidence. It's also sensitive to the choice of generation time made by the authors (seemingly 8 years here).

RESPONSE TO R2.22. Whereas measuring heterozygosity (or nucleotide diversity) from ancient horses could be a direct observation of genetic diversity, it is an indirect and potentially misleading proxy for N_e , as we developed in **RESPONSE TO R2.21**. Time scales are too short and ancient steppe populations were partially admixed with neighboring populations, complicating N_e inference from measures of genetic diversity.

POINT R2.23. The inference of generation times using the recombination clock is quite innovative. I'm not aware of previous work doing this. The idea of using the recombination clock to estimate generation times itself is not entirely novel, it was used by Moorjani et al. 2016 (doi: 10.1073/pnas.1514696113) to estimate generation times in humans, but that was more ad-hoc and dependent on dating Neanderthal admixture in those populations that have it (but perhaps a citation at least in the supplements might be appropriate). The method proposed here is more general, and could potentially be very useful across ancient genomics more broadly. Part of me wants to express some scepticism along the lines of "if this works so well, why hasn't it been done before, but that would not really be a fair critique! I am not really able to evaluate the equations in the supplements. The authors show some results on simulated data and there the method performs very well. The results also seem realistic in the sense that there is no other obvious reason why domestic horses would behave differently in these analyses.

RESPONSE TO R2.23. Thanks! We suspect that a similar method was not developed before because most of ancient DNA research relies on capture data. The targeted SNPs are indirectly partly inherited from the HapMap project, ultimately, where the signal of linkage disequilibrium was simply avoided. One of the logics of the HapMap project was to identify tagSNPs, representative of a whole haplotype block, to obviate linked genomic positions (the signal of the recombination clock was removed, on purpose to avoid redundancy). Likewise, the mutation clock disappears if researchers target always the same SNPs (as de novo variation cannot be identified by design). Other conditionings on basal populations, such as Africans, are key to measure genetic drift, but also overlook the mutation and recombination clocks. The novel methods presented in our study require full genomes, and we think that the resource delivered with this study (558 genomes) may represent some of the most extensive genome time series to date. The data, and the type of questions that they opened in the context of better characterizing the past breeding practices, pushed us to develop a novel methodological framework, which was inspired by the work from Moorjani et al. (2016, *PNAS*).

POINT R2.24. The authors also use the mutation clock to independently infer generation times. This is slightly more conventional, but still hasn't really been done in this general, large-scale way on ancient genome before, to my knowledge. As above, I'm a bit surprised it seems to work as well as it does, given the typically low quality of ancient genomes and the challenges of identifying low-frequency mutations. As above, the authors show good performance on simulated data (though simulations will not capture all of the quality issues of ancient DNA data). An interesting question is how well the recombination and mutation clock estimates correlate with each other. The authors state in the supplements that "both estimators correlate almost linearly (Pearson correlation; $r = 0.996$); this is an extremely high correlation, it's difficult to believe that any two estimators applied to noisy ancient DNA data would correlate so strongly. Or am I misunderstanding what this correlation is measuring? The results implied by the generation time inferences are quite extraordinary, and would to my knowledge represent the first time accelerated generation times are observed in early domestic populations across an ancient genomic time series.

RESPONSE TO R2.24. Since (and including) Librado et al. 2021, we have applied a different strategy to generate genomes and pseudohaploid sequences. We are extremely clean in the laboratory work, as our data rely on the sequencing of tripled-indexed libraries. We carefully check patterns of contamination and post-mortem damage, trimming sequencing reads more stringently if classified as ancient molecules by PMDtools. We do not pseudohaploidize each position of each genome, but only genomic positions identified as nucleotide transversions in the whole genome panel, using the SNP_pval option in ANGSD. The larger the panel for joint calling with ANGSD, the larger the benefit of the SNP_pval option. This option filters out most of the sequencing errors. Substitution types such as C→A (or G→T), enriched amongst nucleotide transversion due to DNA oxidation are for example proportionally much reduced when employing the SNP_pval option. Even though such joint calling comes with a computational cost (e.g. we had to rerun all our analyses while preparing these revisions, as we significantly extended our genome panel), and may result in filtering a fraction of true singletons, the trade-off is positive, and the data are considerably cleaner and more informative than just pseudohaploidizing each position in the genome and identifying SNP afterwards (we do the reverse, i.e. we identify high-quality SNPs first, and then pseudohaploidize them).

Based on the LD decay for a given set of random SNPs, the recombination clock is in turn unaffected by the removal of singletons. Our analyses suggest that the results are consistent, regardless of the MAF filters considered, and the demographic history. The correlation levels measured between both clocks is in fact encouraging, and provides an excellent cross-validation for the results presented.

POINT R2.25. The results on the Botai horses are particularly interesting, as they contribute to the evidence of domestication, or at least human management, of this otherwise dead-end population. This would be the first time such genetic evidence for management is provided for a population that is not the ancestors of present-day domestic populations.

RESPONSE TO R2.25. Thanks, we agree. The implications are that Botai horses were indeed an almost dead-end domestic population. The ongoing debate about whether or not Botai should be considered as a domestication process would also virtually come to an end, once our results will be published.

Reviewer Reports on the First Revision:

Referees' comments:

Referee #1 (Remarks to the Author):

Librado and colleagues present a substantially revised version of their manuscript, now slightly retitled “Widespread horse-based mobility arose around 2,200 BCE in Eurasia”.

I am very impressed by the detailed and highly insightful responses the authors have provided in reply to the reviewers’ points. These not only reflect the careful considerations and efforts that went into the revised manuscript and accompanying analyses, but will provide in its own an extremely useful resource to the community in addition to the revised manuscript and the Supplementary Information files.

In general, I find my previous points well-addressed and the suggested changes implemented accordingly, both in the main text and the figures, and am happy with the resulting revision. The vital parts of the manuscript, the novel findings and methods, are now given sufficient emphasis and space, which in addition to the uniquely powerful dataset for any non-human species merits publication in my opinion. Furthermore, as insights into the evolutionary and domestication history unfolded over the last couple of years, this paper is the most succinct and accessible synthesis thus far.

At this stage, I have only a few (very) minor points to address:

line 173

...”into Europe...”

This might seem pedantic, but the mainstream geographic and political definition of Europe extends to the Ural Mountains and regions north of the Caucasus and the Caspian Sea, and thus includes the regions where both human ‘steppe ancestry’ as well as the DOM2 lineage likely originated, i.e., eastern Europe. As such, the phrasing should be more precise. Given the word limit in the abstract I suggest using “across the rest of Europe” or something similar.

I encourage the authors to thoroughly double-check the remaining parts of the manuscript for the sake of consistency.

lines 186-187

“This chronology implies that the massive steppe migration associated with the spread of the Yamnaya culture...”

I would also phrase this part more carefully. The Yamnaya culture does not spread too far into southeastern and central Europe, remained in the steppe-like lowlands and just reached the interfluvial zone in the Carpathian Basin.

There are two possible options:

- 1) either focus on the genetics, i.e., the spread of steppe-like ancestry across nearly all regions of central and western Europe over the course of the 3rd millennium BCE, or
- 2) acknowledge the main archaeological cultures, complexes, phenomena that facilitated this spread, i.e., Yamnaya, Corded Ware/Single Grave/Battle Axe, and Bell Beaker, followed by locally emerging Bronze Age groups.

Line 245

“This CWC-related ancestry...”

To not fuel the pots=people=horses equation further, and in particular after convincingly having shown that the CWC-associated horses are a central European Neolithic horse lineage (maximizing the brown ancestry component), I wonder whether it would make more sense to decouple the genetic ancestry from the CWC association and use a more neutral term, perhaps NEOCEU.

Even if the owners of the horses were carriers of the Corded Ware culture, it remains an n=1 site scenario in which the six attested horses could have been acquired from a contemporaneous farmer community. Having dug into the archaeological literature a bit myself, I noticed how scarce the evidence for any horses finds is for this time period. Thus, until further evidence from other sites surfaces, I remain unconvinced that carriers of the Corded Ware culture had enough horses to even entertain this scenario.

Lines 256-262

“This timeline is consistent with the first evidence of DOM2 horses outside of the steppe reported by Librado and colleagues (2021) in Moldavia ~2,063 BCE (2,140-1,985 BCE), Anatolia ~2,125 BCE (2,205-2,044 BCE), and Czechia ~2,037 BCE (2,137-1,936 BCE). Yamnaya-related steppe migrations and the spread of DOM2 horses are chronologically incompatible.”

I feel this argument could even be more explicit. I suggesting the following:

“This timeline is consistent with the first evidence of DOM2 horses outside of the steppe reported by Librado and colleagues (2021) in Moldavia ~2,063 BCE (2,140-1,985 BCE), Anatolia ~2,125 BCE (2,205-2,044 BCE), and Czechia ~2,037 BCE (2,137-1,936 BCE), post-dating the arrival of human steppe-related ancestry in the respective regions by 400-600 years (e.g., Mathieson et al. 2018, Papac et al. 2021, Lazaridis et al. 2022, Penske et al. 2023). Yamnaya-related steppe migrations and the spread of DOM2 horses are thus chronologically incompatible.”

I do acknowledge that this point is further emphasized in lines 278-279.

Line 286

“reached Europe...” > “reached Central Europe”, see above.

Line 427-428, legend Figure 4

“...E) Same for Botai horses, which evolved more generations than past and contemporaneous horses from the region,...”

The meaning is unclear to me. Either ‘evolved more generations ago...’ or ‘involved more generations...’?

Lines 552-557, legend Extended Data Figure 9.

“a) Representation showcasing that a younger specimen (left) has greater variance along its genome in the time to the most common recent ancestor (MRCA) of the whole sample. This greater variance is because it experienced more generations of evolution and hence more recombination events than older specimens. This ultimately results in a more even distribution of mutations (stars). This contrast with the older specimen (right), which underwent less recombination events, and thus carries longer haplotype blocks. The genomic positions within these blocks have equal expectations to have mutated or not, leading to an uneven distribution of mutations along its genome.”

The phrasing could be clearer and a little less convoluted, how about this:

a) Schematic representation that illustrates the expectation that the variance along the genome is greater in a younger specimen (left) as the result of more generations of evolution and hence more recombination events than in older specimens with regards to the time to the most common recent ancestor (MRCA) of the whole sample. It is also expected that the distribution of mutations (stars) is less even older specimen (right), which underwent fewer recombination events, and thus carry longer haplotype blocks, in which mutations are equally likely to have occurred or not.”

Lines 566-567, legend Extended Data Figure 10.

Consider adding how correlation was tested and the resulting coefficient r , or alternatively a reference to pg. 29 in the SI.

Referee #2 (Remarks to the Author):

I think the authors have produced an ambitious revision, addressing reviewer comments in great depth. I think the message of the paper is now more strongly supported, and I think the slight shift in emphasis helps the paper overall. The readability of the quite compact paper might be helped by the introduction of a few section headings – if the format allows this, I am not sure, I guess this is also up to the editor.

I have no further comments on the parts of the paper that relate to demography, generation times etc, and I think the authors do a good job defending their analyses in this space. I think these are very technically impressive results that will be quite influential in the field (including the results on the Botai horses).

The one part of the paper that I still find challenging to navigate is the admixture graph results, which are quite complex and so not necessarily straightforward to represent or to digest. I also appreciate that, with less emphasis on this in the main text, the results have to be described very compactly in the main text. In any case, I would still personally argue that the limitations of admixture graphs means that even the new admixture graph results do not allow highly confident conclusions on the question of steppe gene flow into Corded Ware horses, and that the paper would benefit from instead emphasizing other analyses of relevance to this question (in particular the new qpAdm results). But it is ultimately up to the authors how much weight they want to give this particular strand of analysis. In any case, I outline my remaining concerns/thoughts about the admixture graph analyses below. I also find the presentation

of these results confusing at times, and it requires a lot of attention to keep track of what graphs are what, and what exactly the authors refer to at any given point.

The authors push back on the standard approach in the field of evaluating admixture graph fit by looking at the number and Z-score magnitude of residuals (difference between fitted and empirical f-statistics). They instead argue for direct contrasts of likelihood values between competing graphs. I agree such relative contrasts are valuable, but I think the authors are too dismissive of the standard idea of using residuals to assess fit in an absolute sense. For any two graphs, one can always compare them and say one fits better than the other – but it might be that both are terrible fits in an absolute sense, or that they both are fully compatible with the data. I think any graph that fits in an absolute sense has to be taken seriously, even if it's possible that other graphs fit relatively better. At the very least, even if the authors disagree with me on the value of residuals, given they are so standard in the field I think it would be appropriate to at least somewhere document what the residuals are for the graphs presented in the paper (i.e. number of $|Z| > 3$ residuals, and magnitude of worst residuals). The many readers who expect to find this information will then be able to find it.

Working within the framework of 8 admixture events, the authors now identify a new graph that fits significantly better than the Maier graph, according to the likelihood value contrast. This is some valuable progress, but I don't think it completely negates the concerns raised by Maier et al. Maier still showed that there exist alternative graphs that, in an absolute sense (using the residuals, as discussed above), largely explain the data. I don't think Maier et al. necessarily wanted to strongly push that the particular graph they identified is the true one, or even the best-fitting one that can be identified – indeed, their paper states “we are not arguing here that our alternative model is right—indeed we are nearly certain it is wrong in important aspects” - but rather the more technical point that the existence of quite different topologies that are all compatible with the data highlights the limitations of admixture graph inference. The authors still do not really seem to want to acknowledge this technical point, and consequently downgrade the weight they give to this specific line of analysis. They are mainly just focussed on chasing better-fitting graphs. Following the same reasoning, I don't find the authors' discussion in the supplements on how Maier et al. actually identified another graph that might be a slightly better fit (“LL = 36 vs. 37”) particularly illuminating, and slightly beside the point. To me, those two graphs both fit the data quite well, and it's not so important which one has a slightly better fit than the other.

On another point, in my view a valuable finding from Maier et al. was that the first graph published in Librado et al. 2021 (figure 3b in that paper), with only 3 admixture events, was in fact a very bad fit to the data. However, the authors still defend this graph to some extent (in the supplements), with the argument that the parts of the graph that fit poorly are not so relevant. I don't think this is a very viable line of argument. I think it would be fair for the authors to just acknowledge that this was a poorly fitting

graph, and that three admixture events among those 10 populations is simply not enough to model the complexity of horse history. If they still actually want to defend this original graph, at the very least I think they should then apply the same treatment and scrutiny that they apply to the Maier et al graph. E.g. perform the same direct contrast (using `qpgraph_resample_multi`) between this original graph and their new favored 8-admixture-edges graph – if the original graph is significantly worse, I think that should be acknowledged. If it's not worse, that's interesting to report too.

“population graphs that fit extant and new data significantly better ($p\text{-value} < 0.01$ ”. I think it's better to give the exact p-value obtained, rather than just a value it is smaller than (likewise in the supplements where this p-value is referred to). There's quite a big difference between say $p=0.009$ and $p=1e-10$.

The main text Fig 2 displays new results introduced in revision: an admixture graph with a larger number of populations, including more relevant European populations, with 13 admixture events. I think it's a good idea to include these populations of more relevance to Corded Ware horses. But in moving to much larger graphs than previously considered by the authors themselves or by Maier et al, it's tricky to present these results in the context of a rebuttal to Maier et al. The main text does not clearly distinguish between the Maier rebuttal in the strict sense and this new extended analysis, and many readers will likely get the impression that the authors went out and identified this new Fig 2 graph that is simply better than the Maier graph. But one cannot actually say that it's better, because the set of populations used is different. It would be more appropriate, and transparent to the reader, to present this new analysis along the lines of “we expanded the admixture graph search to also include more proximal European horse populations of potential relevance to Corded Ware horse origins, and the best graph identified does not support XXX...”.

Related to the above, the legend of Extended Data Figure 4 states “The drift and admixture estimates are based on our extended dataset, but the underlying topology was found to also fit the original dataset from Librado et al. 2021 better than the graph presented by Maier et al. 2023”. I'm guessing the “underlying topology” here refers the smaller graphs in Extended Data Figures 3 and/or 4 – if so it would be good to refer to those here, so that readers can follow. Also, simply saying “underlying topology” kind of gives the impression that the topology of the smaller graph EXACTLY matches the larger one for the populations that overlap, but I don't think that's quite the case. There's some degree of subjective judgement involved in saying how similar they are. It might be more appropriate to describe this along the lines of e.g. “graphs with a similar core topology but fewer populations were found to also fit ... [reference figures]”.

“predicted a covariance matrix deviating negligibly from the empirical one, according to allele frequency differences of $\sim 1.5e-7$ at most.” This is some kind of absolute assessment of graph fit, which is potentially valuable. But the value has no unit, and I don’t know how to interpret it. It seems like a small number, but I have no frame of reference for it, and most other readers won’t either.

I would encourage (but not require) the authors to send their supplementary section on “Rebuttal to Maier et al. 2023” to the authors of Maier et al. 2023 for comments before publication. It would be understandable to not want to do that, but it would probably benefit the scientific process to give those authors an opportunity to identify anything that they would view as not appropriately representing their work.

The authors introduce new qpAdm analyses, which I think are very valuable and seemingly much stronger than the admixture graph analyses. If I were the authors, I would decrease emphasis on the admixture graphs and increase emphasis on the qpAdm results (including swapping these for main figure attention). But again, it is the authors’ paper to write.

In any case, I think more information on the qpAdm analyses are needed. Which version of qpAdm was used should be indicated, and any relevant parameters used (including allsnps=YES or NO?). I think the table S2 should include the standard errors of the ancestry proportions, which are returned by the software. It seems the authors have only included in the table models that fit, but it is equally (or maybe even more) important to document the models that don’t fit – including the relevant ones referred to in the main text (“All the remaining two-way models were explicitly rejected ($P < 0.01$)”). I think the rejection of qpAdm models involving steppe ancestry is key to the paper, and so it would be good to fully document them in this table and show precisely how strongly they are rejected (that is their exact p-values). There’s really nothing stopping the authors from comprehensively listing models in this table. E.g. for two-source models enumerated from 15 populations, that’s only 105 possible models – all could be listed, and e.g. sorted by p-value. For three-source models it’s a bit more (455 models), but this is what supplementary tables are for arguably. I think the authors could also perhaps think of some graphical representation summarizing these results, i.e. plotting the p-values of different models in some way and highlighting those that have steppe sources vs those that don’t. This should hopefully quite clearly show that the steppe models are very strongly rejected.

“negligible levels of steppe ($<1.7\%$) or DOM2 (0.003)”: missing percentage sign for second number.

“we mapped the genetic ancestry characteristic of horse populations living across the steppe before the expansion of DOM2”. This paragraph does not explain how these results were obtained (f4-struct? qpAdm? Admixture graphs?), and does not refer to any figure or table.

“Therefore, the spread of steppe-related horse genetic ancestry into Europe must predate ~14,646 BCE...”. Here, it will be obvious to many readers that this spread of steppe-related ancestry would just reflect movement among wild horses, but actually stating this explicitly might help some other readers who might otherwise think this is some kind of early human activity.

Slight geographical terminology nit-pick, unrelated to the scientific content of the sentence: “However, we predict that the geographic origins of CWC horses is exclusively within Europe “. The Yamnaya were from eastern Europe, if we take Europe to end at the Ural mountains, as is commonly done. Saying “exclusively within Europe” does thus not exclude the Yamnaya. Perhaps better phrased as “western to central Europe” or something like that.

Another nit-pick on the same sentence: “However, we predict that the geographic origins of CWC horses is exclusively within Europe “. Simply saying “we predict” is not very informative about what was done. What method/result this actually refers to needs to be indicated at least briefly in some way.

“suggest that the last Ice Age may have impacted horse generation times, though to a lesser extent than domestication (Fig. 4A).” The Last Ice Age lasted ~100,000 years, so it’s not clear what the authors are referring to here. Perhaps they mean to say “the end of the last Ice Age” – as there is a slight increase in the curve after 13,000 BCE? Could also be useful to indicate that in the figure, to more clearly connect the statement and the figure, and also all readers might know necessarily know when the Ice Age ended.

Referee #3 (Remarks to the Author):

Disclaimer: The following review focuses on the radiocarbon measurements and its application.

I think the publication is a very nice example of a comprehensive study combining genetic information (of horses) with age relationship obtained mainly obtained from radiocarbon ages on the very same samples in order to obtain a detailed history of (horse) evolution.

The amount of samples measured with ^{14}C is impressive! Also, it seems the quality of measurement is very high, the given errors very low (often 15 - 20 years only!). While there is only one repeated sample (one could wish some more replication for such an important dataset), they seem to be consistent, as can be seen on same context samples.

Minor critics:

No information on how the radiocarbon ages were obtained are given. I suggest writing a short paragraph for the supplementary information. How were the samples treated? Can you give references?

While most samples are independently dated and can be treated in a consistent way, there are some samples, where I think no independent measurements are available. I think a clear differentiation would be good to have (see also comments below). The authors seem to work primarily with mean ages for individual samples instead of the age range obtained from the calibrated age ranges (for simplicity). This is not always considered the right procedure and can be problematic, when high temporal resolution is interpreted. Having said that, I think the publication focuses primarily on the interpretation based on medium to low time resolution and thus the presented interpretation is valid based on the age information given.

In the following are some comments on the listed radiocarbon measurements.

Your own measurements:

- For sample GVA9046_Mon_716 no Lab number was given (what is standard and strongly recommended for ^{14}C measurements). By the way, you call the column in the excel sheet "UCIAMS", what is a lab prefix - however, you have also listed numbers from other labs. The supplementary information only indicates that the date may come from ref. 91 (that I could not access). Is the date directly based on bone dating? I suggest giving a reference also in the excel list with together with the dating (one might have the impression that you actually dated the sample, what I think is not the case).

- I only counted 122 radiocarbon measurements in the excel list, while you write in the text there were 149 new measurements performed.

- There are five samples from Denmark (GIN1020_Den_m3050, GIN1055_Den_m3050, GIN396_Den_m3225, GIN489_Den_m3225, GIN561_Den_m3225), where no radiocarbon measurement was given, but still age limits (Min / Max) were given. Your supplementary information gives a citation, where bones were dated and it looks like the age limits come from. Also here I suggest giving a reference in the excel together with the dates. Maybe even one or the other bone was directly dated - then you could also give the radiocarbon age? (The advantage of a radiocarbon age is, that it can be re-calibrated when a new calibration curve appears). What is maybe most important here is, that the five samples are not independently dated (with some possible implication on modelling).

Measurements of others:

Some similar critics as for your own samples are given here.

- For a relatively large number of samples, there is a calendar age given, but no ¹⁴C measurement (for samples from Librado et al. 2021).

- There are also samples with ¹⁴C dates, but with no Lab number given

I have to admit, the information is also missing in the original publications, but it would be good if more information could be collected here for completeness. I think it can be crucial to know, which samples were dated independently (and are they all radiocarbon dated?).

Author Rebuttals to First Revision:

Referees's comments:

Referee #1 (Remarks to the Author):

Librado and colleagues present a substantially revised version of their manuscript, now slightly retitled “Widespread horse-based mobility arose around 2,200 BCE in Eurasia”.

I am very impressed by the detailed and highly insightful responses the authors have provided in reply to the reviewers' points. These not only reflect the careful considerations and efforts that went into the revised manuscript and accompanying analyses, but will provide in its own an extremely useful resource to the community in addition to the revised manuscript and the Supplementary Information files.

POINT 20: In general, I find my previous points well-addressed and the suggested changes implemented accordingly, both in the main text and the figures, and am happy with the resulting revision. The vital parts of the manuscript, the novel findings and methods, are now given sufficient emphasis and space, which in addition to the uniquely powerful dataset for any non-human species merits publication in my opinion. Furthermore, as insights into the evolutionary and domestication history unfolded over the last couple of years, this paper is the most succinct and accessible synthesis thus far.

RESPONSE 20: We thank the reviewer for his/her kind words.

At this stage, I have only a few (very) minor points to address:

POINT 21: line 173 “... into Europe...” This might seem pedantic, but the mainstream geographic and political definition of Europe extends to the Ural Mountains and regions north of the Caucasus and the Caspian Sea, and thus includes the regions where both human “steppe ancestry” as well as the DOM2 lineage likely originated, i.e., eastern Europe. As such, the phrasing should be more precise. Given the word limit in the abstract I suggest using “across the rest of Europe” or something similar. I encourage the authors to thoroughly double-check the remaining parts of the manuscript for the sake of consistency.

RESPONSE 21: We agree, and have edited the abstract (“across Europe” instead of “into Europe”), and double-checked the remaining sections of the main text as suggested.

POINT 22: lines 186-187

“This chronology implies that the massive steppe migration associated with the spread of the Yamnaya culture...” I would also phrase this part more carefully. The Yamnaya culture does not spread too far into southeastern and central Europe, remained in the steppe-like lowlands and just reached the interfluvial zone in the Carpathian Basin. There are two possible options:

1) either focus on the genetics, i.e., the spread of steppe-like ancestry across nearly all regions of central and western Europe over the course of the 3rd millennium BCE, or
2) acknowledge the main archaeological cultures, complexes, phenomena that facilitated this spread, i.e., Yamnaya, Corded Ware/Single Grave/Battle Axe, and Bell Beaker, followed by locally emerging Bronze Age groups.

RESPONSE 22: We agree with the reviewer, and have opted for his/her first suggestion. This results in the following sentence in the main text (lines 192-194, page 3):

“This chronology implies that the spread of steppe-like ancestry that reshaped the human genetic landscape of nearly all regions of central and western Europe over the course of the 3rd millennium BCE^{8,9} was not driven by DOM2 horseback riding.”

POINT 23: Line 245

“This CWC-related ancestry...” To not fuel the pots=people=horses equation further, and in particular after convincingly having shown that the CWC-associated horses are a central European Neolithic horse lineage (maximizing the brown ancestry component), I wonder whether it would make more sense to decouple the genetic ancestry from the CWC association and use a more

neutral term, perhaps NEOCEU. Even if the owners of the horses were carriers of the Corded Ware culture, it remains an n=1 site scenario in which the six attested horses could have been acquired from a contemporaneous farmer community. Having dug into the archaeological literature a bit myself, I noticed how scarce the evidence for any horses finds is for this time period. Thus, until further evidence from other sites surfaces, I remain unconvinced that carriers of the Corded Ware culture had enough horses to even entertain this scenario.

RESPONSE 23: For the sake of consistency, we do not think that changing labels would be appropriate since those horses were referred to as ‘CWC’ in the original Librado et al. (2021) publication, as well as the Maier et al. (2023) publication criticizing the validity of the population graph models originally presented. We, however, agree with the reviewer that CWC horses are scarce in the archaeological record, and some future revisions of labels may be beneficial, especially once a more extensive genome mapping of ancient horses from Central Europe will be available.

Lines 256-262

“This timeline is consistent with the first evidence of DOM2 horses outside of the steppe reported by Librado and colleagues (2021) in Moldavia ~2,063 BCE (2,140-1,985 BCE), Anatolia ~2,125 BCE (2,205-2,044 BCE), and Czechia ~2,037 BCE (2,137-1,936 BCE). Yamnaya-related steppe migrations and the spread of DOM2 horses are chronologically incompatible.”

I feel this argument could even be more explicit. I suggesting the following: “This timeline is consistent with the first evidence of DOM2 horses outside of the steppe reported by Librado and colleagues (2021) in Moldavia ~2,063 BCE (2,140-1,985 BCE), Anatolia ~2,125 BCE (2,205-2,044 BCE), and Czechia ~2,037 BCE (2,137-1,936 BCE), post-dating the arrival of human steppe-related ancestry in the respective regions by 400-600 years (e.g., Mathieson et al. 2018, Papac et al. 2021, Lazaridis et al. 2022, Penske et al. 2023). Yamnaya-related steppe migrations and the spread of DOM2 horses are thus chronologically incompatible.”

I do acknowledge that this point is further emphasized in lines 278-279.

RESPONSE 23: We agree with the reviewer’s edit. It is now found on lines 275-279, page 5.

POINT 24: Line 286

“reached Europe”... “reached Central Europe”, see above.

RESPONSE 24: Agreed.

POINT 25: Line 427-428, legend Figure 4

“Same for Botai horses, which evolved more generations than past and contemporaneous horses from the region,..” The meaning is unclear to me. Either “evolved more generations ago”; or “involved more generations...”

RESPONSE 25: We have opted for the second suggestion: “involved more generations”. Thanks for the suggestion.

POINT 26: Lines 552-557, legend Extended Data Figure 9.

-“a) Representation showcasing that a younger specimen (left) has greater variance along its genome in the time to the most common recent ancestor (MRCAs) of the whole sample. This greater variance is because it experienced more generations of evolution and hence more recombination events than older specimens. This ultimately results in a more even distribution of mutations (stars). This contrast with the older specimen (right), which underwent less recombination events, and thus carries longer haplotype blocks. The genomic positions within these blocks have equal expectations to have mutated or not, leading to an uneven distribution of mutations along its genome.”

The phrasing could be clearer and a little less convoluted, how about this:

a) Schematic representation that illustrates the expectation that the variance along the genome is greater in a younger specimen (left) as the result of more generations of evolution and hence more recombination

events than in older specimens with regards to the time to the most common recent ancestor (MRCA) of the whole sample. It is also expected that the distribution of mutations (stars) is less even in the older specimen (right), which underwent fewer recombination events, and thus carry longer haplotype blocks, in which mutations are equally likely to have occurred or not.”

RESPONSE 26: We agree and follow the reviewer’s suggestion (lines 910-919, page 16).

POINT 27: Lines 566-567, legend Extended Data Figure 10.

Consider adding how correlation was tested and the resulting coefficient r , or alternatively a reference to pg. 29 in the SI.

RESPONSE 27: We apologize for overlooking this. The underlying statistical test was added to the revised captions of Extended Data Fig. 10: “(Pearson correlation; $r = 0.999$; p -value $< 2.2e-16$)”.

Referee #2 (Remarks to the Author):

POINT 28: I think the authors have produced an ambitious revision, addressing reviewer comments in great depth. I think the message of the paper is now more strongly supported, and I think the slight shift in emphasis helps the paper overall. The readability of the quite compact paper might be helped by the introduction of a few section headings; if the format allows this, I am not sure, I guess this is also up to the editor.

RESPONSE 28: We have added the following section headings:

“*Datasets and Experimental design*” (line 224, page 4)

“*Spread of DOM2 horses across Europe*” (line 240, page 4)

“*DOM2 demographic history*” (line 300, page 5)

“*DOM2 generation times shortened 2,200 BCE*” (line 326, page 6)

“*New evidence of horse husbandry at Botai*” (line 354, page 6)

POINT 29:

I have no further comments on the parts of the paper that relate to demography, generation times etc, and I think the authors do a good job defending their analyses in this space. I think these are very technically impressive results that will be quite influential in the field (including the results on the Botai horses).

RESPONSE 29: We thank the reviewer for supporting and acknowledging the importance of our work.

POINT 30:

The one part of the paper that I still find challenging to navigate is the admixture graph results, which are quite complex and so not necessarily straightforward to represent or to digest. I also appreciate that, with less emphasis on this in the main text, the results have to be described very compactly in the main text. In any case, I would still personally argue that the limitations of admixture graphs means that even the new admixture graph results do not allow highly confident conclusions on the question of steppe gene flow into Corded Ware horses, and that the paper would benefit from instead emphasizing other analyses of relevance to this question (in particular the new qpAdm results). But it is ultimately up the authors how much weight they want to give this particular strand of analysis.

RESPONSE 30: We thank the reviewer for acknowledging the validity of our new qpAdm analyses, and for suggesting those analyses in the first place.

Following the editorial advice to focus on our new consistent findings, rather than on a rebuttal to Maier et. al, we have also reduced the space dedicated in the main text to admixture graphs, and nuanced the corresponding section in the Supplementary Information. Now that the paper has been restructured accordingly, a long section presenting admixture graphs is not needed anymore (and the original Figure 2 presenting our best admixture graph is now provided as Extended Figure 3). The following response is not developed in our revised manuscript, but only here to respond to the points raised by reviewer #2 in respect of the quality of the scientific discussion that s/he developed, and as an exercise of full transparency.

We believe that the discrepancies between our population graphs and that of Maier and colleagues can be tracked to changes introduced in data treatment. Maier et al., indeed, modified the block size for leave-one-out jackknifing, used 7.4M nucleotide transversions instead of the 7.9M originally included in our dataset, and also altered the original population group configurations. For example, the removal of a NEOANA and a DONKEY individual left these two population groups with a single pseudohaploid representative. Together with TARP, which encompassed the only Tarpan genome sequenced at the time, this resulted in the presence of 3 population groups represented by only one single pseudohaploid genome each, out of the 10 population groups considered.

As elaborated in our previous round of responses, and as originally acknowledged by Maier and colleagues, f3 permutations involving any of these three pseudohaploid populations are biased. Additionally, the configuration of one of the remaining 7 population groups (CPONT) was also altered in their work, as two genomes from this group (UR17x26_Rus_m2751 and BZNK1002x4_Rus_m3450) were removed, while another genome not belonging to this group was included (LR18x12_Rus_m2995). This choice was probably made given the spatio-temporal information associated with LR18x12_Rus_m2995;

it, however, dismissed the fact that this individual showed a genomic makeup different from the other CPONT individual (hence, why this specimen was originally labeled ‘RepinAdmix’ in Librado et al. (2021)).

While the CPONT group from Maier et al. (2023) included two samples, namely LR18x12_Rus_m2995 and LR18x09_Rus_m3089, it was almost pseudohaploid because LR18x12_Rus_m2995 has a missingness rate of $\sim 70\%$. This could extend the bias introduced in the calculation of f_3 -statistics described above to all permutations including CPONT. The new data presented in our manuscript allow us to overcome such limitations as all population groups considered, including DONK, NEOANA, TARP and CPONT, are represented by more than one individual genome.

As a result, our extended dataset allows us to employ the $\text{minac2} = 1$ option without biasing f_3 -calculations, as this option filters out SNPs from the input matrix if covered by a single pseudohaploid individual in a population. Nonetheless, using this option leaves sufficient numbers of transversion SNPs for our calculations ($>1.5\text{M}$). This contrasts with the analyses conducted by Maier and colleagues, mostly assuming $\text{minac2} = 2$, an option flagged as “experimental” in the Admixtools2 documentation (Maier et al. 2023). This option filters sites from the input matrix if covered by a single pseudohaploid individual in population groups composed of two or more individuals. Considering the CPONT definition that Maier et al. used, turning on the $\text{minac2} = 2$ option resulted in the exclusion of most SNPs (nearly 72%), due to the high missingness rate of LR18x12_Rus_m2995.

Applying Admixtools2 to the extended dataset presented in our new study, we now recover an even more refined population graph (assuming $M = 8$ migrations) than both, the one we originally presented in Librado et al. (2021) and the ones presented by Maier et al. (2023). This new graph (see below) fully supports our original conclusions regarding the origins of CWC and Tarpan horses (score = 8.81 and $|WR| = 1.93$).

Moreover, we compared the fit of this new refined population graph to that presented in the main article from Maier and colleagues, using three possible input datasets, named ‘Original’ (from Librado et al. (2021), i.e. containing the TARP as the only pseudohaploid population group), ‘Reduced’ (from Maier et al. (2023), i.e. 3 pseudohaploid population groups), and ‘Extended’ datasets (this study, i.e. 0 pseudohaploid population groups). The analysis was run considering minac2 values of 0, 1 and 2 to assess their possible consequences. We also considered two sets of values for the diag and diag-f3 parameters, whose importance is often overlooked and relevant for the discussion about Worst Residual (WR) values (see below, RESPONSE 35). More specifically, the diag and diag-f3 values are simply small quantities added to the diagonal of the covariance and f3 matrices, both in AdmixTools and AdmixTools2. Their main purpose is to avoid these matrices to be singular, ensuring their invertibility for easy mathematical estimation of the model parameters. These diag and diag-f3 parameters have no other meaning than that, and lack any inherent biological definition. The default values for diag and diag-f3 are totally arbitrary, and any other smaller quantity may accomplish the same statistical purpose (Lipson et al. 2020). The following table presents the results of those comprehensive analyses, which helps understand the discrepancy between our results and those presented by Maier and colleagues. Two important conclusions can be drawn from this comparison:

- (i) the Score and WR of any model are extremely dependent on the choice of diag and diag_f3 values. For example, considering diag and diag_f3 values of 10^{-8} for the ‘Extended’ (this study) dataset provides a Score of ~ 8.81 , while diag and diag_f3 values of 10^{-4} returns a Score of ~ 29.8 with the same data. WR values also change accordingly from ~ 1.9 to ~ 4.9 , the latter of which is considered as indicative of poor fitting, according to the reviewer. This example, and more generally this table, highlight the score and WR as poor and arbitrary indicators of absolute fit, unless diag and diag_f3 are sufficiently reduced to minimally impact the covariance and f3 matrices, while ensuring that these matrices are still invertible;
- (ii) the new refined population graph is almost systematically and significantly better than that from Maier and colleagues, in terms of score and WR, as assessed by qqgraph_resample_multi (p -value $< 1e-4$ for all contrasts, except for the Extended dataset, where p -value $< 1e-5$; see RESPONSE 36). The only exception pertains to minac2 = 2, a scenario in which both graphs fit the data comparably, for both the Reduced and Original datasets. Bootstrap resampling, indeed, indicates that their difference in fit is not statistically significant (p -value > 0.05).

10-pop data	diag & diag_f3	Minac2	Score (Maier et al. 2023)	WR (Maier et al. 2023)	Score (this study)	WR (this study)	P-value (our model fits better if $p < 0.05$)
Reduced	1.00E-04	0	139.5652	7.954433	107.726	8.370772	0.076
Original	1.00E-04	0	233.0443	13.64316	124.0013	8.95483	0
Extended	1.00E-04	0	217.4905	13.11261	80.46593	7.255115	0
Reduced	1.00E-04	1	NA	NA	NA	NA	NA
Original	1.00E-04	1	NA	NA	NA	NA	NA
Extended	1.00E-04	1	96.25536	5.796156	29.83058	4.928046	0
Reduced	1.00E-04	2	45.2638	3.775434	59.73442	4.374429	0.843
Original	1.00E-04	2	64.09915	5.625056	42.29141	3.916438	0.105
Extended	1.00E-04	2	96.57098	5.761589	29.83088	4.929437	0
Reduced	1.00E-08	0	157.1419	7.698408	138.1967	7.627356	0.27
Original	1.00E-08	0	320.4031	13.40151	152.1653	7.281971	0
Extended	1.00E-08	0	303.5723	13.11261	47.67331	3.392434	0
Reduced	1.00E-08	1	NA	NA	NA	NA	NA
Original	1.00E-08	1	NA	NA	NA	NA	NA
Extended	1.00E-08	1	106.6313	6.067639	8.810991	1.933206	0

Reduced	1.00E-08	2	51.35297	3.775434	69.71367	3.775434	0.869
Original	1.00E-08	2	61.71092	5.121983	38.53382	4.120076	0.117
Extended	1.00E-08	2	112.8742	5.91419	8.813957	1.934211	0

Note: NA indicates that no SNPs were retained after applying $\text{minac2}=1$ in the Original and Reduced datasets, as expected (both datasets include at least one pseudohaploid population group). Those population graphs receiving highest support are reported in red, i.e. as associated with minimal Score values.

To further understand why the population graph presented by Maier et al. (2023) received equal statistical support to ours when applied to the ‘Reduced’ dataset with $\text{minac2} = 2$, we calculated the permutations of f_3 -statistics amongst the 10 population groups, and compared them against those obtained considering $\text{minac2} = 1$ and the ‘Extended’ dataset (i.e. the best-supported population graph, refined for this study). The latter prevents from the $\text{minac2} = 2$ biases described above, and, thus, provides the best approximation of f_3 genetic distances. The following plot presents the different values calculated for the same f_3 -statistics permutations under both situations. If such calculations were unbiased, f_3 -statistics would be expected to be linearly correlated along the main diagonal.

While a majority of f_3 -statistics project linearly along the main diagonal (black dots), a number of permutations emerge as significantly under-estimated (dark blue dots), or over-estimated (turquoise, magenta and brown dots) when considering the ‘Reduced’ dataset with $\text{minac2} = 2$. Under-estimated values (dark blue dots) pertain to f_3 permutations in which ELEN is used as a target population (this population was also modified by Maier and colleagues, who kept only ELEN individuals beyond the limits of radiocarbon dating, hence changing the temporal structure within this group). Over-estimated values correspond to f_3 permutations in which NEOANA (turquoise), CPONT (brown) and TARP (magenta) are used as target populations, all of which represent groups where the choice of the minac2 value is predicted to have an impact, due to the presence of only one pseudohaploid genome (or almost one due to

extensive missingness in the case of CPONT). This analysis clearly indicates that the values of f_3 -statistics used for population graph reconstruction by Maier et al. (2023) were strongly biased. Therefore, the best population graph from Maier et al. did not necessarily represent the true underlying population history.

We conclude that the population graph presented in our study provides a better depiction of the horse evolutionary history than that of Maier et al. (2023). We save those arguments, and those presented in previous rounds of reviews, for a rebuttal paper that will be prepared at a later stage, as suggested by the editor. We shall, thus, insist that the new revised version was restructured to present only our new data and findings, as opposed to confront the technical issues resulting from the work of Maier et al. (2023).

POINT 31:

In any case, I outline my remaining concerns/thoughts about the admixture graph analyses below. I also find the presentation of these results confusing at times, and it requires a lot of attention to keep track of what graphs are what, and what exactly the authors refer to at any given point.

The authors push back on the standard approach in the field of evaluating admixture graph fit by looking at the number and Z -score magnitude of residuals (difference between fitted and empirical f -statistics). They instead argue for direct contrasts of likelihood values between competing graphs. I agree such relative contrasts are valuable, but I think the authors are too dismissive of the standard idea of using residuals to assess fit in an absolute sense. For any two graphs, one can always compare them and say one fits better than the other; but it might be that both are terrible fits in an absolute sense, or that they both are fully compatible with the data.

RESPONSE 31: We have simplified and more clearly presented our admixture graph results. With this spirit, we want to emphasize that we do not ‘push back’ from the standard approach to evaluate admixture graphs, but simply make clear that the criterion based on the worst $|Z$ -scores| does not represent a formal test for selecting the most appropriate model complexity, because: (i) it obviates the overall likelihood; (ii) it does not explicitly penalize an increasing number of parameters, and; (iii) it is sensitive to the choice of diag and diag_{f_3} values (see RESPONSE 30).

Model selection must account for model complexity when assessing statistical fit, in line with well-established practices in the field of statistical inference. One such example is provided by the software `qpAdm` and `qpWave`, both included within `AdmixTools` and `AdmixTools2`. There, simpler submodels (i.e. including fewer population sources) are automatically compared to assess whether they can explain the data no worse than the complex one (through a Chi-square test accounting for the difference in model parameters). Other than this, we fully agree that Z -scores can still be useful to evaluate graphs, in particular to identify regions of the graph fitting poorly the observed permutations of the f -statistics.

POINT 32: I think any graph that fits in an absolute sense has to be taken seriously, even if it’s possible that other graphs fit relatively better. At the very least, even if the authors disagree with me on the value of residuals, given they are so standard in the field I think it would be appropriate to at least somewhere document what the residuals are for the graphs presented in the paper (i.e. number of $|Z| \geq 3$ residuals, and magnitude of worst residuals). The many readers who expect to find this information will then be able to find it.

RESPONSE 32: While we value the stimulating discussion with referee #2, we perceive that it may extend beyond the scope of this paper in its present form, and we respectfully disagree with his/her line of argumentation. We understand that a model can be better than others and still fit the data poorly. But the present situation is different. In RESPONSE 30, we show that the model from Maier and colleagues returns $|WR|$ values superior to 3. In fact, all the $|WR|$ presented in their original publication for all their models are superior to 3.4. It, thus, appears that none of their models can be strictly speaking considered as an absolute fit. Even if we had to consider a relaxed threshold, the following demonstrates that the new model presented in this study provides a better description of the data (p -value $< 1e-5$; see RESPONSE 36).

Specifically, considering the extended dataset presented in this study, which is devoid of biases due to pseudohaploid population groups (see RESPONSE 30), the less favorable $|Z$ -score| for our model is 1.92, while that for the model reported by Maier and colleagues is 5.9. The absolute score of our model is 8.81, and that from Maier et al. is 112.9.

POINT 33: Working within the framework of 8 admixture events, the authors now identify a new graph that fits significantly better than the Maier graph, according to the likelihood value contrast. This is some valuable progress, but I don't think it completely negates the concerns raised by Maier et al. Maier still showed that there exist alternative graphs that, in an absolute sense (using the residuals, as discussed above), largely explain the data. I don't think Maier et al. necessarily wanted to strongly push that the particular graph they identified is the true one, or even the best-fitting one that can be identified; indeed, their paper states "we are not arguing here that our alternative model is right"; indeed we are nearly certain it is wrong in important aspects; - but rather the more technical point that the existence of quite different topologies that are all compatible with the data highlights the limitations of admixture graph inference. The authors still do not really seem to want to acknowledge this technical point, and consequently downgrade the weight they give to this specific line of analysis. They are mainly just focussed on chasing better-fitting graphs.

RESPONSE 33: It is true that Maier and colleagues question their own graphs, and their methodology of admixture graphs itself. In this context, we completely align with their view: we firmly believe that cautioning about the limitations of admixture graphs is paramount to avoid misleading inference. This is in fact what we are doing here, in our previous round of revisions, and what we already did in Librado et al. (2021) (*Identifying discrete populations and modelling admixture as single unidirectional pulses, however, was highly challenging given the extent of spatial genetic connectivity*). The original choice to write a Rebuttal to Maier et al. was motivated by the fact that their article explicitly discussed some population graphs more than others, which may be falsely understood as a formal demonstration by most readers. For example, they clearly suggest that horses fueled the human migrations from the steppe:

"Equally important, however, is our finding that there are plausible models that are inconsistent with other inferences in Librado et al., 2021. (Table 2). For example, 13 of these 16 models are inconsistent with the suggestion that there was no gene flow connecting the CWC group and the cluster maximized in the Western steppe (DOM2, C-PONT, and TURG) (Figure 3—source data 7j-r). In the eight-admixture-event best-fitting plausible model (Figure 3b and the second model in Figure 3—source data 7j), CWC actually derives appreciable ancestry from the early domestic horse lineage (DOM2) associated with the Sintashta culture to the exclusion of the more distant Yamnaya-associated TURG and C_PONT horses. This scenario presents a parallel to the one observed in humans, with individuals associated with the CWC receiving admixture from Steppe pastoralists albeit in different proportions: ~75% for humans, versus ~20% in horses. These models specifying a substantial Steppe horse contribution to CWC horses would weaken support for the inference in Librado et al., 2021. That 'Our results reject the commonly held association between horseback riding and the massive expansion of Yamnaya steppe pastoralists into Europe around 3000 BC.'

While dismissing all other analyses conducted in Librado et al. 2021, such as Structf4, developed to overcome the caveats of admixture graphs, and consistently supporting otherwise:

"We are not aware of other lines of evidence in the paper (apart from the fitted AG) that support the claim of no Yamnaya horse impact on CWC horses."

We firmly believe that the role that horses played on human mobility is of paramount importance to understand our past, and that any claim should be based on solid evidence. The work presented in the three rounds of revisions shows that their own methodology, either Admixtools2 or qpAdm, fully rejects their own interpretation, a position further supported by the new data included in this study. Additionally, the posterior distribution provided by AdmixtureBayes, a Bayesian method superseding admixtools2, clearly demonstrates that the graph suggested by Maier et al is at best unlikely, compared to those that are really top-scoring. Therefore, it is unfair to reduce our study as an attempt to 'simply chasing best-fitting graphs'. It should be viewed instead as a genuine search for identifying the most credible population history accounting for inherent the uncertainty associated with graph inference, through a proper Bayesian framework. The analyses presented in our study, as well as those presented in the different rounds of reviews (and now excluded to avoid confrontation with Maier et al. (2023)), demonstrated that it is simply not acceptable to acknowledge: "different topologies that are all compatible with the data", if these topologies include introgression from DOM2 into CWC horses.

POINT 34: Following the same reasoning, I don't find the authors' discussion in the supplements on how Maier et al. actually identified another graph that might be a slightly better fit ("LL = 36 vs. 37") particularly illuminating, and slightly beside the point. To me, those two graphs both fit the data quite well, and it's not so important which one has a slightly better fit than the other.

RESPONSE 34: See RESPONSE 31 and RESPONSE 32. We shall insist that this debate does not depend on personal opinions, but should instead be based on solid statistical grounds. In statistical inference, there is simply no justification to select a complex model fitting the data worse than a simpler one.

POINT 35: On another point, in my view a valuable finding from Maier et al. was that the first graph published in Librado et al. 2021 (figure 3b in that paper), with only 3 admixture events, was in fact a very bad fit to the data. However, the authors still defend this graph to some extent (in the supplements), with the argument that the parts of the graph that fit poorly are not so relevant. I don't think this is a very viable line of argument. I think it would be fair for the authors to just acknowledge that this was a poorly fitting graph, and that three admixture events among those 10 populations is simply not enough to model the complexity of horse history. If they still actually want to defend this original graph, at the very least I think they should then apply the same treatment and scrutiny that they apply to the Maier et al graph. E.g. perform the same direct contrast (using `qpgraph_resample_multi`) between this original graph and their new favored 8-admixture-edges graph; if the original graph is significantly worse, I think that should be acknowledged. If it's not worse, that's interesting to report too.

RESPONSE 35: Already in Librado et al. 2021, we openly presented the residuals of the fitting graphs, hence, admitting poor fit for the lineages with ghost contributions (ELEN and NEOANA). We never stated that these parts of the graph are *not relevant*, but that they are too uncertain, owing to the presence of unsampled data.

We wished that we could simply apply `qpgraph_resample_multi` to contrast Librado 2021 vs. Maier et al. 2023, but this function is meant to compare models of equal complexity (eg. with eight admixture edges). The score would be always best for the most complex models otherwise, regardless of the bootstrap pseudo-replicate.

We alternatively devised a solid statistical strategy to test our statement. In particular, we fitted both competing models (OrientAGraph *versus* Maier et. al 2023) to the extended dataset generated in this study, and asked the `qpgraph` function of the Admixtools2 package to return all the residual f values (encompassing all the f_2 , f_3 and f_4 permutations), including their associated p -values (likelihood for the fit of the model to each permutation). With this, and the number of parameters to be estimated from each model, we then calculated their Bayesian Information Criterion (BIC) scores. The BIC method is widely used in statistics to identify the best fitting model, when competing alternatives are not nested. It explicitly penalizes for increasing model complexities. In brief, the lower the BIC value, the better.

Fitting the graph from Maier and colleagues to the extended dataset presented in this study returns a BIC value of 76,798.44. This reduces to 29,717.63 while excluding f_4 permutations including ELEN and NEOANA. Doing the same exercise for the graph presented in Librado and colleagues (2021) returns lower (better) BIC values of 58,573.44 and 22,661.63 respectively. Therefore, the need for increased complexity, up to 8 migration edges, is statistically unjustified according to the BIC criterion. Note RESPONSE 30 presents a graph with $|WR| < 2$, meaning that we could probably still reach $|WR| < 3$, if eliminating one (or more) admixture edges.

We also highlight that Maier and colleagues did not apply the same treatment to the data as Librado et al. 2021 (see RESPONSE 30). The request for us to apply the same treatment as Maier and colleagues raises concerns, as they did not adhere to a consistent methodology in the first place when (1) changing group labels, (2) choosing different block sizes for jackknifing, and (3) used a different data matrix than that originally released.

We shall insist that these concerns are not developed in our revised manuscript as per the editor's request, but are only presented in response to the reviewer's comments for the sake of scientific discussion and transparency.

POINT 36:

“population graphs that fit extant and new data significantly better (p -value < 0.01)”. I think it's better to give the exact p -value obtained, rather than just a value it is smaller than (likewise in the supplements where this p -value is referred to). There's quite a big difference between say $p=0.009$ and $p=1e-10$.

RESPONSE 36: It is perhaps worth reminding that their function relies on bootstrapping. Achieving a resolution in the order of $1e-10$ would require to conduct 10 billion bootstrap pseudo-replicates, which is

computationally unfeasible within the time frame of this revision. It is also absolutely unnecessary. In our previous version of the manuscript, we had performed 100 bootstrap pseudo-replicates, and none favored their model, hence, the p -value indicated below 0.01. We have now expanded this to 10,000 pseudo-replicates, once again finding that none favored their model (p -value $< 1e-5$). Below is the score distribution across the 10,000 bootstrap pseudoreplicates, for the model reported by Maier et al. 2023 (salmon) vs. for our model (turquoise). Our graph has a significantly lower score, indicating better statistical fit.

POINT 27: The main text Fig 2 displays new results introduced in revision: an admixture graph with a larger number of populations, including more relevant European populations, with 13 admixture events. I think it's a good idea to include these populations of more relevance to Corded Ware horses. But in moving to much larger graphs than previously considered by the authors themselves or by Maier et al, it's tricky to present these results in the context of a rebuttal to Maier et al. The main text does not clearly distinguish between the Maier rebuttal in the strict sense and this new extended analysis, and many readers will likely get the impression that the authors went out and identified this new Fig 2 graph that is simply better than the Maier graph. But one cannot actually say that it's better, because the set of populations used is different.

It would be more appropriate, and transparent to the reader, to present this new analysis along the lines of “we expanded the admixture graph search to also include more proximal European horse populations of potential relevance to Corded Ware horse origins, and the best graph identified does not support ...”.

RESPONSE 27: We agree with the reviewer that the previous phrasing could be ambiguous, and we apologize for it. In the revised version of the manuscript, due to space constraints, we have transformed the former Figure 2 as Extended Data Figure 3. We have also clarified the main text, as suggested by reviewer #2, to ensure maximum transparency.

POINT 28: Related to the above, the legend of Extended Data Figure 4 states “The drift and admixture estimates are based on our extended dataset, but the underlying topology was found to also fit the original dataset from Librado et al. 2021 better than the graph presented by Maier et al. 2023”. I'm guessing the “underlying topology” here refers the smaller graphs in Extended Data Figures 3 and/or 4 2013; if so it would be good to refer to refer to those here, so that readers can follow.

RESPONSE 28: Reviewer #2 and his/her interpretation of our previous phrasing is correct. More specifically we referred to Extended Data Figure 4. Conditioning on 8 migrations and using the extended dataset, we can find graph with Admixtools2 (see RESPONSE 30), which is slightly better than the one presented in the previous round of reviews, and considerably better than that presented by Maier et al. (2023). We have clarified the legend accordingly, to avoid any misinterpretation (Extended Data Fig 4).

POINT 29: Also, simply saying “underlying topology” kind of gives the impression that the topology of the smaller graph EXACTLY matches the larger one for the populations that overlap, but I don’t think that’s quite the case. There’s some degree of subjective judgement involved in saying how similar they are. It might be more appropriate to describe this along the lines of e.g. “graphs with a similar core topology but fewer populations were found to also fit” [reference figures].

RESPONSE 29: We thank the reviewer for his/her suggestion, we clarified all this in the revised version of our manuscript.

POINT 30: “predicted a covariance matrix deviating negligibly from the empirical one, according to allele frequency differences of $\sim 1.5e-7$ at most”. This is some kind of absolute assessment of graph fit, which is potentially valuable. But the value has no unit, and I don’t know how to interpret it. It seems like a small number, but I have no frame of reference for it, and most other readers won’t either.

RESPONSE 30: That specific value referred to changes in allele frequency, and we apologize if this was unclear. A hypothetical change of 0.1 would mean for example that the allele frequency was 0.2 in the observed covariance matrix, while 0.3 in the covariance matrix predicted by the model. Our value was neglectable, meaning that our AdmixtureBayes graph fits the data almost perfectly.

We, however, understand that this metric for assessing absolute graph fit may be unfamiliar to most readers. We, thus, evaluated the AdmixtureBayes graph, extended to the 14 populations presented in Extended Data Figure 3, using Admixtools2. Despite the increased complexity of this dataset, and the statistical differences between AdmixtureBayes and Admixtools2, we remarkably found $|WRs| = 3.34$. This $|WR|$ is comparable (lower) than that reported by Maier et al. 2023, for 10 populations only. We added this $|WR|$ value in the corresponding Extended Data Figure 3.

POINT 31: I would encourage (but not require) the authors to send their supplementary section on “Rebuttal to Maier et al. 2023” to the authors of Maier et al. 2023 for comments before publication. It would be understandable to not want to do that, but it would probably benefit the scientific process to give those authors an opportunity to identify anything that they would view as not appropriately representing their work.

RESPONSE 31: Since (1) the editor did not insist on contacting Maier et al., and as (2) we extensively exchanged with Maier et al. at the time they prepared their publication (suggesting areas in which we thought their study could be improved but were eventually dismissed), and (3) now remove the ‘*Rebuttal to Maier et al.*’ section from the Supplementary Information, we did not see the point to further request comments from the co-authors of the Maier et al. publication.

POINT 32: The authors introduce new qpAdm analyses, which I think are very valuable and seemingly much stronger than the admixture graph analyses. If I were the authors, I would decrease emphasis on the admixture graphs and increase emphasis on the qpAdm results (including swapping these for main figure attention). But again, it is the authors’ paper to write.

RESPONSE 32: Thanks for the suggestion. We do not think that we have to favor one method over the other as both are fully consistent. We now, however, present the admixture graphs in a manner not focused on rebutting Maier et al. 2023, centering the discussion in our own results, as suggested by the editor.

POINT 33: In any case, I think more information on the qpAdm analyses are needed. Which version of qpAdm was used should be indicated, and any relevant parameters used (including allsnps=YES or NO?). I think the table S2 should include the standard errors of the ancestry proportions, which are returned by the software. It seems the authors have only included in the table models that fit, but it is equally (or maybe even more) important to document the models that don’t fit, including the relevant ones referred to in the main text (“All the remaining two-way models were explicitly rejected ($P < 0.01$)” I think the rejection of qpAdm models involving steppe ancestry is key to the paper, and so it would be good to fully document them in this table and show precisely how strongly they are rejected (that is their exact p-values). There’s really nothing stopping the authors from comprehensively listing models in this table. E.g. for two-source models enumerated from 15 populations, that’s only 105 possible models; all

could be listed, and e.g. sorted by p-value. For three-source models it's a bit more (455 models), but this is what supplementary tables are for arguably. I think the authors could also perhaps think of some graphical representation summarizing these results, i.e. plotting the p-values of different models in some way and highlighting those that have steppe sources vs those that don't. This should hopefully quite clearly show that the steppe models are very strongly rejected.

RESPONSE 33: We used the last qpAdm version (1520), within the last Admixtools package (7.0.2). We used allsnps=NO, because using whole genomes allowed us to retain millions of nucleotide transversions despite strict filtering. Table S2 includes now the number of SNPs analysed, together with the corresponding standard errors for each qpAdm model, as suggested by referee #2. We thank the reviewer again because this shows that the admixture proportions provided by Maier and colleagues are well outside the confidence interval estimated by qpAdm.

POINT 34:

“negligible levels of steppe (~1.7%) or DOM2 (0.003)”: missing percentage sign for second number.

RESPONSE 34: Fixed.

POINT 35:

“we mapped the genetic ancestry characteristic of horse populations living across the steppe before the expansion of DOM2”. This paragraph does not explain how these results were obtained (f4-struct? qpAdm? Admixture graphs?), and does not refer to any figure or table.

RESPONSE 35: We apologize for overlooking this. The paragraph is now rephrased as follows (lines 280-283, page 5):

“To investigate this, we mapped the genetic ancestry identified by Struct-f4²⁴ as characteristic of horse populations living across the steppe before the expansion of DOM2 (C-PONT, TURG, and NEONCAS; ~5,616-2,636 BCE; Fig. 1b).”

POINT 36:

“Therefore, the spread of steppe-related horse genetic ancestry into Europe must predate ~14,646 BCE”. Here, it will be obvious to many readers that this spread of steppe-related ancestry would just reflect movement among wild horses, but actually stating this explicitly might help some other readers who might otherwise think this is some kind of early human activity.

RESPONSE 36: Agreed. The sentence now reads (lines 292-295, page 5):

“Therefore, the spread of steppe-related horse genetic ancestry into Europe must predate ~14,646 BCE, which is considerably earlier than any claimed evidence for horse husbandry³ and, thus, occurred through natural contacts between wild populations, most likely dispersing in the aftermath of the Last Glacial Maximum (~24,000-17,500 BCE)²⁵”

POINT 37:

Slight geographical terminology nit-pick, unrelated to the scientific content of the sentence: “However, we predict that the geographic origins of CWC horses is exclusively within Europe”. The Yamnaya were from eastern Europe, if we take Europe to end at the Ural mountains, as is commonly done. Saying “exclusively within Europe” does thus not exclude the Yamnaya. Perhaps better phrased as “western to central Europe”, or something like that.

RESPONSE 37: Agreed. The sentence now reads (lines 244-246, page 4):

“However, Locator²¹ analyses predict that the geographic origins of CWC horses is exclusively within Central Europe (Extended Data Fig. 4cd).”

POINT 38:

Another nit-pick on the same sentence: “However, we predict that the geographic origins of CWC horses is exclusively within Europe”. Simply saying “we predict” is not very informative about what was done. What method/result this actually refers to needs to be indicated at least briefly in some way.

RESPONSE 38: We apologize for overlooking this in the first place. The requested information was added, as follows: *“However, Locator²¹ analyses predict that the geographic origins of CWC horses is exclusively within Central Europe (Extended Data Fig. 4cd).”*

POINT 39:

“suggest that the last Ice Age may have impacted horse generation times, though to a lesser extent than domestication (Fig. 4A).” The Last Ice Age lasted ~100,000 years, so it’s not clear what the authors are referring to here. Perhaps they mean to say “the end of the last Ice Age” as there is a slight increase in the curve after 13,000 BCE? Could also be useful to indicate that in the figure, to more clearly connect the statement and the figure, and also all readers might know necessarily know when the Ice Age ended.

RESPONSE 39: We have added the Last Glacial Maximum (LGM, 19-26 ky BP, Before Present) in what was formerly Figure 4 and is now Figure 3, as this is the only period of the Last Ice Age that can be visualized in this figure.

Referee #3 (Remarks to the Author):

Disclaimer: The following review focuses on the radiocarbon measurements and its application.

POINT 40: I think the publication is a very nice example of a comprehensive study combining genetic information (of horses) with age relationship obtained mainly obtained from radiocarbon ages on the very same samples in order to obtain a detailed history of (horse) evolution.

The amount of samples measured with ^{14}C is impressive! Also, it seems the quality of measurement is very high, the given errors very low (often 15 - 20 years only!). While there is only one repeated sample (one could wish some more replication for such an important dataset), they seem to be consistent, as can be seen on same context samples.

RESPONSE 40: We thank the reviewer for acknowledging the quality and importance of our data set, which we hope will become a useful resource for the scientific community in the future.

POINT 41: Minor critics:

No information on how the radiocarbon ages were obtained are given. I suggest writing a short paragraph for the supplementary information. How were the samples treated? Can you give references? While most samples are independently dated and can be treated in a consistent way, there are some samples, where I think no independent measurements are available. I think a clear differentiation would be good to have (see also comments below).

RESPONSE 41: We have developed the methodology underlying radiocarbon dating, both in the main Methods section to be published online, and the Supplementary Information (section 'Radiocarbon Dating'). They read respectively as follows:

Methods:

"A total of 140 new radiocarbon dates were obtained in this study, all of which at the Keck Carbon Cycle AMS Laboratory, UC Irvine (Table S1). Collagen was extracted and ultra-filtered following mechanical cleaning of ~200 mg of cortical bone. Radiocarbon dates were calibrated using OxCalOnline⁴⁹ and the IntCal20 calibration curve⁵⁰"

Supplementary Information:

"Radiocarbon Dating

The vast majority of the specimens analyzed in the present and previous studies was radiocarbon dated at the Keck AMS laboratory, University of California Irvine (USA), with the following methodology. Samples of cortical bone were cleaned mechanically and aliquots of ~200mg were crushed to mm-sized chips. If contaminating conservation materials were present, samples were sonicated in acetone, methanol and ultrapure MQ water in a water bath cooled to well below the melting point of collagen. Bone was decalcified overnight at room temperature, using a measured amount of 1N HCl just sufficient to dissolve all of the bone mineral, if no collagen was present. The demineralized samples were washed with MQ water and gelatinized overnight at 60°C and pH 2, ultrafiltered in precleaned Vivaspin 15 devices to select the >30kDa molecular weight fraction, and freeze dried overnight. Aliquots of 2mg of collagen were combusted under vacuum in quartz at 900°C with CuO and silver wire and the resulting CO₂ was cryogenically purified and graphitized on Fe by hydrogen reduction for ^{14}C measurement by AMS on an NEC 0.5MV. 0.7mg collagen aliquots were sealed in tin capsules and flash combusted in a Fisons NA1500NC elemental analyzer interfaced to a Finnigan Delta Plus isotope ratio mass spectrometer for elemental analyses and $\delta^{13}\text{C}$ and $\delta^{15}\text{N}$ measurements.

Additionally, horse remains from Ulaan Tolgoi and Zunii Gol⁴⁰ and Biluut^{41,142} were dated using accelerator mass spectrometry (AMS) at Beta Analytic following laboratory protocols (<https://www.radiocarbon.com/carbon-dating-pretreatment.htm>), while horse remains from Zeerdegchingiin Khoshuu and Zuunkhangai were dated at the AMS Laboratory at the University of Arizona, following protocols outlined by Taylor et al (2017)¹⁴³. Briefly, samples were surface cleaned and rinsed for one hour in an ultrasonic water bath. Exterior and cancellous portions of the sample were removed with a sterilized drill bit, before crushing and sieving to produce a 500 mg sample with uniform particle size (0.5e1 mm diameter). The bone powder was then loaded into a flow cell (a modified chromatography column) and, using a computer controlled pumping system, demineralized using 0.5 M hydrochloric acid, extracted with 0.1M sodium hydroxide (NaOH) to remove humic and fulvic acids, and then rinsed with weak acid (0.001M HCl). After ensuring each sample had a final pH of 3, the insoluble collagen fraction was removed from the flow cell and gelatinized by heating to 70°C for 20 h. Each gelatinized sample was cooled to

room temperature, filtered through a 0.45mm glass microfiber filter with polypropylene housing, and then lyophilized. Carbon isotope stable ratios of the samples were measured by the Accelerator Mass Spectrometry laboratory at the University of Arizona, Tucson, AZ. Finally, a few specimens from Mongolia were radiocarbon dated following the bone collagen extraction procedure described in Zazwko et al. (2019)¹⁴⁴. Here, AMS measurements were performed at LMC14 and LSCE using the (Artemis platform and ECHO-MICADAS facilities, respectively. For the Artemis-dated sample, CO₂ was extracted and purified by combustion at the MNHN laboratory, then sealed in a glass tube. Graphitization was carried out at LMC14. Collagen samples dated by ECHO-MICADAS were combusted then reduced to graphite form at LSCE via AGE3 equipment before measurement on ECHO-MICADAS according to the procedure described in Zazwko et al. 2019¹⁴⁴.

Table S1 provides raw radiocarbon dating measurements (uncal. BP, Before Present), and calibrated dates, resulting from calibration in OxCal online (<https://c14.arch.ox.ac.uk/oxcal/OxCal.html>), using the IntCal2020 calibration curve⁵⁰. Dates generated as part of this study are flagged ('1') on the column labelled 'ThisStudy' on Table S1, section 'Radiocarbon Dating Information'. Likewise, those previously published are flagged on the column labelled 'PreviousStudy', with reference to the original publication reporting the date for the first time (column 'Reference', section 'Radiocarbon Dating Information'). Finally, the column 'Lab Reference' reports the official name of the radiocarbon date considered. »

POINT 42:

The authors seem to work primarily with mean ages for individual samples instead of the age range obtained from the calibrated age ranges (for simplicity). This is not always considered the right procedure and can be problematic, when high temporal resolution is interpreted. Having said that, I think the publication focuses primarily on the interpretation based on medium to low time resolution and thus the presented interpretation is valid based on the age information given.

RESPONSE 42: Following the reviewer #3's suggestion, we have evaluated the impact associated with the uncertainty of radiocarbon dating. We assumed that each specimen had a date randomly sampled within the confidence interval provided by radiocarbon dating, according to a uniform distribution. We repeated 20,000 times this process of random sampling each date, and for each replicate, we re-calculated the GAM regression model to retrieve its corresponding adjusted R². The distribution of adjusted R² values is:

As expected under the assumption that the midpoint is a truly informative indicator for the age of the sample, our observed adjusted R² (0.894) falls almost exactly at the 50% quantile of the distribution. The best replicate provided limited improvement to the GAM regression, with the maximum adjusted R² reaching 0.898. Re-plotting as a function of this best replicate returns an almost equivalent relationship with the generation time estimates:

The most important difference is still modest and pertains to the oldest specimen (KB218). This is precisely the one with the widest radiocarbon date confidence interval, as expected given the lower resolution of 14C dating in deeper times. The 1,000 replicates that maximize the GAM adjusted R^2 value suggest slightly younger age for this individual, placing its date to about ca. 42,000 ya instead of 44,495 ya.

Interestingly, the posterior distribution of the calibrated radiocarbon age for this specimen also indicates that the most likely age value is between 43,000-41,000 cal. years BCE.

The bootstrapping procedure implemented above indicates that the uncertainty around the dating of not only this specimen, but all specimens, does not impact our conclusions.

In the following are some comments on the listed radiocarbon measurements.

Your own measurements:

POINT 43: - For sample GVA9046_Mon_716 no Lab number was given (what is standard and strongly recommended for ^{14}C measurements). By the way, you call the column in the excel sheet "UCIAMS", what is a lab prefix - however, you have also listed numbers from other labs. The supplementary information only indicates that the date may come from ref. 91 (that I could not access). Is the date directly based on bone dating? I suggest giving a reference also in the excel list with together with the dating (one might have the impression that you actually dated the sample, what I think is not the case).

RESPONSE 43: We apologize for not providing explicit reference to all the original radiocarbon dates present in Table S1. The revised Table S1 now tracks all such information in the column labeled 'Lab Reference'. For example, sample GVA9046_Mon_716, radiocarbon dated at the Artemis platform, is now referenced to SacA47041. The newly added 'Reference' column points to the original reference where this radiocarbon date was released first (here, Zazzo et al. (2019)). The same holds true for samples M17x152x1_Mon_m80, NB46_Rus_m2007, M17x140x1_Mon_m1116, GVA9042_Mon_m1068, M17x188_Mon_m1010, GVA9035_Mon_m1024, M17x159_Mon_m1136, GVA9037_Mon_m1013, M17x126x1_Mon_m942, GVA9036_Mon_m1013, Batagai_Rus_m3136, BPTDG1_Fra_m12493, Rus30x31_Rus_m3435 and M17x174_Mon_m1145. As a result, those samples not directly radiocarbon dated can be easily identified as not associated with an explicit reference, and pointed as such (i.e. 'Dating inferred from archaeological Context and/or associated radiocarbon dates') in two additional columns labeled 'Yes|No' and 'Notes' in the table section presenting 'Radiocarbon Dating Information'. 'Yes' indicates that direct radiocarbon dates are available.

POINT 44:

- I only counted 122 radiocarbon measurements in the excel list, while you write in the text there were 149 new measurements performed.

RESPONSE 44: We have added a new column called 'This Study' to Table S1 in the table section presenting 'Radiocarbon Dating Information'. A value of 1 indicates that the underlying radiocarbon date is presented in this work for the first time, while a value of 0 indicates that it is not the case. There are 140 new radiocarbon dates presented in our study.

POINT 45:

- There are five samples from Denmark (GIN1020_Den_m3050, GIN1055_Den_m3050, GIN396_Den_m3225, GIN489_Den_m3225, GIN561_Den_m3225), where no radiocarbon measurement was given, but still age limits (Min / Max) were given. Your supplementary information gives a citation, where bones were dated and it looks like the age limits come from. Also here I suggest giving a reference in the excel together with the dates. Maybe even one or the other bone was directly dated - then you could also give the radiocarbon age? (The advantage of a radiocarbon age is, that it can be re-calibrated when a new calibration curve appears). What is maybe most important here is, that the five samples are not independently dated (with some possible implication on modelling).

RESPONSE 45: See also RESPONSE 43 and RESPONSE 44. In addition to Table S1, we have clarified the Supplementary Information when presenting specifically the material from Ginnerup, Denmark. It reads as follows, which explains why the specimens can be confidently associated with a context-based date of approximately 3,000 BCE:

“Ginnerup is located on the Djursland peninsula in eastern Jutland, Denmark. At the time of occupation, it was a coastal site on a promontory, bordering the c. 1 kilometer-wide Kolindsund sound located to its south and a narrow branch of the sound on its eastern side. The surrounding, undulating landscape to the north, east and west is made up of sandy moraine and meltwater plateaus between 15 and 60 m.a.s.l. Samples GIN1020_Den_m3000 and GIN1055_Den_m3000 are from horse remains excavated in layer 7 in structure A4, a natural depression filled with an undisturbed sequence of archaeological layers⁷⁵. Layer 7 has been dated to 3000-2920 BC by modelling of 11 ¹⁴C-dates from a stratigraphy comprising four layers (see Klassen et al. (2023)⁷⁵, pages 47-49, where individual dates and details on sample treatment and modelling statistics are given). The archaeological content of this layer derives from the later part of a transitional phase between the Neolithic Funnel Beaker (TRB) and Pitted Ware (PWC) cultures, comprising TRB pottery decorated in MN A II/III and Ferslev style and PWC pottery. Samples GIN396_Den_m3000, GIN489_Den_m3000 and GIN561_Den_m3000 all were obtained from horse remains from context K68 in structure A1 (ditch with numerous recuttings, unpublished, not yet dated by ¹⁴C). K68 represents a deposition of shells containing TRB (MN A II) and PWC pottery. A slightly higher percentage of TRB vs. PWC elements of the pottery from K68 compared to that from layer 7 in structure A4 indicates a slightly older date^{75,76}.”

POINT 46:

Measurements of others:

Some similar critics as for your own samples are given here.

- For a relatively large number of samples, there is a calendar age given, but no ¹⁴C measurement (for samples from Librado et al. 2021).

RESPONSE 46: The reviewer is correct. We have now indicated all instances in which the dates associated with samples correspond to direct radiocarbon dates of the material analysed (‘Yes’ in the Table S1 ‘Yes|No’ Column from the ‘Radiocarbon Dating’ section). We have also filled up an additional column labeled as ‘Notes’, to further track those specimens whose dates are inferred from archaeological Context, or radiocarbon dates associated with radiocarbon dates of material excavated from the same archaeological layers (‘Dating inferred from archaeological Context and/or associated radiocarbon dates^o’).

POINT 47:

- There are also samples with ¹⁴C dates, but with no Lab number given

RESPONSE 47: See RESPONSE 43.

POINT 49:

I have to admit, the information is also missing in the original publications, but it would be good if more information could be collected here for completeness. I think it can be crucial to know, which samples were dated independently (and are they all radiocarbon dated?).

RESPONSE 49: We have collected the requested information to enhance data traceability. See RESPONSE 43.

Reviewer Reports on the Second Revision:

Referees' comments:

Referee #2 (Remarks to the Author):

The authors have produced a very rigorous revision, which I think has benefitted the paper. I think the paper is now more readable and presents the often quite technical material in more accessible way to readers. The new presentation of the admixture graphs results is now much easier to follow, with less potential for confusion between different graphs and analyses. This includes the rewritten corresponding supplementary materials section, which I now find less challenging to digest. I also thank the authors for going down the admixture graph rabbit hole with me and engaging in detail with all of my comments! I have no further comments to make at this stage, and I congratulate the authors on their great work.

Referee #3 (Remarks to the Author):

Dear authors,

I am very pleased with the detailed response on my review!

The requested information about the radiocarbon determinations is given. You even managed to give additional information for previously published radiocarbon analyses, where information was missing! I think full transparency on how the ^{14}C results were obtained is now given!

I am also happy about your statistical analysis to demonstrate that you can use simply mean ages (instead of an age distribution function) for the interpretation of your results.

I have no further requests.